
# Measurement report: Simultaneous multi-site observations of VOCs in Shanghai, East China: characteristics, sources and secondary formation potentials

Yu Han[1], Tao Wang[1], Rui Li[1], Hongbo Fu*[1, 2, 3], Yusen Duan[4], Song Gao[4, 5,,] Liwu Zhang[1], Jianmin Chen[1, 2]

[1]Shanghai Key Laboratory of Atmospheric Particle Pollution and Prevention, Department of Environmental Science & Engineering, Fudan University, Shanghai, 200433, P.R. China; [2]Institute of Eco-Chongming (SIEC), 20 Cuiniao Road, Chenjia Town, Chongming District, Shanghai 202162, China; [3]Collaborative Innovation Center of Atmospheric Environment and Equipment Technology (CICAEET), Nanjing University of Information Science and Technology, Nanjing 210044, P.R. China; [4]Shanghai Environmental Monitoring Center, National Environmental Protection Shanghai Dianshan Lake Science Observatory Research Station, Shanghai, 200235, China; [5]School of Environmental and Chemical Engineering, Shanghai University, Shanghai, 200444, China

*Correspondence to*: Hongbo Fu (fuhb@fudan.edu.cn)

**Abstract.** Volatile organic compounds (VOCs) have important impacts on air quality, climate, and human health. In order to identify the characteristics, sources and secondary formation potentials of the atmospheric VOCs varying with land use types, a concurrent multi-site observation campaign was performed at the supersites of Shanghai, East China, in the first three months of 2019. During the observation period, the average VOC concentrations were 21.39, 21.36 and 11.93 ppb at the Jinshan (JS), Pudong (PD) and Qingpu (QP) sites, respectively. The predominant VOC category was alkanes (49-61 %), followed by aromatics (11-21 %), alkenes (10-15 %) and alkyne (8-14 %) at the above sites. The sampling sites exhibited distinct diurnal variations and "weekend effects" of VOCs. The VOC concentrations increased by 27.15, 32.85 and 22.42 % during the haze events relative to the clean days during the measurements. Vehicle exhaust was determined as the predominant VOC source. The second largest VOC contributor was identified as industrial production at the JS and PD sites, while fuel evaporation was the second important source at the QP site. High potential source contribution function (PSCF) values appeared in the northeastern and northern Shanghai near the sampling sites, suggesting the strong local emissions. The secondary organic aerosol formation potential, mainly contributed by the aromatics, was higher at the JS site (1.00 μg m$^{-3}$) than those at the PD (0.46 μg m$^{-3}$) and QP (0.41 μg m$^{-3}$) sites. The VOCs-PM$_{2.5}$ sensitivity analysis showed that the VOCs at the QP site could be more sensitive to the PM$_{2.5}$ concentration relative to the other two sites. The ozone formation potentials (OFP) were calculated to be 50.89, 33.94 and 24.26 ppb during the campaign at the JS, PD and QP sites, respectively. The VOCs-O$_3$ sensitivity indicated that the VOCs-S$_{O3}$ values varied at the different sites and were primarily controlled by the alkene-related reactions. Alkenes and aromatics are thus the key concerns in controlling the VOC-related pollution of SOA and O$_3$ in the diverse districts of Shanghai. The findings of this study provide new insights into the accurate air-quality control at a city level in China due to a wide variety of land-use types. The results shown herein highlight that the simultaneous multiple-site measurements in the megacity or city cluster could be more appropriate to fully


understand the VOC characteristics relative to a single-site measurement performed normally.

## 1. Introduction

Serious air pollution in China is currently characterized by high level of ozone ($O_3$) and heavy loading of fine particulate matters (PM), both of which formation are greatly contributed by the atmospheric volatile organic compounds (VOCs) (Carter, 1994; Liu et al., 2008; Yuan et al., 2013; Lu et al., 2018; Ma et al., 2019; Yu et al., 2021). Atmospheric VOCs function as an important precursor of $PM_{2.5}$ (PM with an aerodynamic diameter less than 2.5 μm), contributed greatly to the haze formation. In brief, the primary VOCs can be oxidized by HO·, $RO_2$· and $O_3$ to produce secondary VOCs, which further transforms into SOA *via* a series of atmospheric processes (Odum et al., 1997; Ng et al., 2007; Heald et al., 2020). The secondary particles have strong influences on the atmospheric radiative forcing and climate (Sadeghi et al., 2021). On the other hand, VOCs reacts with $NO_x$ and radicals *via* an array of photochemical processes to produce $O_3$. Photochemical Assessment Monitoring Stations (PAMS) have confirmed 57 precursors of $O_3$, including $C_2$-$C_{10}$ alkanes, alkenes, alkynes and aromatics (US EPA, 1990). Atmospheric $O_3$ has profound impact on the atmospheric oxidizing capacity, and has been widely proved to affect agriculture production and ecological conditions (Liu, 1987; Carter, 1994; Mousavinezhad et al., 2021; Sadeghi et al., 2021). Additionally, VOCs, along with derived secondary compounds, can damage human health and lead to many diseases such as cancers, respiratory and nervous system symptoms (Rumchev et al., 2007; Amor-Carro et al., 2020). Therefore, the VOC-related studies provide scientific-based information for the decision-makers who draw up strategies to control $PM_{2.5}$ and $O_3$.

The temporal and spatial variations of VOC characteristics have been widely studied. In terms of seasonal variations, previous reports showed that colder seasons presented higher VOC levels than the ones at the warm seasons (Kumar et al., 2018; Sun et al., 2020; Debevec et al., 2021). For instance, Sun et al. (2020) found that the VOC pollution in the wintertime of Xi'an, China, was almost twice heavier than that in the summer. Such phenomenon could be mainly attributed to the fact that the higher emissions from anthropogenic sources including combustion and paint solvents occurred during the winter than those in the summer (Li et al., 2020). Furthermore, weaker atmospheric stability accelerates dilution and convection of the pollutants during the summer time (Monod et al., 2001; Dumanoglu et al., 2014). Besides, the elevated solar radiation and temperature during the summer were also responsible for the rapid VOCs loss through photochemical reactions (Lai et al., 2013; Kumar et al., 2018). The evolution of VOC concentrations from the clean days to haze events were also widely concerned. The occurrence of haze pollution was always accompanied by the rapidly increased VOC concentrations. For instance, in the Chinese cities, including Beijing (Wu et al., 2016); Shanghai (Han et al., 2017; Liu et al., 2021) and Wuhan (Hui et al., 2019), the VOC concentrations during haze pollution were 2-5 times higher than those in the clean days, which could be attributed to the stagnation condition hindering the vertical and horizontal dispersion of the VOC pollutants. In terms of spatial variations, urban and suburban areas normally suffer from heavier VOC pollution, relative to rural areas (Li et al., 2015a; Shao et al., 2016; Wang et al., 2016; Zhu et al., 2016; An et al., 2017; Zhang et al., 2017; Zhu et al., 2018; He et



al., 2019). Mozaffar et al. (2020) reviewed the VOC characteristics in the 35 worldwide cities, and illustrated that the VOC concentrations were generally higher at urban and suburban sites than those at rural sites. Moreover, comparing with the VOC pollution in the international areas including Tokyo (Hoshi et al., 2008), London (Schneidemesser et al., 2010), Los Angeles (Warneke et al., 2012), India (Kumar et al., 2018) and Houston (Sadeghi et al., 2021), Chinese cities are suffering

from heavier VOC pollution. Although the VOC concentrations of China have decreased in the recent years along with the effective control strategies (Li et al., 2019b; Mozaffar et al., 2020), China is also an important VOC source compared with other countries on a worldwide scale.

VOCs are emitted primarily from biogenic and anthropogenic processes. The biogenic emissions (mainly from vegetations and trees) are responsible for approximately 1150 Tg carbon of the global VOCs per year. (Guenther et al., 1995; Atkinson

and Arey, 2003; Calfapietra et al., 2009; Song et al., 2021). Chen et al. (2019) pointed out that biological emission dominated the North American VOC budget *via* the high-resolution chemical transport model. Despite the biological contributions, anthropogenic sources stand out to be an important VOCs contributor, especially in the megacities (Schwantes et al., 2016; Ahmad et al., 2017). Anthropogenic emission comprises vehicle exhaust, industrial pollution, painting solvent usage, fuel combustion, liquefied petroleum gas (LPG) usage and coal and biomass burning. The VOC sources varied with sampling

locations, and the vehicle exhaust therein was proved by a plenty of studies to be the predominate VOC sources. For instance, vehicle-related sources explained 27 % of the VOCs in Tianjin, China (Han et al., 2011); 34 % in Zhengzhou, China (Nan et al., 2015); 34 % in central Shanghai, China (Cai et al., 2010a), and 24 % in Hawthorne, USA (Brown et al., 2007). Conversely, Zhang et al. (2018) claimed that industrial emission was a considerable source near an industrial site of Shanghai, China, accounting for more than half of the total VOC sources. In addition, LPG usage was discovered to exceed

vehicle exhaust in influencing the VOC concentrations in Hong Kong (Lam et al., 2013). Although the recent emission inventory reveals the increasing contributions from solvent usage and industrial sources, traffic transportation is still highly responsible for the severe VOC pollution events (Liu et al., 2020, Li et al., 2019a; Song et al., 2021).

VOCs plays an important role in many atmospheric secondary processes, e.g. the formation of SOA and $O_3$. It has been widely reported that atmospheric VOCs has profound impacts on the concentration, physical property and chemical

composition of SOA (Huang et al., 2015; Yuan et al., 2013), which acts as a significant component of $PM_{2.5}$ (Huang et al., 2014; Zou et al., 2017). Firstly, in the presence of $NO_x$, the VOCs undergoes an array of photochemical processes to form secondary VOCs. The produced secondary VOCs further undergoes oxidation to generate multi-function-group species that declines the vapor pressure reduction (Seinfeld et al., 2001; Dechapanya et al., 2003a, b; Sjostedt et al., 2011). The intermediate compounds can undergo heterogeneous absorption on the pPM surface *via* gas-particular partition with a

moderate degree of decomposition and a significantly decreased in the vapor pressure, or through homogeneous reactions to transform into SOA (Pankow, 1994; Seinfeld et al., 2001; Yang et al., 2015; Han et al., 2017). Moreover, the $O_3$ formation potential of VOCs has attracted worldwide concern. Influenced by the complex atmospheric processes, the mechanism of $O_3$ formation can be classified into the $NO_x$-sensitive, transition and VOC-sensitive regimes. The $NO_x$-sensitive regimes of China are mainly distributed at the North, Northeast and Central China, part of Sichuan Basin (SCB) and mountainous area



with strong emissions of industrial, vehicular, power and biogenic (Wang et al., 2019a). In North, Central and East China, the formation of $O_3$ is dependent of the $NO_x$ transition regime. By contrast, VOC-sensitive regime does not have the regional distribution as broad as the $NO_x$ control area. The VOC-sensitive regimes are mainly located at the megacities in the Yangtze River delta (YRD), Pearl River delta (PRD) and Beijing-Tianjin-Hebei (BTH) regions (Zhang et al., 2007; Xue et al., 2014; Wang et al., 2019b). The VOCs transition area distributed at the urban area is mostly near the VOCs control regime. In detail,

atmospheric VOCs undergoes degradation to produce oxidants ($HO_2$ and $RO_2$), which further oxidize atmospheric NO, followed by the production of $NO_2$ and finally the formation of $O_3$ through varied photochemical reactions (Wang et al., 2017). Therefore, it is highly desirable to discuss the relationship of VOCs together with $PM_{2.5}$ and $O_3$. The relevant results would provide insight into the question of how VOCs are likely to response to the pollution of $PM_{2.5}$ and $O_3$.

         The atmospheric VOC characteristics have been widely discussed by single-site measurement in the worldwide cities. The

different research results can be explained by the varied emission sources and meteorological situations. However, limited knowledge is available on the multi-site research at a city level, thus making the influences of different land use types on the pollution VOC characteristics remain unclear. The previous multi-site measurements were performed at a regional scale including Pearl River Delta region, China (Tang et al., 2008; Zhang et al., 2013); Xi'an, China (Song et al., 2021); VIR region, Chile (Toro et al., 2015); Delhi, India (Kumar et al., 2018), and the researchers found that the VOC concentrations in

urban and suburban areas were 2-6 times higher than those at the rural area. Moreover, Li et al. (2019b) characterized the VOCs in Zhengzhou, China and emphasized that the VOC concentrations were slightly high at the industrial site, and the VOCs in Zhengzhou were lower than those in other Chinese cities. Therefore, simultaneous multiple-site measurements within the city itself can provide a comprehensive understanding of VOC characteristics for policymakers to develop effective VOC control strategies.

As the junction of transportation and center of industry, finance, economy and technology of East China, Shanghai covers an area of 6340 $km^2$ and the population over 24 million. The expanding urbanization and industrialization make the VOC pollutions in Shanghai more serious than ever before. The reported VOC pollution characteristics of Shanghai varied with the different sampling site. For instance, the VOC concentrations varied approximately from 20 to 40 ppb at the urban area (Geng et al., 2008; Cai et al., 2010b; Huang et al., 2015; Dai et al., 2017; Li et al., 2019c; Liu et al., 2019; Xu et al., 2019;

Ren et al., 2020; Wang et al., 2020; Liu et al., 2021), and ~90 ppb at the industrial area (Zhang et al., 2018; Wang et al., 2021). However, up to now, the simultaneous observations of VOC pollution characteristics via concurrent multiple-site measurements at Shanghai remains poorly understood, thus hindering the accurate atmospheric pollution controls. Cai et al. (2010b) investigated the VOC pollution in Shanghai, China and highlighted that VOC concentrations in urban administrative area (Pudong) was ~1.5 times lower than those at the industrial area (Baoshan), and slightly lower than those at the busy

commercial area (Xujiahui). This study provided information for understanding the characteristics of VOCs in Shanghai, while the specific discussion about sources and secondary formation potentials of VOCs remained unclear. Moreover, the research performed ten years ago might not reflect how land use type in the present influences in Shanghai. Therefore, it is quite necessary to perform concurrent multiple-site and high time resolution measurements of the VOCs for their



characteristics, sources and secondary formation potentials in Shanghai.

Hereby, in order to distinguish the atmospheric VOCs influenced by the different land use types, we performed the concurrent online VOC measurements at three supersites. The objectives of this study are to (1) figure out the VOC characteristics influenced by the different land use types of Shanghai, (2) identify the predominate sources related to the VOCs pollution, and (3) illustrate the roles of VOCs in the formation of $PM_{2.5}$ and $O_3$. The findings could advance our knowledge in the atmospheric characteristics of VOCs, as well as the relevant control methods designed for the megacity.

**2. Measurements and methods**

**2.1 Sampling site description**

The three sampling sites are located at the different districts of Shanghai: Jinshan district (JS site, 121.18 °N, 30.44 °E), Pudong district (PD site, 121.54 °N, 31.23 °E) and Qingpu district (QP site, 120.98 °N, 31.09 °E) (Fig. S1). The online instruments were 30 m above the ground level to avoid airflow obstruction.

The JS site locates in the Second Jinshan Industrial Area of Shanghai and could be regarded as an industrial site. The site locates in approximately 20 km southwest of the Shanghai Chemical Industrial Park and Garbage Disposal Incinerator-2 (Fengxian District). Moreover, a large number of chemical factories, such as rubber factories, paint solvent factories and oil refineries locate within 2 km of the JS site. In particular, Shanghai Petrochemical Industrial Limited Company (one of the largest refining-chemical integrated petrochemical companies in Shanghai) is 1.5 km southeast of the JS site. Additionally,

Weihong Road and Jinshan Road (the arterial road of Jinshan district) locate 100 and 300 m away from the JS site, respectively. Thus, both industrial processes and vehicle exhaust are plausible to influence the VOC emissions at the JS site.

The PD site locates in the Pudong New Area with the population up to 5.6 million (the Seventh Census of China), which is the most populous district of Shanghai, and 59 km northeast of the JS site. The BaoSteel Group Corporation, International Cruise Port wharf, Waigaoqiao Shipbuilding LTD and Domestic Waste Incinerator locate within 30 km of the PD site. The

Yangshan Harbor (the biggest harbor of Shanghai) and Garbage Disposal Incinerator-1 (Pudong District) locate in 51 and 37 km southeast of the PD site respectively. Furthermore, the PD site is surrounded by three main airports of Shanghai, including Hongqiao International Airport, Longhua Airport and Pudong International Airport. In addition, the site is a junction of roads, commercial and financial areas with intense human activities, which is ~1.5 km away from Jinxiu Road, Jincai High School, Jinnan High School, Jincai Experimental Primary School. Therefore, the PD site could be largely

influenced by the diverse anthropogenic sources, particularly the vehicle exhaust.

The QP site locates near the southeast of Dianshan Lake in the Qingpu district, and is normally taken as a city background site, which is located at 50 km northwest of the JS site and 54 km southwest of the PD site, respectively. Surrounded by vegetation and plants, the QP site situates in approximately 7.6 and 4.6 km northeast of the Grand View Garden and Chenguang Garden, respectively. Moreover, the site is 3.6 km northeast of the Qingxi Country Park and 7.5 km southwest of

the Zhuxi Garden. Besides, the QP site is close to Hu-Yu Highway, with a distance of approximately 1.0 km. Hence, VOC





concentrations at the QP site could be influenced by biogenic sources to some extents relative to JS and PD sites. Overall, considering the intensive traffic transportation near the sampling sites, vehicle emissions could act as an important VOC source, while the distinct surrounding environments may also lead to the varied local sources.

## 2.2 Sampling and analysis

The concentrations of VOCs, CO, $NO$-$NO_2$-$NO_x$, $PM_{2.5}$, and meteorological factors including temperature, wind speed and relative humidity (RH) were measured in one-hour resolution. The sampling campaign was performed simultaneously at the three sites from January 1 to March 31 2019.

At the JS site, the VOCs was collected and analyzed by an online gas chromatography (GC866, Chromato, France) equipped with flame ionization detector (FID). In brief, after the removal of water, the samples were separated for low-carbon ($C_2$-$C_6$)

and high-carbon ($C_6$-$C_{12}$) compounds at the temperature of -5℃ and 25℃, respectively. Then the gas was analyzed by FID after high temperature desorption (380℃) and column chromatographic separation. At the PD site, VOCs was measured by gas chromatography (GC580-FID, PE, USA) and TD300. The samples were separated at -30℃, after water was removed. Then the gas was determined by FID, after high temperature desorption (325℃) and column chromatographic separation. At the QP site, VOCs were determined by gas chromatography (GC5000 BTX/VOC, AMA, German) and a flame ionization

detector (FID). The samples were condensed low-carbon ($C_2$-$C_6$) compounds and high-carbon ($C_6$-$C_{12}$) compounds under the temperatures of 15℃ and 30℃, respectively. Then the gas was analyzed by FID after high temperature desorption (230℃) and column chromatographic separation. The $R^2$ of all of the VOCs were ≥ 0.995. The accuracy of 95 % of compounds was ≤ ± 20 %. Totally 43 species of VOCs were observed, including 16 alkanes, 11 alkenes, 16 aromatics and 1 alkyne. The minimum detection limit (MDL) of most VOC components was ≤ 0.15 ppb (Tab. S1).

The $O_3$, $NO$-$NO_2$-$NO_x$ were characterized by trace instruments (49i ozone analyzer and 42i nitrogen oxide analyzer, produced by Thermo Environmental Instruments Inc., USA). The detection limits are 0.50 and 0.40 ppb, respectively. $PM_{2.5}$ was monitored by a TEOM 1405-F. The meteorological variables including temperature, RH and wind speed were simultaneously acquired from a weather station about 10 km northwest of the SAES site.

## 2.3 Spatial analysis

The spatial heterogeneity of VOCs concentration at each site was determined by the coefficient of divergence (COD) (Wongphatarakul et al., 1998; Sawvel et al., 2015). The COD was calculated by the following Eq. (1):

$$COD_{jk} = \sqrt{\frac{1}{p} \sum_{i=1}^{p} \left(\frac{x_{ij} - x_{ik}}{x_{ij} + x_{ik}}\right)^2} \qquad (1)$$

where $x_{ij}$ presents the mass concentration in $i$ time at the $j$ site, $j$ and $k$ are two datasets, $p$ presents the number of observations. The values of COD represented the degree of similarity of the two datasets, i.e., the value of COD approached 1, illustrating

that the big difference exist in the two datasets (Song et al., 2017).



## 2.4 Positive matrix factorization (PMF) model

PMF was a receptor model designed for the source identification of atmospheric pollutants (Li et al., 2015b; Hopke et al., 2016; Pallavi et al., 2019). In brief, it measured the contributions and sources of pollutant by the least squares, and was based on the mass balance instead of the spectrum of source component (Yang et al., 2013; Hui et al., 2018). In the PMF 5.0, the mass balance equation was calculated by the following Eq. (2):

$$x_{ij} = \sum_{k=1}^{p} g_{ik} f_{kj} + e_{ij} \tag{2}$$

where $x_{ij}$ is the $j$th species concentration in the sample $i$, $g_{ik}$ represents the concentration of the $k$th source to the sample $i$, $f_{kj}$ is the mass contribution of the $j$th species in the $k$th factor, $e_{ij}$ is residue result of $g_{ik}$ and $f_{kj}$, $p$ is the number of independent sources (Paatero, 1997). The function $Q$ was an important factor of PMF (Brown et al., 2015), and calculated by iterative minimization algorithm (Hui et al., 2018). The objective function Q was shown in the Eq. (3):

$$Q = \sum_{i=1}^{n} \sum_{j=1}^{m} \left[ \frac{x_{ij} - \sum_{k=1}^{p} g_{ik} f_{kj}}{u_{ij}} \right] \tag{3}$$

where $u_{ij}$ is the uncertainty of $j$th species in the sample $i$.

The uncertainty of sampling was calculated by the Eq. (4).

$$Unc = \sqrt{(EF \times conc)^2 + (0.5 \times MDL)^2} \quad (conc > MDL) \tag{4}$$

where $MDL$ is the minimum detection limit, $EF$ is the error fraction and can be set to 0.05-0.2 (Song et al., 2007). In this study, four to eleven factors were utilized to determine the option solution. According to the results, seven factors were regarded as the greatest solution.

## 2.5 Potential source contribution function (PSCF) and Cluster

PSCF and Cluster were widely used to observe the back trajectories, source and direction of pollutants (Draxier and Hess, 1998; Hong et al., 2019; Liu et al., 2019), as well as designed to measure the potential VOC source and primary transport pathway of trace elements, which was derived from RTA (Ashbaugh et al., 1985; Xie et al., 2007; Zheng et al., 2018; Liu et al., 2020).

This study was determined the 24-h back trajectories (one hour interval) at the height of 500 m *via* the MeteoInfoMap software. Relevant parameters were acquired from the National Oceanic and Atmospheric Administration. The study area covered by back trajectories was divided into an array of 0.25° × 0.25° grid cell. The higher PSCF indicated that the area was a great contributor to the VOC pollution. The PSCF could be defined as the Eq. (5).

$$PSCF_{ij} = \frac{m_{ij}}{n_{ij}} \tag{5}$$





where the $m_{ij}$ is the number of polluted trajectories through the grid, $n_{ij}$ is all of the trajectories through the grid. In order to distinguish the value of PSCF and increase the accuracy, the weight function $W_{ij}$ was applied to reveal the uncertainty of small values of $n_{ij}$ (Polissar et al., 1999). The $W_{ij}$ could be calculated using the Eq. (6) as follows:

$$W_{ij} = \begin{cases} 1.00 & 80 < n_{ij} \\ 0.70 & 20 < n_{ij} \leq 80 \\ 0.42 & 10 < n_{ij} \leq 20 \\ 0.05 & n_{ij} \leq 10 \end{cases} \quad (6)$$

**2.6 Secondary organic aerosol formation potential (SOAFP)**

As discussed by Grosjean and Seinfeld (1989), SOAFP could be used to evaluate quantitatively the VOC influences on the secondary aerosol formation based on a variety of assumed interactions between VOCs and OH under the sun-light irradiation (8:00-17:00).

$$SOAFP_i = [VOC_i] \times FAC_i \quad (7)$$

where $VOC_i$ is the concentration of the $i$ VOC species, $FAC_i$ is fraction aerosol coefficient of the $i$ compound. The $FAC_i$ was obtained from the previous studies (Grosjean and Seinfeld, 1989; Zhu et al., 2017; Mozaffar et al., 2020).

**2.7 Ozone formation potential (OFP)**

The photochemical activity of VOCs was normally assessed by the $O_3$ formation potential (OFP) (Niu et al., 2016). The value of OFP was effected by many factors like meteorological conditions, VOC concentrations and VOC sources. The OFP was calculated by the following Eq. (8):

$$OFP_i = [conc]_i \times MIR_i \quad (8)$$

where $OFP_i$ is the ozone formation potential of the $i$ species, $[conc]_i$ is the concentration of VOC species $i$, $MIR_i$ is the maximum incremental reactions of the $i$ VOC species, as reported by Carter (1994).

**2.8 Sensitivity analysis**

In order to further understand the relationship between VOCs and $O_3$, a gradient model was applied to investigate the sensitivity of $O_3$ pollution to the VOC concentration. The Eq. (9) was below:

$$\eta = \frac{\Delta_{VOCs}}{\Delta_{O_3}} \quad (9)$$

where $\Delta_{VOCs}$ and $\Delta_{O_3}$ is the concentrations of VOCs and $O_3$ in the specific $O_3$ gradients, respectively. The characteristic





structure and reactivity could influence the contribution of VOCs to $O_3$ formation (Cater, 1994). To determine the quantitative relationship between the VOCs and $O_3$, the VOCs-sensitivity coefficient (VOCs-$S_{O3}$) was used as the Eq. (10):

$$VOCs - S_{O_3} = \frac{\Delta_{VOCs} / B_{VOCs}}{\Delta_{O_3} / B_{O_3}} \tag{10}$$

where $B_{VOCs}$ and $B_{O3}$ are the background concentrations of VOCs and $O_3$. The $O_3$ concentrations below 100 μg m$^{-3}$ were
averaged to be the background level. The background concentration of VOCs ($B_{VOCs}$) was the corresponding VOC concentrations. The high value of VOCs-$S_{O3}$ implied that the concentration of VOCs was greatly affected by the variations of $O_3$ concentration. In order to quantify the sensitivity of VOCs to $O_3$, VOCs and $O_3$ were classified into different groups with a $O_3$ concentration interval of 5 μg m$^{-3}$. It could be calculated by the Eqs. (11-14):

$$y = a \cdot x^b \tag{11}$$

$$\ln y = b \cdot \ln(a \cdot x) \tag{12}$$

$$\ln VOCs - S_{O_3} = b \cdot \ln a + b \cdot \ln \frac{\Delta_{O_3}}{B_{O_3}} \tag{13}$$

$$\ln \frac{\Delta VOCs}{BVOCs} = k \cdot \ln \frac{\Delta_{O_3}}{B_{O_3}} + c \tag{14}$$

where $k$ represents the linear coefficient between $\ln(\Delta_{VOCs}/B_{VOCs})$ and $\ln(\Delta_{O3}/B_{VOCs})$, $c$ is the intercept. The method was also applicable to the calculation of VOCs-$S_{PM2.5}$, the background concentration of PM$_{2.5}$ ($B_{PM2.5}$) was determined by that less than
20 μg m$^{-3}$ (Han et al., 2017). More details can be found in Han et al. (2017). This method was appropriate for understanding the sensitivity of VOC pollution level to $O_3$ (or PM$_{2.5}$) at polluted environments.

## 3. Results and discussion

### 3.1 Characteristics of air pollutants

#### 3.1.1 Data overview

The time series of meteorological factors, VOC categories, PM$_{2.5}$ and $O_3$ are shown in Fig. 1. During the observation campaign, 60 VOC species including 32 alkanes, 11 alkenes, 16 aromatics and 1 alkyne were measured. The temperatures were averaged to be 8.69 ± 3.24, 9.02 ± 3.24 and 7.73 ± 2.92℃, and the hourly mean RH were 83.77 ± 11.38, 75.37 ± 13.29 and 71.80 ± 9.28 % at the JS, PD and QP sites, respectively. The wind speed ranged from 0.86 to 3.39, 0.60 to 4.78 and 2.03 to 8.89 m sec$^{-1}$ with the averages of 1.91 ± 0.49, 1.30 ± 0.62 and 4.37 ± 1.47 m sec$^{-1}$ at the JS, PD and QP sites, respectively.
Under the similar temperature and humidity conditions, the QP site presented higher wind speed than those of the JS and PD sites, and the large gap could be used to explain the distinct VOC characteristics, as discussed later. The results of wind speed were higher than those obtained in Shanghai, China (Zhang et al., 2018), Zhengzhou, China (Li et al., 2019b) and



Xi'an, China (Song et al., 2021).

The VOC concentrations ranged from 7.65 to 71.07, 7.87 to 47.52 and 3.79 to 33.36 ppb with the averages of 21.39 ± 12.58, 21.36 ± 8.58 and 11.93 ± 6.33 ppb at the JS, PD and QP sites, respectively. Obviously, the mean VOC concentrations at the QP site were approximately twice lower than those of the JS and PD sites, due to the emission strength and atmospheric diffusion condition. The VOC levels measured were comparable with those measured in London, England (22.1 ppb), while lower than those observed in Houston, USA (33.88 ppb), Los Angeles, USA (41.3 ppb) and Tokyo, Japan (43.3 ppb) (Hoshi et al., 2008; Schneidemesser et al., 2010; Warneke et al., 2012; Sadeghi et al., 2021). Also, the VOC concentrations shown herein was comparable or lower than those of other studies relevant to China. For instance, the studies in Wuhan (22.2 ppb) (Hui et al., 2018), Chongqing (24.0 ppb) (Li et al., 2018) and Hong Kong (12.7 ppb) (Liu et al., 2019) showed that the VOC concentrations were similar with ones in the present study. However, the present result was lower than those reported by the studies in Nanjing (29.6 ppb) (Shao et al., 2016), Zhengzhou (28.8 ppb) (Li et al., 2019b), Heshan (29.3 ppb) (Song et al., 2019a) and Xi'an (33.8 ppb) (Song et al., 2021). Additionally, compared with the relevant measurements performed previously in Shanghai at the same sampling sites, this study presented lower VOC concentrations. In detail, at the JS site, the VOC concentration was approximately 4 times lower than the findings of Zhang et al., (2018) (94.14 ppb). At the PD and QP sites, the results in this study were slightly lower than those obtained by Cai et al. (2010b) (24.3 ppb) and Zhang et al. (2020a) (15.41 ppb). Changes in meteorological conditions and boundary layer height were the important reasons for this phenomenon. For instance, the occurrence of high wind speed, which was usually accompanied with better convection conditions, resulted in the relatively decreasing VOC levels. Furthermore, a variety of control strategies, such as prohibition of fireworks in the open air and strengthening VOC detection standards and control technology, especially in the Shanghai Baosteel Group Corporation, Shanghai Chemical Industrial Park and Shanghai Petrochemical Industrial Limited Company, and the system of "one factory, one strategy", targeted at mitigating VOC emissions were implemented by Shanghai government.

The maximum VOC concentrations appeared on the 23 January at the JS and PD sites and 27 January at the QP site, respectively. According to the Shanghai Municipal Bureau of Statistics (http://tjj.sh.gov.cn), the traffic flow in January was ~10 % higher than that in the following two months. The phenomenon was likely attributed to the Spring Festival Travel rush, i.e., population travel intensively occurred around the Chinese Spring Festival, thereby causing the elevated pollutant emissions. However, the minimum values presented in the 5, 10 February and 18 March at the JS, PD and QP sites, respectively. The result was due to the fact that a large number of organizations surrounding the sampling sites, such as schools, factories, shopping malls and institutes, were closed for celebrating Chinese Spring Festival, thus causing the decreased concentrations of VOCs especially in aromatics (mainly benzene and toluene) at the JS and PD sites in the February. The reduced VOC concentrations coincide to the response of varied atmospheric pollutants to the COVID-19 lockdown that occurs one year later (Pakkattil et al., 2021), suggesting that the human activities might indeed greatly affect the atmospheric pollution characteristics. Compared with the JS and PD sites, the minimum VOC value at the QP site obviously appeared at different time, suggesting that the impact of local source strength on VOC pollution was varied. It is





worth noting that the distinct spatial heterogeneity of VOCs was observed with the highest value of the coefficient of divergence (COD = 0.36), appeared between the JS and QP sites, followed by the PD and QP sites (COD = 0.33), with that between the JS and PD sites (COD = 0.20) being the lowest. Hence, the spatial heterogeneity of VOCs between the JS and

310 PD site was narrow, while the QP site is largely different from other two sites at the pollution level. This phenomenon was because there were similar VOC emission intensity and atmospheric stability at the JS and PD sites, both of which showed obvious discrepancies at the QP site.

During the observation period, the average $PM_{2.5}$ values were $45.57 \pm 27.59$, $48.51 \pm 27.22$ and $40.27 \pm 27.78$ µg m$^{-3}$, and the mean $O_3$ concentrations were averaged to be $73.59 \pm 23.59$, $57.48 \pm 20.49$ and $99.30 \pm 24.00$ ppb at the JS, PD and QP

sites, respectively. The minimum hourly $PM_{2.5}$ levels were observed in 31 March at the JS site, 6 and 10 February at the PD and QP sites with the values of 10.67, 12.33 and 7.44 µg m$^{-3}$, respectively, while the relatively high $O_3$ concentrations and low VOC concentrations were determined. Statistically, VOCs were found to be positively correlated with $PM_{2.5}$, due to the fact that VOCs were a significant precursor of $PM_{2.5}$. It was well documented that the elevated VOC concentrations led to the increasing rate of $PM_{2.5}$ production *via* photochemical oxidation, gas-particle partition and heterogeneous absorption

(Seinfeld et al., 2001; Yang et al., 2015; Han et al., 2017). Moreover, there was homology between the VOCs and $PM_{2.5}$ to a certain degree. For instance, Wu et al. (2019) showed that the predominant source of $PM_{2.5}$ was the traffic exhaust in South Korea. Meanwhile, a multitude of studies showed the traffic-related source was the dominating source of VOCs including Shanghai, China (Liu et al., 2019), Chongqing, China (Li et al., 2018), Beijing, China (Li et al., 2016), Paris, France (Gaimoz et al., 2011) and Hawthorne, USA (Brown et al., 2007). However, the VOC concentrations were negatively

correlated with $O_3$. The termination and titration ($NO + O_3 \rightarrow NO_2 + O_2$) were more efficient and lots of factors, rather than the emission of precursors, impacted on the surface $O_3$. Li et al. (2019b) emphasized that the absolute concentration of precursor was not the only factor during the $O_3$ formation in Zhengzhou, China. Additionally, the ratio of VOCs/$NO_x$ was less than 5.5 at three sampling sites, suggesting that there was a VOC control system i.e., $O_3$ formation was sensitive to VOCs and a higher proportion of OH radical reacted with $NO_2$ to suppress the $O_3$ formation. Therefore, the reduction of

VOCs might be partly attributed by the chemical loss pathway, and counteraction was imposed by uncertain factors during the formation of $O_3$.

### 3.1.2 Chemical compositions

The VOC compositions at the JS, PD and QP sites are presented in Fig. 2. During the sampling period, the most abundant species were alkanes, followed by aromatics, and the contributions of alkenes and alkyne were close at the above sites.

Specially, ethane, propane, iso-butane, n-butane, ethylene, benzene, toluene and ethyne were the most abundant compositions, contributing to > 75 % of TVOCs (total VOCs), suggesting the strong vehicle-related source at three sampling sites.

Among the four major organic classes, the fractions of alkanes were 49, 71 and 61 % at the JS, PD and QP sites, respectively. Analogously, Zhang et al. (2020a) investigated the contribution of VOC categories to the TVOCs in Shanghai, China, and





showed that the largest contribution was alkanes, comprising 59.36 % of the total VOCs, due to their widespread emission

sources and longer atmospheric lifetime. The great contributions of ethane, propane and $C_4$-$C_6$ branches alkanes led to the

highest proportions of alkanes at the three sites, as the findings of Zhang et al. (2020a) and Song et al. (2021), while the

relative abundances of the compounds varied with the sampling sites. The fraction of alkanes at the PD site was obviously

higher than those at the JS and QP sites, which was mainly contributed by the obvious high proportion of propane at the PD

site. This phenomenon was greatly influenced by the vehicle emission, and the PD site was situated at the junction of roads,

metros, commercial and financial areas, which was ~1.5 km away from Jinxiu Road and metro line 6 and ~2 km away from

the Waihuan Highway. Moreover, the contribution of ethane at the JS site was nearly half lower than that at the PD and QP

sites. Ethane is a major compound of natural gas, which is associated with the incomplete combustion (Guo et al., 2011). It

was thus supposed that the vehicle emission exerted a relatively weak influence on VOC emission at the JS site, as compared

to the other two sites.

The second-largest group was aromatics accounting for 21, 11 and 14 % at the JS, PD and QP sites, respectively. As reported

by Cai et al. (2010a), aromatics was found to be abundant species in addition to alkanes, accounting for 24.9 % of the total

VOCs at the Xujiahui site in Shanghai, China, which was slightly higher than the results obtained in this study. Note that the

dominating compounds of aromatics were toluene, m/p-xylene, benzene, ethylbenzene and o-xylene (BTEX), and the result

was in line with that in Shanghai, China (Zhang et al., 2018) and Xi'an, China (Song et al., 2021). The proportion of

aromatics at the JS site was approximately twice greater than those at the PD and QP sites. In particular, the contribution of

toluene at the JS site was markedly increased (~3 times) relative to the other two sites. Similarly, Li et al. (2019b) studied the

VOC concentrations at different sites in Zhengzhou, China, and showed that the fraction of toluene was approximately twice

higher at the JK site (an industrial site) than those at the other sites. The JS and JK sites are close to the industrial area and

have heavy industrial emissions. The industrial activities in paint and printing factory, manufacturing factory and rubber

factory were the potential reason for such phenomenon (Zheng et al., 2010; An et al., 2014; Debevec et al., 2021).

The alkenes accounted for 15, 10 and 11 % at the JS, PD and QP sites, respectively, and the discrepancies of alkenes

contribution to TVOCs among three sites were narrow. The ethene, propylene and butene were predominant compounds in

the specie of alkenes, and their contributions were comparable to the results of Shao et al. (2016). Meanwhile, the alkyne

contributed 15, 8 and 14 % on the TVOCs at the above sites. Compared with the result reported by Liu et al. (2017) in

Hangzhou, China, the contribution of alkyne was comparable to the present study, accounting for 16.6 % of the TVOCs. As

shown in Fig. 2, the fraction of alkyne was twice lower at the PD site than that at the JS and QP sites, which may be caused

by the weak combustion sources including chemical- and bio-fuels burning (Zhu et al., 2016; Li et al., 2019b).

### 3.1.3 Diurnal variations

The diurnal variations of VOCs on the weekdays and weekends at the JS, PD and QP sites are shown in Fig. 3. On the

weekdays, the average VOC concentrations were 21.03 ± 2.66, 22.00 ± 4.88 and 11.09 ± 1.58 ppb at the JS, PD and QP sites,

respectively, whereby the highest peaks were observed during the rush-hour traffic in the morning (8:00-10:00 LT), reaching



26.61, 35.11 and 13.54 ppb. The highest peak was more significant at the PD site than those at the JS and QP sites, illustrating that the vehicular emission accounted for more VOC emissions at the PD site than those at the other two sites. During the rush-hour traffic in the evening (18:00-21:00 LT), the VOC concentrations also intended to increase, and the values were 18.46, 20.82 and 10.22 ppb at the JS, PD and QP sites, respectively. Such results could be attributed to the strong influence of the vehicle exhaust. The VOC concentrations dropped to the minimum at the 16:00 LT with the values of 16.96, 15.62 and 8.77 ppb at the JS, PD and QP sites, respectively. The strong radiation, high temperature and boundary layer resulted in the most intense air convection condition and promoted the photochemical loss and dilution of VOCs, leading to the lowest VOC concentration (Zhang et al., 2018). It is worthwhile to note that the VOC concentrations remained relatively high value in the early morning (around 5:00 LT) at the JS site compared with those at the other two sites. This phenomenon was attributed to the fact that a large number of rubber factories, paint solvent factories and oil factories located within 2 km of the JS site, whereby some factories usually work continuously for 24 hours, leading to the increased VOC concentrations. Compared with the previous studies (Velasco et al., 2007; Zhang et al., 2018; Wang et al., 2021), the diurnal variation of the VOC concentrations in this study did not show apparent bimodal feature at the sampling sites. The industrial processes and biogenic emission surrounding the sampling sites also had great influence on the VOC variations except for the traffic exhaust, which was the potential reason for such scenario. Therefore, it is necessary to analyze the diurnal variation of VOC concentrations at the different land-use types.

On the weekends, the JS site exhibited the highest average VOC concentration of 20.36 ± 2.23 ppb, followed by the PD and QP sites with the concentrations of 19.96 ± 2.37 and 10.96 ± 0.67 ppb, respectively. The VOC concentrations on the weekends were 3.31, 10.19 and 1.19 % lower than those on the weekdays. Similarly, the elevated VOC concentrations on the weekdays were found in other cities including Shanghai, China (Geng et al., 2008), Sacramento (Murphy et al., 2007) and Los Angeles, USA (Nussbaumer and Cohen, 2020), due to the low anthropogenic activities occurring on the weekends, reflecting the low VOC emission (Murphy et al., 2007). It should be noted that there were narrow discrepancies of VOC concentrations at the JS site between the weekdays and weekends except for the rush hour in the morning, which was attributed to the influence of the continuous operation of industrial processes. At the PD site, the highest value during the morning peak on the weekends (26.03 ppb) was obviously lower than that on the weekdays (~34.88 % decreased). Such scenario resulted from the enhancement of anthropogenic VOC emissions intensity on the weekdays, suggesting that there were anthropogenic-dependent VOC emissions at the PD site. At the QP site, the elevated VOC concentrations were presented during the 18:00-20:00 CST on the weekends relative to the weekdays. The phenomenon was associated with the fact that a large number of people had a trip around the QP site on the weekends, leading to the VOC emissions increased. Furthermore, the VOC concentration gradually increased during the rush-hour traffic in the morning and evening both on weekends and weekdays. Additionally, the minimum VOC concentrations occurred around the same time, which appeared on the 16:00 CST at the JS and PD sites and 15:00 CST at the QP site on the weekends, respectively, with the values of 17.14, 17.56 and 10.33 ppb. Such phenomenon could be explained by the meteorological factors, and would be expounded in detail in the last section.





### 3.1.4 Clean-haze discrepancy

Referring to the previous documents (Li et al., 2017; Hui et al., 2019), haze pollution was defined as the condition with visibility < 10 km and RH > 80 %. During the observation period, there were three sequential haze pollution events at the three sites on the basis of this criteria, i.e., 18 to 25 January, 23 February to 3 March and 21 to 26 March. Note that the haze break was usually accompanied with the increasing VOC concentrations. The VOC concentrations between the clean and polluted days are shown in Fig. 4. The VOC concentrations on the clean days were averaged to be 20.53 ± 12.10, 19.29 ± 7.60 and 11.04 ± 6.67 ppb at the JS, PD and QP sites, respectively. When it comes to haze days, the VOC concentrations increased by 27.15, 32.85 and 22.42 % at the JS, PD and QP sites, respectively. Wu et al. (2016) also emphasized the increased VOC concentrations during the haze days relative to clean days in Beijing, China. Obviously, the 'haze-clean' discrepancy of VOC concentrations was narrow at the QP site compared with the JS and PD sites. Such phenomenon could be attributed to the rapid industrialization and urbanization, resulting in the stagnant weather conditions and high anthropogenic emissions, which further led to the severe haze pollution at the JS and PD sites. In detail, the 'clean-haze' discrepancy was dominated by the aromatics (*m*-ethyltoluene, *p*-ethyltoluene, *m*-ethylbenzene, 1, 2, 4-trimethylbenzene) at the JS and PD sites, as reflected by the 44.54 % and 36.05 % higher concentrations on haze days relative to clean days. The elevated concentrations reflected the concentrated emission sources of industrial production, painting/coating and vehicle exhaust. Zhang et al. (2021) studied the VOC characteristics during the haze pollution in Zhengzhou, China, and showed that aromatics present an upward tendency (~34.04 % uplift) on the haze days, which was slightly lower than the results shown herein. at the QP site, alkanes (2, 2, 4-trimethylpentane, *n*-hexane, *n*-heptane) presented significant 'clean-haze' discrepancy (~36.58 % uplift), implying the great influences of vehicle exhaust and fuel evaporation. By contrast, Hui et al. (2018) observed the elevated concentration of alkanes during the haze days in Wuhan, China, and found that there were 37.28 % increment of alkanes, which was comparable to the result in the present study. Thus, it could be deduced that, the haze occurrence is mainly associated with vehicle exhaust, industrial production and painting/coating at the JS and PD sites, whereas the vehicle exhaust and fuel evaporation at the QP site. The result agreed well with the studies about the haze pollution in Shanghai, China (Li et al., 2019b; Wei et al., 2019). Accordingly, VOC species could probably serve as the source indicators when exploring the haze pollution.

### 3.2 Source apportionment

#### 3.2.1 Special VOC ratio analysis

Different VOC species have different sources, and accordingly the ratio of different species could be used to preliminary distinguish the emission sources (An et al., 2014). Herein, the characteristics of toluene/benzene (T/B), *iso*-pentane/*n*-pentane (P/P) and *m, p*-xylene/ethylbenzene (X/E) varying the sampling sites to identify the source types. The results are shown in Fig. 5 and Fig. S2-4.

It was well documented that the varied VOC sources could be identified by the different T/B ratios. The ratio in the range of





0.2-0.4 indicates different combustion processes (Li et al., 2011; Mo et al., 2016). The ratio in the range of 0.9-2.2 indicates

the vehicle sources varying with fuel types (Dai et al., 2013; Zhang et al., 2013; Yao et al., 2015; Deng et al., 2018). The T/B ratio about vehicle emission exhibited approximately 1.5, which approaches the results of the tunnel experiments, due to the different vehicle emissions including diesel and gasoline vehicle emissions (Deng et al., 2018; Song et al., 2021). The ratio in the scope of 1.4-5.8 reflects the industrialization impact (Mo et al., 2015; Shi et al., 2015). In addition, the ratio greater than 8.8 implies the effect of paint solvent usage (Yuan et al., 2010; Zheng et al., 2013). The mean T/B ratios were 4.59 ± 4.3 (r =

0.41) and 1.61 ± 0.79 (r = 0.65), respectively at the JS and PD sites, with most of the ratios (68.89 and 84.15 %) distributing the reference range of vehicle emissions and industrial emissions, suggesting that both sources exerted a significant influence on VOC concentrations (Fig. 5a-b). It is worth noting that the range of T/B ratios at the JS site (0.44-22.86) were wider than that at the PD site (0.48-4.38), suggesting that the VOC concentrations at the JS site were influenced by diverse emission factors. The average ratio at the QP site was 1.01 ± 0.66 (r = 0.70) with 43.02 % of T/B ratios distributing the range of

vehicle emissions and burning emissions, suggesting that both of the sources contributed significantly to VOC pollution (Fig. 5c). The temporal variations of T/B ratio and TVOC concentrations are shown in Fig. S2. The T/B ratios were mainly located in the scope of 0.9-2.2 and 1.4-5.8 at the JS and PD sites during the high VOC concentrations, respectively; while the specific values at the QP site were only concentrated in the scope of 0.9-2.2. Such phenomenon showed that vehicle exhaust was a significant contributor to the VOC pollution at the three sampling sites, and VOC concentrations were less influenced

by the industrial emission at the QP site relative to the JS and PD sites, due to the fact that the QP site was normally taken as a city background site and far from the industrial park.

The fossil-fuel-derived sources (vehicle exhaust, fuel evaporation and coal combustion) could be further distinguished by the P/P ratio. Iso-pentane and n-pentane have similar atmospheric lifetimes, and therefore are significantly correlated, as evidenced by the linear relation coefficients (r) were 0.91, 0.91 and 0.82 (p < 0.01) at the JS, PD and QP sites, respectively.

The phenomenon supported the fact that the two compounds varied with a similar trend and had constant emission sources (Jobson et al., 1998; Yan et al., 2017). Lower P/P ratios (0.56-0.80) are often identified for coal combustion (Li et al., 2019a), and it was well known that the P/P ratios in the range of 2.2-3.8 are characterized by vehicle emission (Liu et al., 2008; Wang et al., 2013). Figure 5d shows that the linear correlation coefficients between the iso-pentane and n-pentane were 0.96 ± 0.30 (r = 0.91), 1.36 ± 0.22 (r = 0.91) and 2.46 ± 1.49 (r = 0.81) at the JS, PD and QP sites, respectively. The result at the QP site

was comparable to that measured at a Pearl River tunnel (2.93), suggesting that the vehicle emission is an outstanding VOC source (Liu et al., 2008). The lower ratios at the JS and PD sites highlighted the mixed emission sources with significant coal combustion included. The temporal variations of P/P ratio and TVOC concentrations are shown in Fig. S3. The P/P ratios were approaching 2.2-3.8 at the sampling sites during high VOC values, indicating the great impact of vehicle emission on VOC pollution, in accordance with the findings of Song et al. (2021).

Besides the local emissions influencing the VOC pollution characteristics, regional transport has been probed as a potential VOC source. Herein, the ratio of X/E was used to evaluate the transport impacts. *M, p*-xylene and ethylbenzene are found to be similar in emission sources, while the former exhibits ~3 times greater reactivity toward OH radical than the latter



(Nelson and Quigley, 1983; Chang et al., 2006; Vardoulakis et al., 2011). Hence, lower X/E ratios normally suggested more significant air mass aging, that is, more influences from external transport. The X/E ratios were averaged to be 2.33 ± 0.37 (r = 0.98), 2.18 ± 0.42 (r = 0.97) and 2.03 ± 1.52 (r = 0.74) at the JS, PD and QP sites, respectively (Fig. 5e). The results showed that the X/E ratios were similar at the sampling sites while the values were slightly lower at the QP site relative to the JS and PD sites, indicating that the aged VOCs contributed by the polluted air masses *via* the regional transport toward the QP site. The temporal variations of the X/E ratio and TVOC concentrations are shown in Fig. S4. During the VOC pollution, the X/E ratios were approximately 2.3, 2.5 and 1.8 at the JS, PD and QP sites, respectively. Obviously, the X/E ratio was lower at the QP site than those at the JS and PD sites. In particular, the high VOC concentration (27.17 ppb) was observed in 7 February at the QP site, corresponding to the minimum X/E ratio (0.27), while the X/E ratios were 3.07 and 1.95 at the JS and PD sites, corresponding to the VOC concentrations of 12.45 and 22.45 ppb. Such results illustrated that the influence of external transport during the high VOC concentrations was greater at the QP site compared with the JS and PD sites.

### 3.2.2 The PMF analysis

The PMF analysis can quantitatively determine the VOC source contributions, compared with the specific VOC ratio analysis (Hui et al., 2019). In this study, 39 VOC species were put into the PMF model, followed by the output of seven resolved factors including vehicle exhaust, industrial source, LPG usage, paint solvent, fuel evaporation, coal combustion and biogenic source. The factor profiles and contributions to VOCs at the JS, PD and QP sites are shown in Fig. 6.

Vehicle exhaust was characterized by the high proportions of alkanes, some alkenes and certain percentage of aromatics (benzene, xylene and trimethylbenzene) (Liu et al., 2008; Cai et al., 2010; Ling et al., 2011; An et al., 2017; Hui et al., 2018). In this study, the high contributions of the ethane (69.59 and 48.01 %), *iso*-butane (35.14 and 26.12 %) and some alkenes especially the propylene (34.72 and 19.74 %) and *trans*-2-butene (50.49 and 17.40 %) at the JS and PD sites, as well as the ethane (42.41 %), ethylene (49.14 %) and ethyne (45.94 %) at the QP site were observed, and all of these compounds are widely regarded as vehicular emission tracers (Cai et al., 2010a; Ling et al., 2011; An et al., 2017; Hui et al., 2018). The vehicle contributions were 23.18, 33.37 and 32.12 % at the JS, PD and QP sites, respectively, indicating that vehicle exhaust was the predominant source of VOCs, as the results of T/B and P/P ratios mentioned above. Shanghai owns the motor vehicles more than 4.4 million at 2019, and vehicle exhausts had been proved to be one of the main causes of the local air pollution in Shanghai (Huang et al., 2015; Dai et al., 2017; Liu et al., 2019; Cai et al., 2010b).

Industrial emission was featured by high percentage of aromatics (benzene, toluene and trimethylbenzene) and certain percentage of alkanes and alkenes (Guo et al., 2011; Dumanoglu et al., 2014; Sun et al., 2016). At the JS and PD sites, the high contributions of *n*-nonane (17.84 and 33.99 %), benzene (13.89 and 39.16 %), toluene (21.11 and 19.54 %), trimethylbenzene (94.18 and 84.05 %), hexane (14.81 and 38.96 %) and ethylene (14.99 and 36.91 %) were observed. It was widely acknowledged that asphalt could release *n*-nonane, which was mainly used in industrialization and transportation (Brown et al., 2007; Liu et al., 2008). Moreover, benzene and toluene are primarily applied for industrial solvent production,





and trimethylbenzene is frequently used in manufacturing (Morrow, 1990; Ling et al., 2011). Therefore, the recognized factor can be identified as industrial source. Industrial events were calculated to contribute 22.39 % of the VOCs at the JS site, as well as 18.81 % at the PD site, indicating the considerable contributions of industrialization next to the aforementioned traffic factors. The QP site is far away from the industrial area, and therefore no industrial factors were disclosed at this site.

Coal combustion factor was characterized by propane, ethylene, ethyne, and benzene (Liu et al., 2008; Ling et al., 2011; Song et al., 2018). Moreover, the relatively low proportions of propene, *n*-hexane, *n*-heptane and toluene should be considered as well due to their detectable levels in the coal combustion processes (Hui et al., 2018). The PMF analysis illustrated that coal combustion was the third-largest source of VOCs at the three sites, due, in part, to a variety of combustion methods. Coal combustion is responsible for 14.95 % of the VOC emission at the PD site, as well as 13.85 % at

the QP site. However, the proportion reached up to 21.48 % at the JS site, reflecting the frequent fossil usage in the industrial area of Shanghai.

LPG usage was determined by some alkanes (propane, n-butane and iso-butane) and alkenes (ethylene, propylene and butene). At the JS and PD sites, high proportions of propane (21.24 and 14.41 %), *n*-butane (15.53 and 13.79 %), *iso*-butane (12.61 and 11.40 %), ethylene (31.18 and 10.08 %) and propylene (34.94 and 31.38 %) were observed, as well as the

*iso*-butane (43.88 %), propylene (25.08 %) and *cis/trans*-2-butene (53.10% and 57.65 %) at the QP site, highlighting the LPG-related sources (Yang et al., 2013; Lyu et al., 2016). Herein, LPG usage explains 13.17, 14.76 and 12.14 % of the VOCs monitored at the JS, PD and QP sites, respectively. At the end of 2015, the amount of LPG users about household and catering achieved 3.3-6.5 million in Shanghai (Hui et al., 2018; Zhang et al., 2018a). Overall, LPG usage was one of the important VOC sources, and its VOC contributions accounted for the half of the vehicle emissions.

Fuel evaporation can be identified by $C_3$-$C_7$ alkanes, especially *n*-pentane and *iso*-pentane (Zheng et al., 2020), as well as the $C_3$-$C_5$ alkenes such as *trans/cis*-2-butene (Geng et al., 2009; Hui et al., 2018; Zhang et al., 2018a; Zheng et al., 2020). There were high percentages of *n*-pentane (37.94, 25.20 and 71.95 %), *iso*-pentane (41.45, 28.18 and 23.16 %) and butene (26.73, 70.83 and 22.52 %) at the JS, PD and QP sites, respectively. Some literature showed that alkanes like *n*-pentane and *iso*-pentane were gasoline tracers, and some low-carbon alkanes could evaporate from the unburned fuels (Guo et al., 2004;

Wang et al., 2013). The VOC contributions from fuel evaporation were calculated to be 4.62, 10.35 and 20.15 % at the JS, PD and QP sites, respectively.

Paint solvent usage is normally characterized by $C_6$-$C_8$ alkanes and some aromatics like toluene, ethylbenzene and *o/m/p*-xylene (An et al., 2017; Hui et al., 2018; Song et al., 2019a). Wang et al. (2013) further proposed that these aromatic compositions could be emitted from the usages of painting, adhesives, and coating activities. Herein, emission factors that

meet the characteristics could be explained by paint solvent usage, which accounted for 15.15, 7.77 and 10.36 %, respectively at the JS, PD and QP sites.

Biogenic source was distinguished by a high contribution of isoprene (Liu et al., 2008; Wu et al., 2016). Factors that coincide to the specific characteristic were regarded as biogenic source in this study. Our analysis indicated that biogenic source could explain 11.39 % of the VOCs at the QP site, reflecting the vegetation surrounding this site. However, this emission factor





could not be reproduced when it comes to the analysis of the JS and PD sites, implying the limited influences of biogenic sources in the population- and industrialization-concentrated areas that were suffered from the anthropogenic emissions. Generally, the contribution from vehicle exhaust at the sampling sites accounted for nearly one third of total VOC sources. Obviously, the vehicle contribution at the JS site was lower than those at the PD and QP sites, which could be attributed to the heavier emissions of industrial production and coal combustion at the JS site. Moreover, vehicular emission was also a

dominating source of VOCs in other cities, and the contributions were higher than 20 % except in Calgary, Canada (17.2 %) (Fig. 7). The effective and continuous control strategies about vehicle exhaust were still of priority to alleviate VOC pollution in Shanghai. The fraction of industrial source was the second-largest source at the JS and PD sites, which were relatively high compared with the other studies, except for Xi'an, China (29.7 %) and Paris, France (35 %). This was partially due to the fact that the JS site located in Second Jinshan Industrial Area of Shanghai and could be regarded as an

industrial site, and Waigaoqiao Ship building LTD and BaoSteel Group Corporation located within 30 km of the PD site, illustrating the great influence of industrial processes on VOC pollution at the JS and PD site. Combustion was the third-largest source at the three sites, and its contribution varied largely in different studies, from 4 % (Song et al., 2019b) to 46 % (Bari and Kindzierski, 2018), which was ascribed to the different definition of the combustion among various studies. For instance, Song et al. (2019b) only regarded the wood combustion as the combustion source in the study of Seoul, South

Korea, while the sum of fuel combustion and oil/natural gas extraction/combustion were referred to the combustion source in the study of Calgary, Canada (Bari and Kindzierski, 2018).

The contribution of fuel evaporation varied by site locations, i.e., at the JS site (4.62 %) was lower than that in the previous studies (6.28-20 %), while the value at the PD site (10.35 %) was relatively average (Li et al., 2018; Hui et al., 2018; Liu et al., 2019; Song et al., 2019a, b; Song et al., 2021). At the QP site, the result (20.15 %) was comparable with that in the other

studies (Hui et al., 2018; Li et al., 2018; Song et al., 2019a, b; Song et al., 2021). The proportions of LPG usage were approximately twice higher at the sampling sites than those reported by the other studies (Hui et al., 2018; Song et al., 2019a). The fractions of paint solvent usage shown herein were lower than the average of other cities (8.2-41 %) (Gaimoz et al., 2011; Bari and Kindzierski, 2018; Li et al., 2018; Liu et al., 2019; Hui et al., 2018; Song et al., 2019a, b; Liu et al., 2021; Song et al., 2021). The contribution of biogenic source at the QP site was relatively average, compared to the estimates from

other studies (4.9-15 %) (Cai et al., 2010b; Gaimoz et al., 2011; Bari and Kindzierski, 2018; Hui et al., 2018; Li et al., 2018; Song et al., 2019a, b). Obviously, the results of source contribution estimates varied with different studies (Lyu et al., 2016; Li et al., 2020). Additionally, the VOC sources also showed different at a regional scale within the city (Tang et al., 2008; Li et al., 2019b; Cai et al., 2010; Song et al., 2021). Such phenomenon was partially due to the different factors during the observation period, inclusion of VOC species, emission strength and/or land-use types (Li et al., 2019b; Cai et al., 2010; Li

et al., 2020; Song et al., 2021). Thus, the source apportionment for the different land-used types at a regional scale among various studies is necessary to identify the high-quality localized source.

### 3.3 Back trajectories and PSCF results





Regional transport could also contribute to VOC pollution, besides the direct influence of local source (Hui et al., 2018). Figure 8 exhibited the 24-h backward trajectories, PSCF results and the VOC proportion of each trajectory during the entire

observation at the JS, PD and QP sites. There were 6 clusters (northern trajectories: Cluster 1 + 2; eastern trajectories: Cluster 3; southern trajectories: Cluster 4; southwestern trajectories: Cluster 5; northwestern trajectories: Cluster 6) at the JS site, as shown in Fig. 8a. One can see that the contributions of northern trajectories to both the total trajectories and the total pollution trajectories were significantly higher than other four cluster trajectories, comprising 45.74 and 46.46 %, respectively (Tab. 1). Identified as the north direction cluster, the north long-distance trajectory (Cluster 1) and short-distance

trajectory (Cluster 2), respectively, accounting for 16.67 and 29.07 % of the VOC transportation, indicating that the VOC pollution was mainly impacted by the northern trajectories from the junction of the Bohai Sea, Shandong Province, the East China Sea and Jiangsu Province in addition to local emissions. The contributions of VOCs by each air mass trajectories indicated that alkanes tended to dominate the transported VOC community at the JS site due to its weak reactivity in the long-range transport.

One can see that air mass trajectories could be cluster into northern trajectories (Cluster 1 + 2), northeastern trajectory (Cluster 3), southeastern trajectory (Cluster 4), southwestern trajectory (Cluster 5) and northwestern trajectory (Cluster 6) at the PD site, as shown in Fig. 8b. It is evident that the contributions of northern (Cluster 1) and northeastern (Cluster 3) trajectories to the total trajectories both the total trajectories and the total pollution trajectories were great higher than other four cluster trajectories, accounting for 52.23 and 54.72 %, respectively. The results suggested that VOCs was significantly

influenced by the pollution transportation from East China Sea, Zhejiang Province and Jiangsu Province junction, as well as local sources. The proportions of VOCs by each air mass trajectories indicated that alkanes and alkenes contributed to the transported VOC community at the PD site.

At the QP site (Fig. 8c), the trajectories were clustered into the northern trajectories (Cluster 1 + 2), northeastern trajectory (Cluster 3), southeastern trajectory (Cluster 4), southwestern trajectory (Cluster 5) and northwestern trajectory (Cluster 6).

The proportion of northeastern trajectory to both the total trajectories and the total pollution trajectories were higher than those of the other trajectories, accounting for 31.87 and 27.81 %, respectively. This result indicated that the concentration of VOCs was mainly impacted by the northeastern trajectory from the junction of East China Sea, Zhejiang Province and Jiangsu Province in addition to local sources. Although the contribution of Cluster 2 was relatively small, the VOC concentration was the highest, reaching $23.63 \pm 10.75$ ppb (Tab. 1). Therefore, attention should be paid to the long-distance

transmission of highly polluted from Beijing, Tianjin and Liaoning Province. The alkanes contributed most of VOCs in air mass trajectories at the QP site, which was in line with the results of the JS site.

The region with high PSCF levels indicates highly potential regional transport sources (Hui et al., 2018). Based on the PSCF results, at the JS and QP sites, high values were observed to the north of Shanghai, while at the PD site, high values were observed to the northeast of Shanghai. All of the three sampling sites presented that the highest PSCF levels appeared around

the central area. Shanghai is the junction of economy, finance, trade and technology, there are a large number of local sources such as vehicle emission, industrial source and fuel evaporation, which contributed most to the local VOC concentration,





indicating that local sources played an important role in the pollution of VOCs. Similarly, Song et al. (2021) studied the PSCF values at three different sites in Xi'an, China, and showed that Xi'an had a strong local source, which was also consistent with the findings in Wuhan, China (Hui et al., 2018; Hui et al, 2019), suggesting that local source was a significant
contributor to the VOC pollution.

### 3.4 Secondary transformation potential

### 3.4.1 SOA formation potential

The close linkages between VOC$_s$ and particle formation induced the quantitative discussion on SOAFP (Chen et al., 2007; Guo et al., 2017; Tan et al., 2018), and the Grosjean's methodology has been widely recommended as the standard reference
(Fig. 9). The SOAFP was calculated only for daytime (8:00-17:00 LT) to avoid the emission from the human activities during nighttime (Grosjean and Seinfeld, 1989; Zhang et al., 2020b). During the observation campaign, the SOAFP was averaged to be 1.00, 0.46 and 0.41 μg m$^{-3}$ at the JS, PD and QP sites, respectively. Such levels were comparable with those obtained in Jinan (Zhang et al., 2017) and Taiwan, China (Vo et al., 2018), while were higher than those of Nanjing, China (Mozaffar et al., 2020) (Tab. S2). The secondary potentials accounted for 2.19, 0.95 and 1.02 % of the PM$_{2.5}$ concentrations
at the above sampling sites, all of which were lower than the results in Nanjing, China (3.46%) (Mozaffar et al., 2020) and Wangdu, China (8.4 and 17.84 % under high-NO$_x$ and low-NO$_x$ conditions) (Zhang et al., 2020b). Although the SOA contribution to PM$_{2.5}$ was low, the SOAFP values in the present study were underestimated. Only 21 VOC species were included in the estimations, and the VOC concentrations in the daytime were only 20.80, 9.43 and 7.30 ppb at the JS, PD and QP sites, respectively. Moreover, the published FAC values were obtained by merely considering the OH-related interactions,
conservative estimations of the SOAFP values should be noted in the relevant research (de Gouw et al., 2011; Hennigan et al., 2011; Spracklen et al., 2011; Zhang et al., 2020b). Additionally, the discrepancies could also be induced by the different atmospheric conditions and study area. In the current study, the aromatics, such as BTEX, were determined to be the main SOA contributors, and the five compounds totally accounted for 86.07, 96.21 and 86.38 % at the JS, PD and QP sites, respectively, in accordance with the findings in Jinan, China (Zhang et al., 2017). Therefore, to achieve the better SOA
reduction effects, the concentration of BTEX should be controlled in priority at the sampling sites, and more efficient strategies should be developed to limit the emissions of vehicle exhaust.

### 3.4.2 The VOCs-PM$_{2.5}$ sensitivity

The close associations between VOCs and SOA could induce the sensitive response of VOC concentrations to the different pollution degree of PM$_{2.5}$. In order to further understand the plausible influences on the atmospheric abundance of PM$_{2.5}$, the
VOCs-PM$_{2.5}$ sensitivity (VOCs-S$_{PM2.5}$) was applied to reduce the analysis error. Atmospheric pollution here can be divided into five levels by the mass concentration of PM$_{2.5}$: clean level (PM$_{2.5}$ < 35 μg m$^{-3}$); slight pollution level (35 < PM$_{2.5}$ < 75 μg m$^{-3}$); medium pollution level (75 < PM$_{2.5}$ < 120 μg m$^{-3}$); heavy pollution level (120 < PM$_{2.5}$ < 180 μg m$^{-3}$); extreme pollution



level (PM$_{2.5}$ > 180 μg m$^{-3}$) (Han et al., 2017). The evolutions of VOCs-S$_{PM2.5}$ as a function of PM$_{2.5}$ concentration at the JS, PD and QP sites are shown in Fig. 10. The low values of VOCs-S basically remain constant, while the high values displayed a descending trend. In the clean level, VOCs-S$_{PM2.5}$ varied greatly from 0.19 to 2.50, 0.26 to 2.17 and 0.11 to 3.26 at the JS, PD and QP sites, respectively. In the pollution level, the scope of VOCs-S$_{PM2.5}$ was narrow, especially in the extreme pollution level with the values fluctuating from 0.10 to 0.16, 0.13 to 0.17 and 0.17 to 0.28 at the JS, PD and QP sites, respectively. It is worth noting that the VOCs-S$_{PM2.5}$ values dispalyed a steady decreasing trend, and the ratios of PM$_{2.5}$ values to $B_{PM2.5}$ was increased in the clean and slight polluted levels. Such results demonstrated that the VOC concentrations were sensitive to that of PM$_{2.5}$ in the above two episodes. In order to quantify the sensitivity of VOCs to PM$_{2.5}$, we classified VOCs and PM$_{2.5}$ into different groups with a PM$_{2.5}$ concentration interval of 5 μg m$^{-3}$, on the basis of the reported by Han et al.(2017). The mean VOCs-S$_{PM2.5}$ values followed an exponential function, and the values of $k$ were determined to be 0.39, 0.46 and 0.56 at the JS, PD and QP sites, respectively, according to the Eqs. (11-14). The VOCs at the QP site could be more sensitive to PM$_{2.5}$ relative to other two sites because greater k normally indicated the rapider increment of VOC concentrations. The four groups of VOCs displayed similar linkages with VOCs, and the higher values of $k$ was attributed to the aromatics at the JS and PD sites, while the alkanes at the QP site (Fig. S6). Thus, the optimal choices of controlling VOC species presented difference at the diverse sampling sites. Accordingly, in order to efficiently control the VOC-induced haze pollution, the concentrations of aromatics at the JS and PD sites, as well as alkanes at the QP site should be controlled in priority, as the results of "clean-haze" discrepancy.

## 3.5 Photochemical reactivity of VOCs

### 3.5.1 Ozone formation potential

The atmospheric photochemical reactivity of VOCs makes them widely considered as the important precursors of O$_3$ (Alghamdi et al., 2014; Kumar et al., 2018). Hereby, the OFPs were calculated to be 50.89, 33.94 and 24.26 ppb at the JS, PD and QP sites, respectively (Fig. 11). Such secondary potentials accounted for 69.15, 59.05 and 24.43 % of the measured O$_3$ concentrations in the above sampling sites. Obviously, the contributions of VOCs to PM$_{2.5}$ (2.19, 0.95 and 1.02 % at the JS, PD and QP sites) were lower than those to O$_3$ at the sampling sites. The VOCs-related O$_3$ contributions were lower than those in Delhi, India (Kumar et al., 2018), and Chinese cities like Shanghai, China (Zhang et al., 2018a); Wuhan, China (Hui et al., 2018); Taiwan, China (Vo et al., 2018); Nanjing, China (Mozaffar et al., 2020); Xi'an, China (Song et al., 2021). The relatively low OFP could be explained by the slight VOC pollution during the sampling periods (Tab. S2). As the predominate contributor of OFP, alkenes category, especially the ethylene and propylene therein, presented great reactivity in atmospheric process and thus explains more than half of the photochemical reactivity at the three sampling sites, in line with the previous findings (An et al., 2014; Guo et al., 2017; Hui et al., 2018). Moreover, the considerable photochemical reactivity of aromatics (Li et al., 2019b), especially the toluene, *o*-xylene, *m/p*-xylene therein, makes them become the second largest contributor of OFP. Hence, alkenes and aromatics played a key role in the formation of O$_3$ and the relevant





emission sources, which are thought to be the industrial production and vehicle exhaust at the sampling sites, should be controlled in priority. By comparison, the fraction of aromatics at the JS site was higher than those at the PD and QP sites. The PD site showed high alkanes contribution compared with the JS and QP sites. In addition, the large contribution of alkenes was found at the QP site. Such phenomenon suggested that the OFP values were varied with the different land-use types. Therefore, it is necessary to study the OFP values at the varied site locations.

The OFPs of the VOC categories and $O_3$ concentration in March were suffered from the $O_3$ pollution reaching 93.13, 70.68 and 107.52 ppb at the JS, PD and QP sites, respectively (Fig. S5). It is worthwhile to note that, the relatively high VOC concentrations were observed at the JS and PD sites, resulting in the higher OFPs than that at the QP site. Under the high OFP (56.09 ± 37.73 and 45.53 ± 21.32 ppb), the concentration of $O_3$ at the JS and PD sites were unexpectedly lower than that at the QP site, indicating the unsatisfactory $O_3$ formation conditions, since the estimated OFP was proceeded within the

optimum conditions. The highest concentration of $O_3$ was observed at the QP site, which was influenced by favorable meteorological conditions and possible external transportation. Such results were in good agreement with the findings in Zhengzhou, China (Li et al., 2019b).

**3.5.2 The VOCs-$O_3$ sensitivity**

It was well documented that $O_3$ formation was sensitive to the concentration of VOCs in many megacities along with high

$NO_x$ levels (Shao et al., 2009; Ou et al., 2016; Gao et al., 2017, Li et al., 2019b). To further examine the relationship between VOCs and $O_3$, the VOCs-$S_{O3}$ sensitivity (VOCs-$S_{O3}$) was used which was the similar method in analyzing the VOCs-$S_{PM2.5}$ sensitivity. As discussed above, the $O_3$ concentrations were classified into clean level ($O_3 < 110$ μg m$^{-3}$), slight pollution level ($110 < O_3 < 120$ μg m$^{-3}$), medium pollution level ($120 < O_3 < 130$ μg m$^{-3}$), heavy pollution level ($130 < O_3 < 160$ μg m$^{-3}$), and extreme pollution level ($O_3 > 160$ μg m$^{-3}$). The variations of VOCs-$S_{O3}$ at three sampling sites are shown in Fig. 12. The

$O_3$ concentrations were more susceptible to the variation of low VOC concentrations than the high atmospheric abundance of VOCs. Therefore, more attention should be paid to the $O_3$ pollution and its evolutions during the clean and slight polluted periods. Besides, the lowest VOCs-$S_{O3}$ values were basically remain constant, while the high values showed the decreasing trends. To deeply understand the sensitivity of VOCs to $O_3$, the VOCs-$S_{O3}$ cases were subdivided into different groups with an $O_3$ concentration interval of 5 μg m$^{-3}$. The results were varied with the sampling locations, i.e., at the JS site, the average

values of VOCs-$S_{O3}$ in each group were fitted with an exponential function (Fig. 12):

$$y = 0.64 \cdot x^{-0.59}$$

The linear coefficient $k$ was calculated to be 0.41 based on the Eqs. (11-14). At the PD site, the low and high values of VOCs-$S_{O3}$ fluctuated slightly, while the remained values increased along the $O_3$ gradient. At the QP site, the VOCs-$S_{O3}$ increased with elevated $O_3$ concentration while decreased. With regard to the four VOC categories, the high VOCs-$S_{O3}$ was

700 primarily contributed from the alkenes and averaged to be 0.37, 0.40 and 0.52 at the JS, PD and QP, respectively (Fig. S7), which were agreed well with the OFP estimation results. The phenomenon suggested that the control measurements on





alkene emissions could provide direct insight into the issue of alleviating O₃ pollution at the sampling sites.

## 4. Conclusions

Herein, a concurrent atmospheric observation campaign was performed at the three supersites of Shanghai from January to March 2019. The sampling sites located at the industrial district (the Jinshan site: JS), residential district (the Pudong site: PD) and background district (the Qingpu site: QP) of Shanghai. Based on the observation data, this study has discussed in detail the pollution characteristics, primary sources and secondary pollution potentials of the atmospheric VOCs.

The mean VOC concentrations at the JS (21.39 ± 12.58 ppb) and PD (21.36 ± 8.58 ppb) sites were approximately twice higher than those at the QP site (11.93 ± 6.33 ppb). Alkanes dominated the VOC community and accounted for 50.33, 71.48 and 60.88 % of the VOCs at the JS, PD and QP sites, respectively. The diurnal variations showed the high VOC concentrations during the rush traffic hours in the morning and evening. However, the VOC concentrations on the weekends were 3.31, 10.19 and 1.19 % lower than those on the weekdays at the JS, PD and QP sites, respectively, due to the low anthropogenic activities occurring on the weekends. Overall, the VOC pollution characteristics varied with the sampling locations. The average PM₂.₅ values were 45.57 ± 27.59, 48.51 ± 27.22 and 40.27 ± 27.78 μg m⁻³ at the JS, PD and QP sites, respectively. The highest O₃ concentration was observed at the QP site (99.30 ± 24.00 ppb), followed by the JS (73.59 ± 23.59 ppb) and PD (57.48 ± 20.49 ppb) sites. The occurrence of haze pollution was characterized by the elevated concentrations of aromatics at the JS and PD sites, while the alkanes at the QP site.

The special ratios and PMF were used to investigate the local sources. Local anthropogenic emissions contributed largely to the VOC pollutions. The vehicle exhaust was determined as the predominate source at the three sites. The second largest VOC contributor was identified as industrial production at the JS and PD sites, while fuel evaporation was the second important source at the QP site. Apart from the local sources, the influence of regional transport was analyzed *via* 24-h back trajectories and PSCF. The results showed that VOCs were impacted by the north or/and northeast trajectories at three sampling sites. The highest PSCF values were observed in the area near the above sites, indicating that local source was a significant contributor to the VOC pollution.

The formation potentials of SOA and O₃ induced by the studied VOCs were also discussed. The results showed that VOCs had higher contribution to O₃ compared with PM₂.₅, and the sensitivity analysis of VOC compositions varied with the sampling sites. Specifically, the SOAFP results illustrated that BTEX was the greatest contributor to SOAFP, comprising 86.07, 96.21 and 86.38 % at the JS, PD and QP sites, respectively. The VOCs-PM₂.₅ sensitivity analysis showed that VOCs at the QP site was more sensitive to PM₂.₅ compared with other two sampling sites. Of the four VOCs categories, aromatics at the JS and PD sites and alkanes at the QP site were more sensitive to PM₂.₅. In terms of OFP, alkenes was the major OFP contributor, and meanwhile, aromatic compounds were also positively associated with the high OFP. The VOCs-O₃ sensitivity analysis showed that the VOCs-S$_{O3}$ values varied with sampling sites, and alkenes was more sensitive to O₃. The relevant results indicated that alkenes and aromatics were key concerns in reducing the atmospheric secondary pollution in



Shanghai. This study broadens the city-scale research from single-site measurement to multi-site observations, and simultaneously narrows the multi-site research from worldwide scale to city scale. The relevant results reveal the influences of land use type on atmospheric pollution, and set an example for the future VOC observation research.

**Data availability**

Measurement data in this study are available in the data repository maintained by Mendeley Data https://doi.org/10.17632/mf4gf36r9n.1 (Han, 2022).

**Acknowledgements**

This work was supported by National Natural Science Foundation of China (Nos. 22176038, 91744205, 21777025).





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

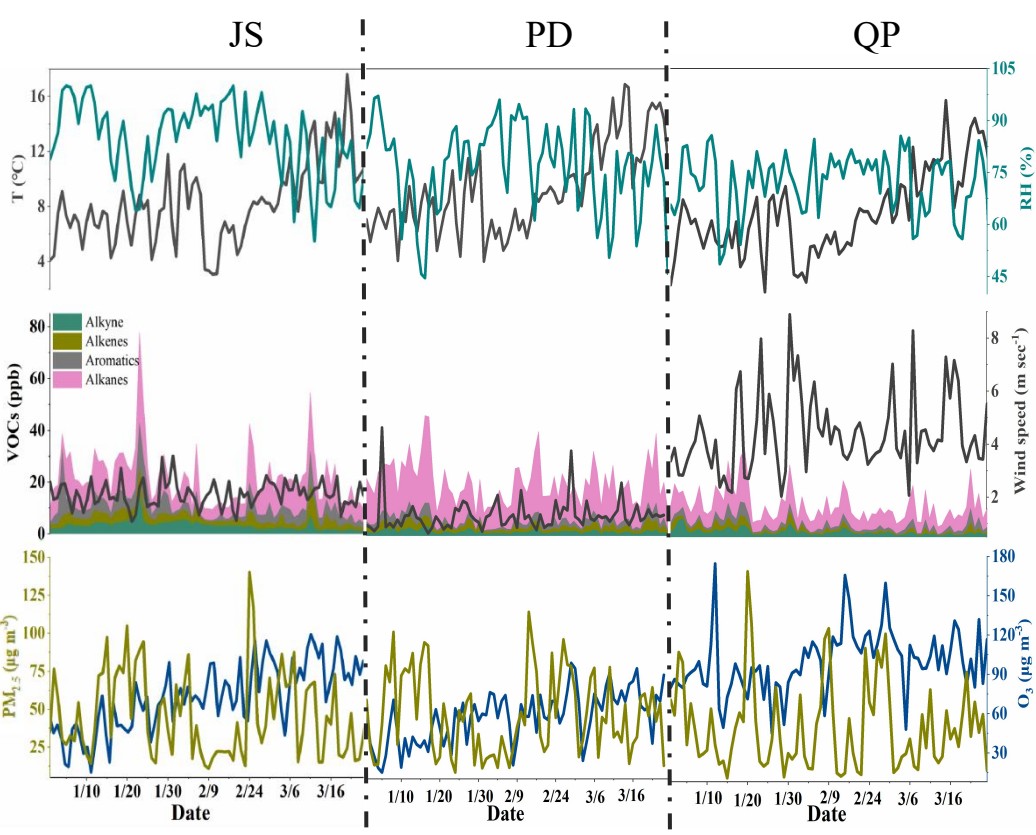

**Figure 1: Time series of temperature (T), relative humidity (RH), wind speed, VOC categories, PM$_{2.5}$ and O$_3$ at the (a) JS, (b) PD and (c) QP sites.**





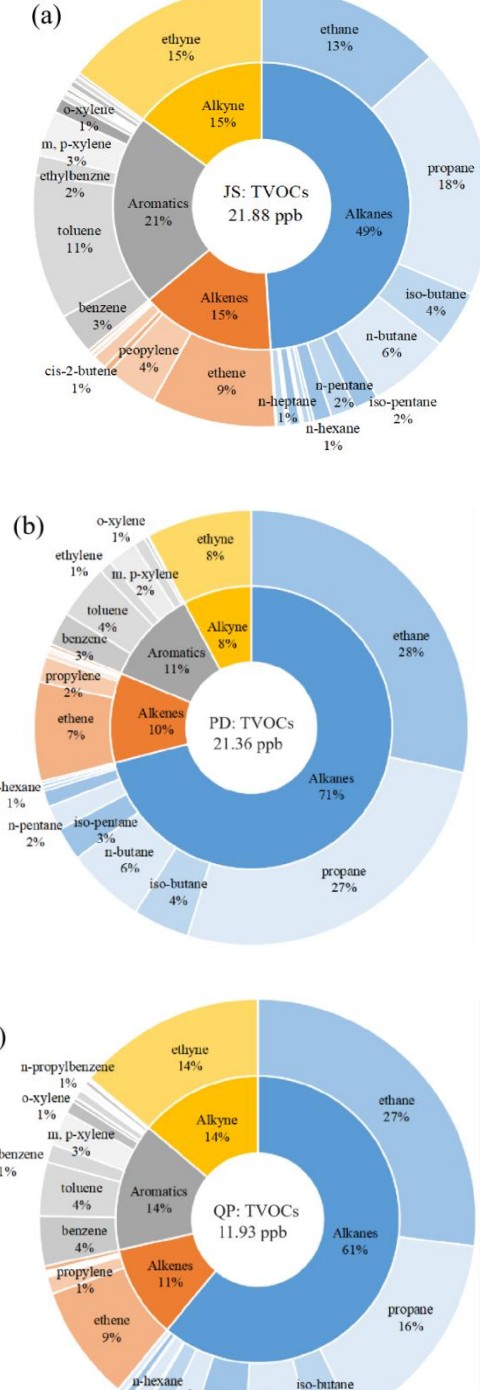

**Figure 2: Contributions of the VOC categories (43 species) at the (a) JS, (b) PD and (c) QP sites (the contributions > 1 % were marked).**


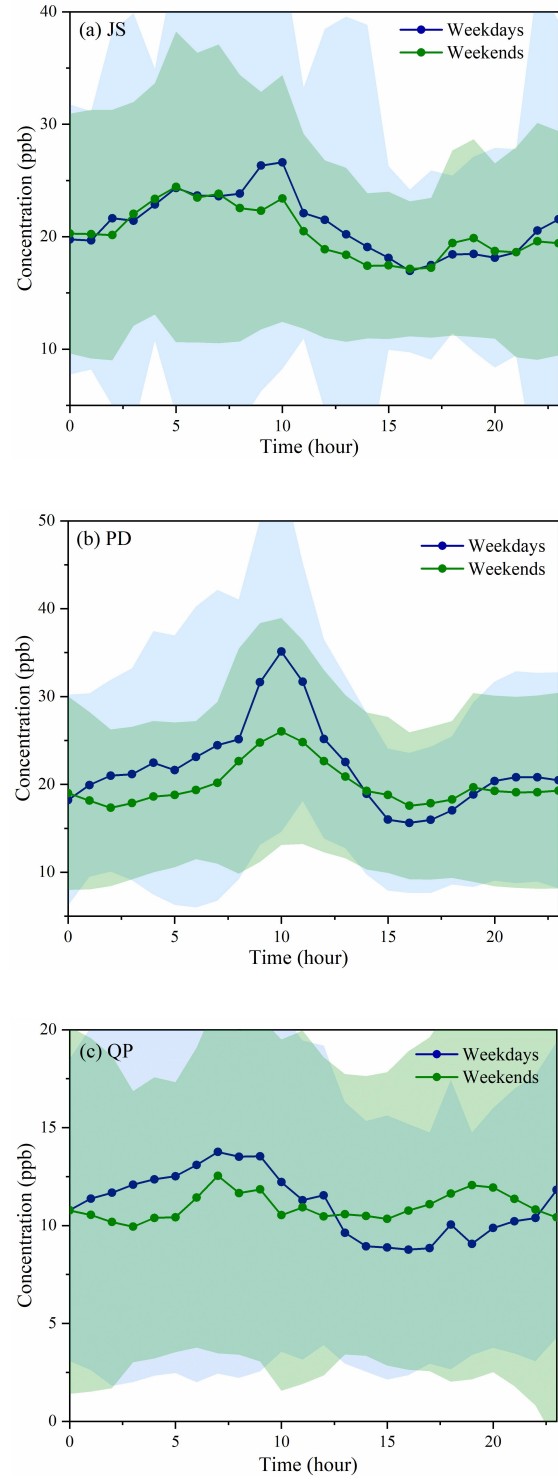

**Figure 3: Weekend effects of the TVOC concentrations at the sampling sites.**



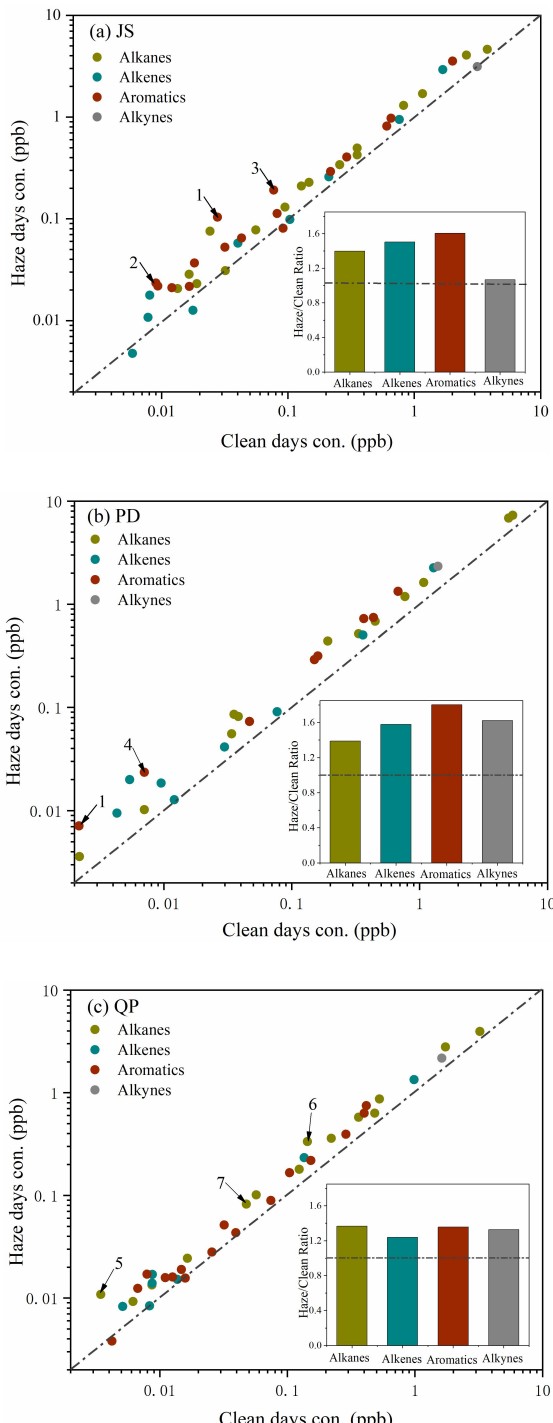

**Figure 4: Differences in VOC concentrations between the clean and haze days at the (a) JS, (b) PD and (c) QP sites. 1: 1, 2, 4-trimethylbenzene; 2: *p*-ethylbenzene; 3: *m*-dimethylbenzene; 4: *m*-ethylbenzene; 5: 2, 2, 4-trimethylpentane; 6: *n*-hexane; 7: *h*-heptane.**

**Figure 5: Specific ratios of the target VOC species for the source identification. The linear correlations between toluene and benzene at the (a) JS, (b) PD and (c) QP sites, and (d)** *m, p***-xylene and ethylbenzene and (e)** *iso***-pentane and** *n***-pentane at JS (orange), PD (yellow) and QP (blue).**





a. JS







b. PD





c. QP

Figure 6: Sources profiles and contributions of VOCs at the (a) JS, (b) PD and (c) QP sites.



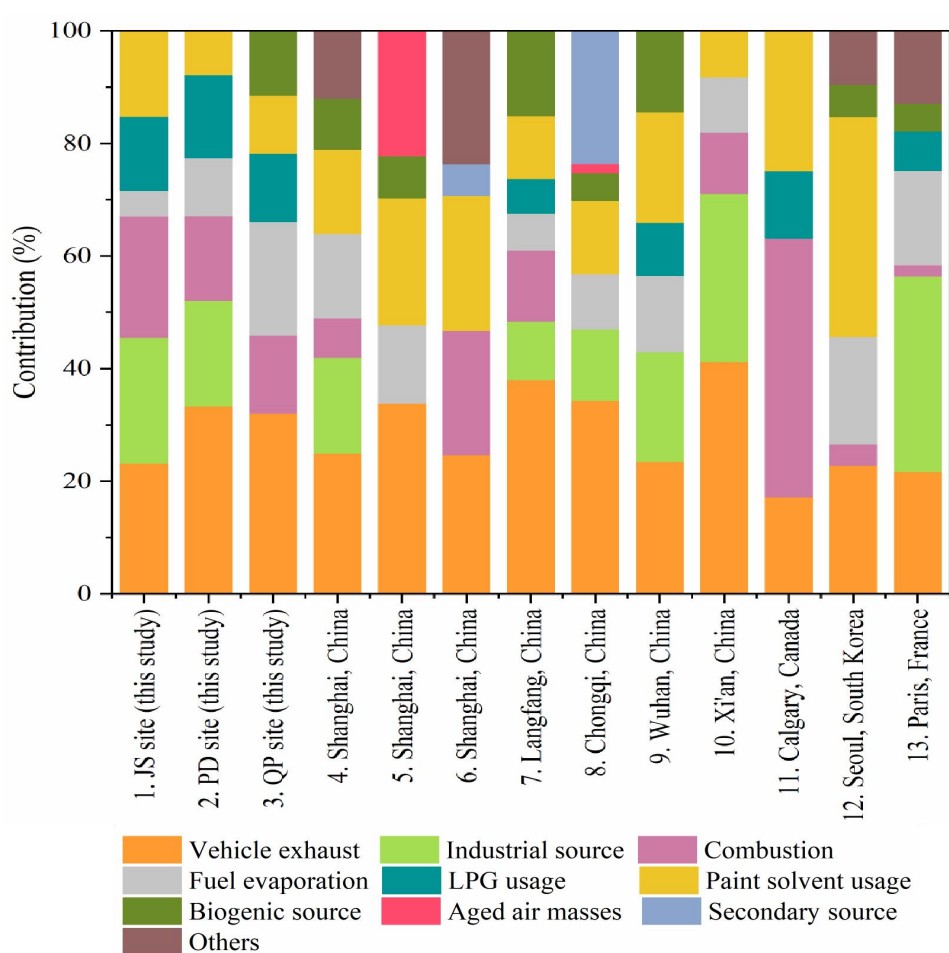

**Figure 7: Primary emissions of the observed atmospheric VOCs of this study, and those measured at different locations or during different periods: 1, 2, 3. Shanghai, China (this study); 4. Shanghai, China (Cai et al., 2010b); 5. Shanghai, China (Liu et al., 2019); 6. Shanghai, China (Liu et al., 2021); 7. Langfang, China (Song et al., 2019a); 8. Chongqi, China (Li et al., 2018); 9. Wuhan, China (Hui et al., 2018); 10. Xi'an, China (Song et al., 2021); 11. Calgary, Canada (Bari and Kindzierski, 2018); 12. Seoul, South Lorea (Song et al., 2019b); 13. Paris, France (Gaimoz er al., 2011)**





a. JS

b. PD

c. QP

**Figure 8: Backward trajectory cluster analysis (24 h) and PSCF analysis at the JS, PD and QP sites.**



Figure 9: The SOAFP analysis of different VOC species at the JS, PD and QP sites.





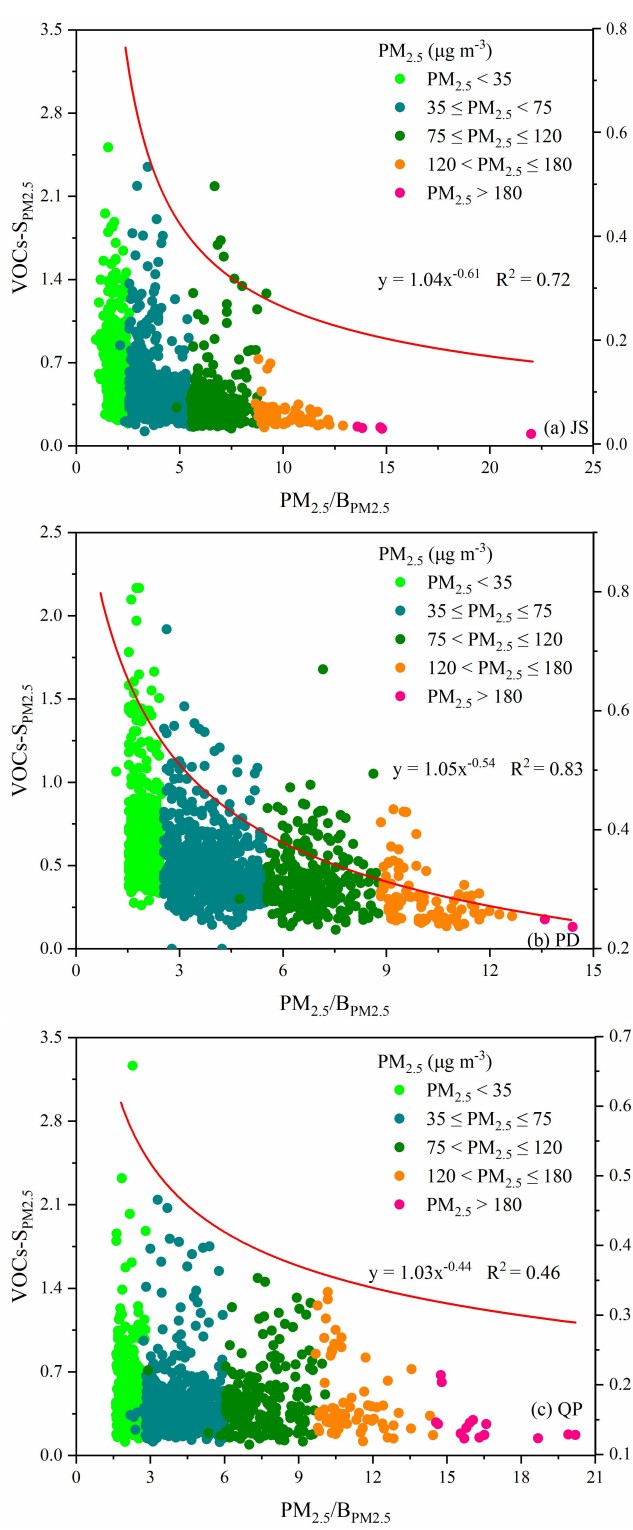

**Figure 10: The variations of VOCs-S$_{PM2.5}$ at the different level of PM$_{2.5}$ at the JS, PD and QP sites.**



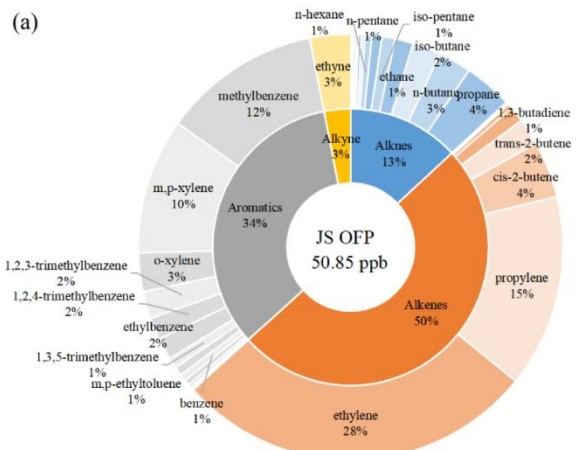

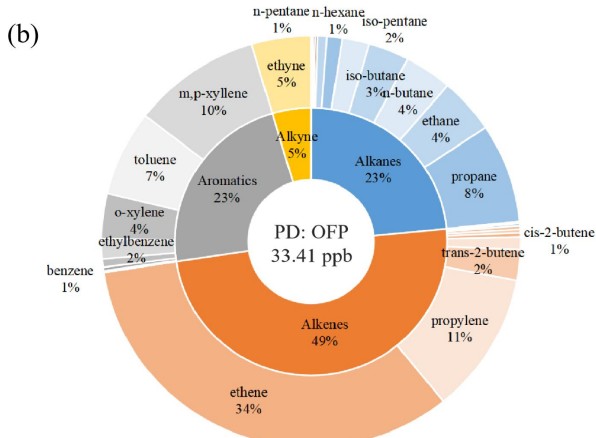

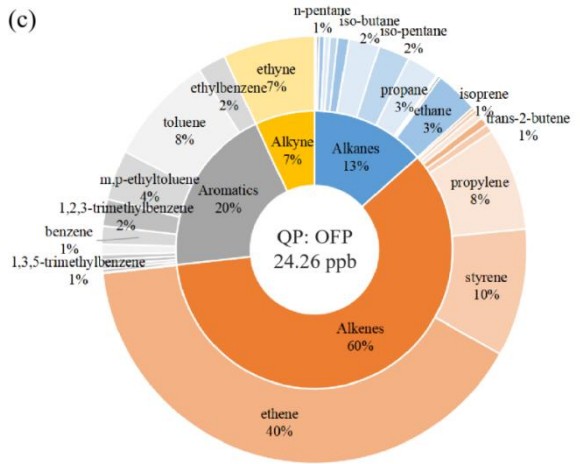

**Figure 11: Contributions of VOCs to OFP at the (a) JS, (b) PD and (c) QP sites (the contributions > 1 % were marked).**

**Figure 12: The variations of VOCs-S$_{O3}$ at the different level of O$_3$ at the JS, PD and QP sites.**

745



**Table 1: Air mass cluster trajectories at the JS, PD and QP sites.**

| Site | Cluster | Ratio (%) | P_Ratio (%) | P_TVOCs (ppb) |
|------|---------|-----------|-------------|---------------|
| JS | 1 | 16.67 | 28.04 | 21.13 |
| | 2 | 29.07 | 18.03 | 34.03 |
| | 3 | 19.46 | 16.62 | 15.62 |
| | 4 | 12.26 | 12.75 | 22.40 |
| | 5 | 5.92 | 16.62 | 17.70 |
| | 6 | 16.62 | 5.93 | 22.34 |
| PD | 1 | 18.04 | 33.13 | 27.03 |
| | 2 | 17.80 | 10.42 | 29.45 |
| | 3 | 34.19 | 21.59 | 53.48 |
| | 4 | 8.57 | 15.28 | 26.85 |
| | 5 | 10.84 | 8.53 | 24.25 |
| | 6 | 10.56 | 11.04 | 39.95 |
| QP | 1 | 14.20 | 18.47 | 10.15 |
| | 2 | 17.57 | 13.50 | 23.63 |
| | 3 | 31.87 | 27.81 | 10.07 |
| | 4 | 15.20 | 16.79 | 7.53 |
| | 5 | 8.71 | 14.52 | 15.39 |
| | 6 | 12.45 | 8.91 | 12.81 |