# Peer review of "Measurement report: VOC characteristics at the different land-use types in Shanghai: spatio-temporal variation, source apportionment, and impact on secondary formations of ozone and aerosol"

_Atmospheric Chemistry and Physics, 2022_

## Referee Comment (RC1)

The manuscript presents a study on volatile organic compounds (VOCs) at three sites in a Chinese megacity, ~Shanghai. The concentration, composition, sources and of VOCs have been extensively studied, especially in the major cities like Shanghai. The authors claim multiple-site comparison as the main selling point. However, it is not enough to provide new insights that the authors expect. Namely, I have reservations about the novelty of this study. Moreover, I have serious concerns on the rationality of the methods and quality of the results presentation. The lots of grammatical errors also make the manuscript very difficult to be reviewed. Overall, the manuscript is well below the average of papers published in Atmospheric Chemistry and Physics, so I do not think it is worth publishing in its current state. However, I do not mind providing some specific comments for the authors' reference and reviewing a resubmitted edition after it is substantially improved.

**Title:**

1. The term "secondary formation potentials" is not a common expression. It is not clear what you are referring to.

**Abstract:**

2. Line 29: VOCs-SO3 has not been defined before. So is SOA in line 31.

3. Line 31: "new insights". I do not think the paper at its current state provides new insights into the accurate air quality management.

**Introduction:**

4. Major comment: The whole section is rather simple and most of the contents (if they are right) can be found in text book. I do not think it is necessary to elaborate them in a research article. What's worse, I am confused by the introduction of some basic knowledge. For example, line 40:  $RO_2$  is formed following oxidation of VOCs, but VOCs are oxidized by OH, O3 and NO3 (NOT RO2). Also, the oxidation does not necessarily lead to formation of secondary VOCs, although some species, e.g., formaldehyde, can be formed through photochemical reactions. Then, not all secondary VOCs can be transformed to SOA. Lines 91-96: This is not an accurate summary of the roles of VOCs in SOA formation. Lines 98-100: I do not get the point why there are strong industrial, vehicular and power plant emissions in mountainous area. Moreover, motor vehicles and power plants are significant sources of NOx. Then, how to explain the NOx-sensitive regime for O3 formation?

5. Major comment: It is not clear what the authors mean by pollution characteristics, which is too general. It is also not clear what the knowledge gap is. The authors must make it clear what the manuscript adds to the current understanding of VOCs in Shanghai.

6. Inaccurate expressions and grammatical errors are everywhere throughout the manuscript. I cannot list all of them, just give some examples here: line 93: "...that declines the vapor pressure reduction", line 94: pPM, gas-particular partition; line 95: a significantly decreased in the vapor pressure; line 98: transition ~ regime, line 100: strong emissions of industrial, vehicular, power

and biogenic, line 101:  $NO_x$  transition regime (what is it? I never saw this kind of expression), same for the "VOCs transition area" in line 104; line 106: varied photochemical reactions; line 108: "VOCs are likely to response to the pollution of PM2.5 and O3" –I am not sure if I understand correctly because of language problem; if my understanding is correct, what is the point of studying the responses of VOCs to PM2.5 and O3, rather than the other way around?; line 112: pollution VOC characteristics.

7. Lines 43-45: What's the point of emphasizing the 57 PAMS VOCs? There are a wide range of VOCs that can be the precursors of  $O_3$  and SOA.

8. Lines 121-122: Is there any evidence proving that VOC pollution in Shanghai is more serious than ever before? It is contradictory to the statement in lines 71-71 "the VOC concentrations of China have decreased in the recent years along with the effective control strategies".

9. Lines 122-130: It reads like pollution characteristics just means concentration, which is not true.

10. Lines 130-131: I cannot agree. In fact, sources and contributions of VOCs to  $O_3$  and SOA have been well documented.

11. Lines 131-132: Which studies are you referring to when you say "ten years ago" – a specific time frame? At least, the studies you are referring to should be discussed.

**Methodology:**

Many problems in this section need to be justified or clarified.

12. Major comment: Different instruments, as well as analytical methods, were used for the analysis of VOCs at the three sites. How did the authors reconcile the data so that they can be compared? What is TD300 (line 177) that is not defined? In general, the small molecule and large VOCs are detected by FID and MSD, separately. Lines 182-184: What are the accuracies and detection limits for the minority of the species, i.e., those beyond the "95% and most VOC components", and what is the range of the precision? Line 188: What is the SEAS site? It has never been defined before.

13. Description about this method in section 2.3 is confusing. How can the spatial heterogeneity be determined for a single site, as stated in line 190? Line 193: Does j represent site or dataset? Contradictory descriptions.

14. Major comment: There are significant deviations in the understandings of PMF. Lines 202-203: the descriptions of  $g_{ik}$ ,  $f_{kj}$  and  $e_{ij}$  are totally wrong. Lines 204-207: The function Q is introduced. However, it is not clear what the purpose of introducing it is and how the authors used it? Lines 210-212: "EF is the error faction and can be set to 0.05-0.2" – What was the EF the authors set in this study? How did the authors determine the solution with seven factors as the optimal one?

15. Lines 223-225: How did you determine the number of polluted and all trajectories in a grid, and how was the weight function  $W_{ij}$  applied?

16. Major comment: The equation (9) calculates the responses of VOC to  $O_3$ , which is opposite to the statement in lines 245-246 that "The characteristic structure and reactivity could influence the contribution of VOCs to  $O_3$  formation". Lines 249-252: The rationality of using 100 as a threshold of background  $O_3$  should be justified. Why 100? Note that it is a quite high value, especially in cool seasons. It is also totally wrong to assign the VOC concentrations during the  $O_3$ -background time period as background VOC concentrations. In most cases, the patterns of  $O_3$ and VOCs are inconsistent. For example,  $O_3$  got lowest values at night and in early morning when VOCs are at high levels. Lines 251-252: What is the logic behind? Why is VOCs influenced by the variation of  $O_3$ , and not the other way around? Lines 254-255: The logarithmic conversion is also problematic. Equation 12 should be written as lny = lna + blnx.

15. Inaccurate expressions and grammatical errors in this section include but are not limited to the followings. Lines 180-181: "The samples were condensed low-carbon ( $C_2$ - $C_6$ ) compounds and high-carbon ( $C_6$ - $C_{12}$ ) compounds ..."; Line 185: "trace instruments"; Lines 211-212: "option solution", "greatest solution"; Line 214: "observe the back trajectories, source and direction of pollutants"; Line 218: "This study was determined the 24-h back trajectory".

**Results and discussion:**

16. Line 266: "60 VOC species" is contradictory to the statement that "Totally 43 species of VOCs were observed" in line 183.

17. Lines 271-273: What's the point of comparing the wind speed that is very spatially uneven?

18. Lines 274-288: The comparisons are rather simple. Are there same number of species, same species, same sampling season and etc.? Without discussion on these factors, the comparisons are meaningless.

19. Lines 295-312: I do not see the necessity of discussing such simple facts with too many words.

20. Lines 314-315: I am surprised to see such high levels of  $O_3$  in the sampling period. Without any doubt, the authors made mistakes in calculation or unit conversion.

21. Lines 315-317: Readers would have no idea what the point of this discussion is. Are the dates special?

22. Lines 317-318: I do not think this was the reason for the correlation. Otherwise, did you see correlation between VOCs and  $O_3$ , where the former was also precursors of the later?

23. Lines 320-324: Why not refer to sources of  $PM_{2.5}$  and VOCs in Shanghai. Transportation as the main source of  $PM_{2.5}$  and VOCs in different cities does not necessarily mean the homology  $PM_{2.5}$  and VOCs in Shanghai.

24. Lines 325-331: First, I do not think the correlation is worth discussing. In most cases, the diurnal patterns of VOCs and  $O_3$  are opposite. Second, the opposite patterns are mainly due to inconsistent patterns of VOC emission (e.g., emissions in morning and evening rush hours) and

 $O_3$  formation (e.g., daytime). The discussions are far-fetched and I do not understand ", and counteraction was imposed by uncertain factors during the formation of  $O_3$ ."

Due to the obvious deficiencies and too many comments if I go on, I have to stop the review here. Please note that to save time I have not reviewed the next sections. I suppose there are also lots of problems therein. However, I would be willing to review a substantially improved manuscript.

---

## Referee Comment (RC2)

The manuscript presents a study based on the concurrent observations of volatile organic compounds (VOCs) at three supersites sites in Shanghai. The characteristics of VOCs, ozone formation potential, secondary organic aerosol formation potential and emission sources are discussed. The influence of different land use type on VOC profiles and atmospheric oxidation capacity is worthy of study. However, the discussion in this manuscript is not enough to highlight this viewpoint. The author should focus more on the discussion of the influence of land use type on VOC profile and atmospheric oxidation capacity. Moreover, I find irrationality of the methods and inaccuracy of some conclusion. Also, the grammar needs to be thoroughly revised. Generally, under its current version, this paper needs substantial revision to reach the standard for publication. However, the scope of this manuscript is good and the measurements can provide deeper understanding of the influence of anthropogenic activities on megacities or city clusters. A revised edition is encouraged for resubmission.

**Major comment**

Major comment 1: As the author have stated in Line 110-113:"limited knowledge is available on the multi-site research at a city level …..", authors should well explain how does multi-site observation bring us new insights different from single site observations. Furthermore, to my knowledge, there are a lot of studies about VOC characteristics in Shanghai during different measurement period, at different locations and at different years. The authors should also elucidate how this work bring new insights different from just comparing the reported results. Finally, the authors should rearrange the whole manuscript emphasizing on the difference of the observation results of three sites and the discussion about the influence of different land use type.

Major comment 2: The instruments applied at three sites are different. I do not think the comparison is convincing, without any illustration about the data reconciliation. Since the main scope is to compare the VOC data observed at different sites, the data quality

is of great importance to the final conclusion. The authors should discuss more about the detect of limit, accuracy of all measured VOC species.

Major comment 3: The authors should discuss more on how to determine the final PMF solution before discussing the PMF results.

**Specified comments**

1. The term" secondary formation potentials" in the title is confusing, it will be better to use "ozone and SOA formation potential".

2. Line 29-30: The sentence "The VOCs-$O_3$ sensitivity indicated that VOCs-$S_{O3}$ values varied at the different sites and were primarily controlled by the alkene-related reactions". It is confusing at this place to see VOCs-$S_{O3}$ without illustrations of the methodologies. It will be better to just state the main collusion here.

3. Line 31-34: How the findings provide new insights into the accurate control of different land-use type? Moreover, how the results highlight the importance of multiple-site measurements?

4. Line 35- 139 I do not think it is necessary to put some many basic knowledge in the introduction part.

5. Line 140-168: It will be much easier to compare the difference of three sites, if the authors can make a table about potential sources at different sites, and the results of published works conducted at the three sites (if there were).

6. Line 177: what's TD300?

7. Line 181-183: $R^2$ of what? Calibration results? Please clarify. Odd expression about "accuracy of 95% of compounds". What about the accuracy of the left 5%?

8. Line 188: what's SEAS site?

9. Line 190: how can one site have spatial heterogeneity? Do you mean each pair of sites?

10. Line 202: I think the authors need to rewrite the description about $g_{ik}$, $f_{kj}$, and $e_{ij}$.

11. Line 205-212: The authors should explain how the Q value works in PMF model, for example, how it can help determining the PMF solution.

12. Line 224-225: How is weight function applied in PSCF?

13. Line 265-266: Please confirm exactly how many VOC species were observed, 60 or 43? It will be better to list the observed VOC species in a table.

14. Line 277-284: How does the comparison meaningful, without clarifying the observed VOC species?

15. Lin310-312: "This phenomenon was because there similar VOC emission intensity …..". The discussion about different COD is too simple.

16. Line 317-318: "Statistically, VOCs were found to be positively correlated with $PM_{2.5}$ due to the fact that VOCs were a significant precursor of $PM_{2.5}$." I do not think this explain is correct.

17. Line 324-331: What's the view point the authors want to discuss? Determining the controlling factor of $O_3$ formation merely based on the ratio of VOCs/NOx ratio is too simple. Moreover, in Line 329 "a higher proportion of OH radical reacted with $NO_2$ to suppress the $O_3$ formation", how the authors get this conclusion from the VOCs/NOx ratio?

18. The authors have highlighted some differences among three sites. Such as in Line 356-357, "the contribution of toluene at JS site was markedly increased (~ 3times) relative to the other two sites." However, such discussion about the difference of VOC markers is insufficient in this section. The authors should discuss more about such difference and make connection of the observed difference with the difference of land use type or emission sources.

19. Line 369-406: the authors discuss too much about the diurnal variation pattern, which is similar with the reported diurnal characteristics in many other studies. There are some interesting parts such as at Line 390-391: "The VOC concentrations on the weekends were 3.31, 10.19 and 1.19% lower than those on the weekdays," and at Line 394-395: "It should be noted that there were narrow discrepancies of VOC concentrations at the site between the weekdays and weekends ….". The difference at weekends and workdays can be attributed to the influence of the local emission sources. The author should rearrange this section focusing on the discussion about the difference of weekend/ holiday effects.

20. Line 415-416: how is industrialization and urbanization results in the stagnant weather condition?

21. Line 430: How about the "clean-haze" discrepancy of other VOC species?

**Grammatical errors and confusing expressions**

Line 36: "…both of which formation are …" better to use two sentences here.

Line 180-181: "The samples were condensed low-carbon (C2-C6) compounds and high-carbon (C6-C12) compounds …". Rearrange the sentence.

Line 185: what is trace instruments?

Line 211: "greatest solutions" should be optimal solutions.

Line 218: Should be "This study was determined by the 24-h back trajectory".

Line 266-267: "The temperatures were averaged to be ….."

Line 300: "a large number of organizations….". Improper use of organizations here.

Line 304: "The reduced VOC concentrations coincide ….". This sentence is confusing.

Line 356-357: "the contribution of toluene at JS site was markedly increased (~ 3times) relative to the other two sites." Should be higher instead of increased.

Line 375: Should be "the VOC concentrations also tended to increase"
Line 424: should be "At the QP site".

I have not listed all the grammatical errors and inaccurate expressions. Please check the whole manuscript and make corrections.

---

## Author Comment (AC1)

**Response to the Comments of the Reviewers**

Dear Editor and Reviewers,

We acknowledge the comments and encouragement of two reviewers, and are also grateful to the efficient serving of the editor. Here we submit our revised manuscript **"Measurement report: VOC characteristics at different land-use types in Shanghai: spatio-temporal variation, source apportionment, and impact on secondary formations of ozone and aerosol" (Manuscript number: acp-2022-250)**, as well as a thorough, point-by-point response to each point raised from the reviewers. The revisions to the manuscript are highlighted in blue words in the provided "Response to the Comments of the Reviewers". Additionally, there is a clean revised manuscript as required.

We greatly appreciate those comments and valuable suggestions from the reviewers. The manuscript has been greatly improved. We do feel that we have demonstrated our efforts in the revised manuscript.

Yours sincerely,

Hongbo Fu and Co-authors.

Department of Environmental Science & Engineering

Fudan University

200000 Shanghai China

E-mail: fuhb@fudan.edu.cn

**Response to the Reviewer #1**

**General Comments:**

The manuscript presents a study on volatile organic compounds (VOCs) at three sites in a Chinese megacity, ~ Shanghai. The concentration, composition, sources and of VOCs have been extensively studied, especially in the major cities like Shanghai. The authors claim multiple-site comparison as the main selling point. However, it is not enough to provide new insights that the authors expect. Namely, I have reservations about the novelty of this study. Moreover, I have serious concerns on the rationality of the methods and quality of the results presentation. The lots of grammatical errors also make the manuscript very difficult to be reviewed. Overall, the manuscript is well below the average of papers published in Atmospheric Chemistry and Physics, so I do not think it is worth publishing in its current state. However, I do not mind providing some specific comments for the authors' reference and reviewing a resubmitted edition after it is substantially improved.

**Response:**

We would like to thank reviewer #1 for carefully reading our manuscript and for your valuable and constructive comments. All your suggestions are very important and they are of great significant to our scientific research. We carefully revised and improved each part according to the reviewer's suggestions.

The long-term VOC emission inventory highlighted that the VOC emission varied with the land-use types (Li et al., 2019). The observation campaign also showed that VOC concentrations were largely influenced by the land-use type (Tang et al., 2008; Kumar et al., 2018; Zhang et al., 2018). Besides, the land-use types not only influence the VOC concentrations but also the sources especially the anthropogenic sources (Yoo et al., 2015; Chen et al., 2017; Wang et al., 2017; Jookjantra et al., 2022). Additionally, the diversities of VOC concentrations among the different land-use types could cause the distinct ozone and SOA formation potentials (OFP and SOAFP), resulting in the variations of $O_3$ and SOA concentrations (Song et al., 2021; Zhan et al., 2021; Liu et al., 2022). Shanghai is regarded as an ideal area to perform atmospheric measurements with the different land-use types. However, many studies

were mainly focused on the single-site measurements, particularly conducted at the urban site in Shanghai, resulting that the impact of land-use type on VOC characteristics is still unclear to date. Moreover, they mainly concentrated on the $O_3$ characteristics, while the $O_3$ and SOA formations from VOCs and the relationship between VOCs together with $PM_{2.5}$ and $O_3$ were rarely analyzed. Additionally, the distinct land-use types among the sampling sites were observed. The JS site is located in the Second Jinshan Industrial Area of Shanghai as the industrial district, and surrounded by many chemical factories. The PD site is located in the Pudong New Area as the residential and commercial mixed districts, and surrounded by residences and administrative areas. The QP site is located at the southeast of Dianshan Lake as the background district, and surrounded by many farmlands and forests. Given the factors mentioned above, in this study, the concurrent multiple-site and high time-resolution measurement of the VOCs with three typical land-use types in Shanghai for their characteristics, sources and ozone and SOA formation potentials was performed. The results at the multiple-site measurement benefit the government to establish efficient and specific environmental control measures according to the specific land-use types. Lastly, we would like to thank reviewer for the positive comments again.

**Comment 1:** The term "secondary formation potentials" is not a common expression. It is not clear what you are referring to.

**Response:**

Thank you for your comments. The "secondary formation potentials" is referred to the secondary organic aerosol formation potential (SOAFP) and ozone formation potential (OFP). We revised the title to

*"Measurement report: VOC characteristics at different land-use types in Shanghai: spatio-temporal variation, source apportionment, and impact on secondary formations of ozone and aerosol"*

**Comment 2:** Line 29: VOCs-$S_{O3}$ has not been defined before. So is SOA in line 31.

**Response:**

Thank you for your comments. We added the definition in the revised manuscript.

Line 30:

> *"Alkenes and aromatics are both the key concerns in controlling the VOC-related pollution of $O_3$ and secondary organic aerosol (SOA) in Shanghai."*

The calculation of VOC-$O_3$ sensitivity (VOCs-$S_{O3}$) was deleted after consideration.

**Comment 3:** Line 31: "new insights". I do not think the paper at its current state provides new insights into the accurate air quality management.

**Response:**

Thank you for your suggestion. We changed the words in the revised manuscript.

> *"These findings provide more information on the accurate air-quality control at a city level in China."*

**Comment 4:** The whole section is rather simple and most of the contents (if they are right) can be found in text book. I do not think it is necessary to elaborate them in a research article. What's worse, I am confused by the introduction of some basic knowledge. For example, line 40: $RO_2$ is formed following oxidation of VOCs, but VOCs are oxidized by OH, $O_3$ and $NO_3$ (NOT $RO_2$). Also, the oxidation does not necessarily lead to formation of secondary VOCs, although some species, e.g., formaldehyde, can be formed through photochemical reactions. Then, not all secondary VOCs can be transformed to SOA. Lines 91-96: This is not an accurate summary of the roles of VOCs in SOA formation. Lines 98-100: I do not get the point why there are strong industrial, vehicular and power plant emissions in mountainous area. Moreover, motor vehicles and power plants are significant sources of $NO_x$. Then, how to explain the $NO_x$-sensitive regime for $O_3$ formation?

**Response:**

Thank you for your question. We have added professional knowledge into the new section to highlight the impact of land-use types on VOC concentrations, sources and $O_3$ and SOA formation potentials, specified in newly lines 39-103.

For example, lines 55-57:

> *"The long-term VOC emission inventory highlighted that there were significant spatial discrepancies of VOC emissions (Li et al., 2019a). The observation campaign also showed that VOC concentrations varied with the sampling sites.*

*These phenomena were attributed to the fact that VOC concentrations were closely correlated with the land-use types."*

Lines 63-65:

*"Besides, the land-use types not only influence the VOC concentrations but also the sources especially the anthropogenic sources (Yoo et al., 2015; Chen et al., 2017; Wang et al., 2017; Jookjantra et al., 2022)."*

Lines 72-73:

*"The diversities of VOC concentrations among the different land-use types could affect the ozone and SOA formation potentials (OFP and SOAFP), resulting in the variations of $O_3$ and SOA concentrations."*

Lines 77-78:

*"In detail, atmospheric VOCs undergoes degradation to produce oxidants ($HO_2$ and $RO_2$), which further oxidizes atmospheric NO, followed by producing $NO_2$ and the formation of $O_3$ finally via the photochemical pathways (Wang et al., 2017). "*

Lines 81-85:

*"As the key precursor of SOA, VOCs can be oxidized to produce the low VOCs, followed by the formation of SOA via homogeneous nucleation (Merikanto et al., 2009). Moreover, the partitioning of semi-volatile products from VOCs and oxidants gas-phase photochemical reactions to form SOA (Pankow, 1994; Lim et al., 2010). Additionally, low VOCs are produced via the aqueous-phase reactions in atmospheric waters e.g., clouds, fogs, and aerosol water which are largely retained in the particle-phase to generate SOA (Lim et al., 2010)."*

Lines 87-94:

*"The expanding urbanization and industrialization jointly aggravate the VOC pollution. Moreover, the $O_3$ concentration at the urban area in Shanghai increased by ~ 67 % from 2006 to 2015 with the growth rate of 1.1 ppbv pear year (Gao et al., 2017). The maximum 1-hour concentration of $O_3$ could exceed 380 μg $m^{-3}$ during polluted days (Shi et al., 2015; Gu et al., 2020). Such scenario suggested that $O_3$ played an important role in atmospheric pollution and*

*Shanghai was suffering from heavy O₃ pollution. Additionally, the large changes of land-use occurred in Shanghai due to the rapid development e.g., many cultivated areas became urban and/or industrial zones, resulting in the diverse land-use types (Tian et al., 2017). Therefore, Shanghai is regarded as an ideal area to perform atmospheric measurements with the different land-use types."*

**Comment 5:** It is not clear what the authors mean by pollution characteristics, which is too general. It is also not clear what the knowledge gap is. The authors must make it clear what the manuscript adds to the current understanding of VOCs in Shanghai.

**Response:**

Thank you for your comment. The VOC pollution characteristics involve the concentration variations, primary sources and the impact on $O_3$ and $PM_{2.5}$ formation. We rewrote some descriptions in the revised manuscript.

For example, line 56:

*"The observation campaign also showed that VOC concentrations varied with the sampling sites."*

Lines 62-63:

*"However, the reported VOC concentrations were widely discussed by single-site measurements, the limited knowledge is available on the multi-site research at a city level."*

Lines 653-655:

*"Based on the observation data, this study carefully discussed the concentration variations, primary sources, ozone and SOA formation potentials of the atmospheric VOCs influenced by land-use types. "*

The knowledge gap is how land-use types influences VOC concentrations, sources and ozone and SOA formation potentials in Shanghai, China. We added the discussion at the Sec. 4. The results herein could provide scientific-based information for policymakers to establish targeted strategies of alleviating VOC pollution. For example, the JS site exhibited higher fractions of aromatics and alkenes, particularly toluene and propylene, than those at the PD and QP sites. The VOC concentration in the early morning (5:00 LT) at the JS site was higher than those at the other two sites.

This result did necessarily correlate with the fact that the JS site is close to the industrial area with heavy industrial emissions, suggesting that industrial activities were key factors of VOC pollution at the JS site. Moreover, the industrial emission and biogenic source showed slight contributions to VOC concentrations at the QP and JS/PD sites, respectively. It was consistent with the regional characteristics of anthrogopenic activities dominated by land-use types. Additionally, the results of VOCs-$S_{PM2.5}$ varied with the land-use types. The aromatics at the JS and PD sites, as well as alkanes at the QP site played crucial roles in the VOC-induced haze pollution. The relevant emission sources, which are thought to be the industrial production at the JS/PD sites and vehicle exhaust at the QP site, should be controlled in priority. Therefore, these findings could provide more information on the accurate VOCs control in Shanghai, China. The results shown herein highlight that the simultaneous multiple-site measurements with the different land-use type in the megacity or city cluster could be more appropriate to fully understand the VOC characteristics relative to a single-site measurement performed normally.

**Comment 6:** In accurate expressions and grammatical errors are everywhere throughout the manuscript. I cannot list all of them, just give some examples here: line 93: "…that declines the vapor pressure reduction", line 94: pPM, gas-particular partition; line 95: a significantly decreased in the vapor pressure; line 98: transition ~ regime, line 100: strong emissions of industrial, vehicular, power and biogenic, line 101: $NO_x$ transition regime (what is it? I never saw this kind of expression), same for the "VOCs transition area" in line 104; line 106: varied photochemical reactions; line 108: "VOCs are likely to response to the pollution of $PM_{2.5}$ and $O_3$" –I am not sure if I understand correctly because of language problem; if my understanding is correct, what is the point of studying the responses of VOCs to $PM_{2.5}$ and $O_3$, rather than the other way around?; line 112: pollution VOC characteristics.

**Response:**

Thank you for your suggestion. We have revised the statement in the revised manuscript.

For example, lines 62-63:

*"However, the reported VOC concentrations were widely discussed by single-site measurements, the limited knowledge is available on the multi-site research at a city level."*

Lines 76-78:

*"In detail, atmospheric VOCs undergoes degradation to produce oxidants ($HO_2$ and $RO_2$), which further oxidizes atmospheric NO, followed by producing $NO_2$ and the formation of $O_3$ finally via the photochemical pathways (Wang et al., 2017)."*

Lines 82-83:

*"Moreover, the partitioning of semi-volatile products from VOCs and oxidants gas-phase photochemical reactions to form SOA (Pankow, 1994; Lim et al., 2010)."*

**Comment 7:** Lines 43-45: What's the point of emphasizing the 57 PAMS VOCs? There are a wide range of VOCs that can be the precursors of $O_3$ and SOA.

**Response:**

Thank you for your suggestion. The reason for emphasizing the 57 PAMS VOCs is that these VOCs contribute more on the $O_3$ formation compared with other VOC species. We revised the sentence in the new manuscript.

*"Photochemical Assessment Monitoring Stations (PAMS) have confirmed that totally 57 VOCs, including $C_2$-$C_{10}$ alkanes, alkenes, alkynes and aromatics are extremely contributed to the formation of $O_3$ (US EPA, 1990)."*

**Comment 8:** Lines 121-122: Is there any evidence proving that VOC pollution in Shanghai is more serious than ever before? It is contradictory to the statement in lines 71-71 "the VOC concentrations of China have decreased in the recent years along with the effective control strategies".

**Response:**

Thank you for your comment. The original lines 71 and 121-122 have been deleted.

**Comment 9:** Lines 122-130: It reads like pollution characteristics just means concentration, which is not true.

**Response:**

Thank you for your suggestion. The VOC pollution characteristics include the VOC concentrations, sources, ozone and SOA formation potentials. We revised the sentence in the new manuscript.

For example, line 56:

*"The observation campaign also showed that VOC concentrations varied with the sampling sites."*

Lines 62-63:

*"However, the reported VOC concentrations were widely discussed by single-site measurements, the limited knowledge is available on the multi-site research at a city level."*

**Comment 10:** Lines 130-131: I cannot agree. In fact, sources and contributions of VOCs to $O_3$ and SOA have been well documented.

**Response:**

Thank you for your suggestion. The original lines 130-131 have been deleted. We added the effect of land-use types on $O_3$ and SOA formation potentials.

**Comment 11:** Lines 131-132: Which studies are you referring to when you say "ten years ago" - a specific time frame? At least, the studies you are referring to should be discussed.

**Response:**

Thank you for your comment. The original lines 131-132 have been deleted. We added the effect of land-use types on VOC concentrations.

**Comment 12:** Major comment: Different instruments, as well as analytical methods, were used for the analysis of VOCs at the three sites. How did the authors reconcile the data so that they can be compared? What is TD300 (line 177) that is not defined? In general, the small molecule and large VOCs are detected by FID and MSD, separately. Lines 182-184: What are the accuracies and detection limits for the minority of the species, i.e., those beyond the "95% and most VOC components", and what is the range of the precision? Line 188: What is the SEAS site? It has never been defined before.

**Response:**

Thank you for your comment. The three instruments (GC5000, GC580+TD300, GC866) all can be used to analyze VOCs. Their actual attainments were very similar in practice.

By using GC580+TD300, all PAMS substances meet the standard "curve correlation coefficient ≥ 0.995", all substances meet the standard "precision ≤ 10%", more than 95% of the target compounds meet the standard "accuracy ≤ ± 20%", all target compounds meet the standard "detection limit ≤ 0.15ppb", and more than 90% of the target compounds have blank response less than 0.1 ppb.

By using GC 5000 BTX/VOC, more than 90% of PAMS substances meet the standard "curve correlation coefficient ≥ 0.995", all substances meet the standard "precision ≤ 10%", more than 95% of the target compounds meet the standard "accuracy ≤ ± 20%", more than 98% of the target compounds meet the standard "detection limit ≤ 0.15ppb", and more than 95% of the target compounds have blank response less than 0.1 ppb.

By using GC866, more than 95% of PAMS substances meet the standard "curve correlation coefficient ≥ 0.995", all substances meet the standard "precision ≤ 10%", more than 95% of the target compounds meet the standard "accuracy ≤ ± 20%", more than 90% of the target compounds meet the standard "detection limit ≤ 0.15ppb", and more than 90% of the target compounds have blank response less than 0.1 ppb.

Because VOCs in three sites were collected and analyzed separately, we used the most suitable detection instruments for the three stations.

Lines 138-139:

> "At the PD site, VOCs was measured by gas chromatography (GC580-FID, PE, USA) and TD300 (a transformer driver)."

Lines 150-153:

> "The meteorological variables including temperature, RH and wind speed were simultaneously acquired from a weather station about 10 km northwest of the Shanghai Academy of Environmental Sciences."

**Comment 13:** Description about this method in section 2.3 is confusing. How can the spatial heterogeneity be determined for a single site, as stated in line 190? Line 193: Does j represent site or dataset? Contradictory descriptions.

**Response:**

Thank you for your comment. We revised the sentence in the new manuscript.

Lines 154-155:

*"The spatial heterogeneity of VOC concentration between two different sites was determined by the coefficient of divergence (COD) (Wongphatarakul et al., 1998; Sawvel et al., 2015)."*

Line 157:

*"where $x_{ij}$ presents the mass concentration in i time, j and k are two datasets, p presents the number of observations."*

**Comment 14:** Lines 202- 203: the descriptions of $g_{ik}$, $f_{kj}$ and $e_{ij}$ are totally wrong. Lines 204-207: The function Q is introduced. However, it is not clear what the purpose of introducing it is and how the authors used it? Lines 210-212: "EF is the error faction and can be set to 0.05-0.2" -What was the EF the authors set in this study? How did the authors determine the solution with seven factors as the optimal one?

**Response:**

Thank you for your comment. We revised the sentence in the new manuscript.

Lines 166-167:

*"$g_{ik}$ represents the species contribution of the kth source to the ith sample, $f_{kj}$ is the jth species fraction from the kth source, $e_{ij}$ is the residual result for jth species in ith sample."*

Lines 174-180:

*"where MDL is the minimum detection limit, EF is the error fraction and can be set to 0.05-0.2 (Song et al., 2007). It was 0.1 in this study. In this study, four to eleven factors were utilized to determine the option solution. $Q_{true}/Q_{robust}$ and and $Q_{true}/Q_{expected}$ are important parameters for characterizing the rationality of the PMF results (Brown et al., 2015). Seven factors were regarded as the optimal solution, comparing the ratios of $Q_{true}/Q_{robust}$, $Q_{true}/Q_{expected}$ and the PMF results. The $Q_{true}/Q_{robust}$ values were set to 1.0 at the three sampling sites. The $Q_{true}/Q_{expected}$ values were 1.3, 1.1, and 1.0 at the JS, PD and QP sites, respectively."*

**Comment 15:** Lines 223-225: How did you determine the number of polluted and all trajectories in a grid, and how was the weight function $W_{ij}$ applied?

**Response:**

Thank you for your comment. In this study, the pollution trajectory was defined as the trajectories corresponding to the total VOC (TVOC) concentration that exceeded the 75th percentile concentration of TVOCs. The $m_{ij}$ is the number of endpoints of the pollution trajectory passing through the grid $(i, j)$, and $n_{ij}$ is the number of endpoints of all the trajectories falling within the grid $(i, j)$. The weight function $W_{ij}$ was used to increase the model accuracy.

*"Therefore, the $PSCF_{ij}$ can be calculated using the Eq. (7) as follows:*

$$PSCF_{ij} = \frac{m_{ij}}{n_{ij}} \times W_{ij} = \frac{m_{ij}}{n_{ij}} \times \begin{cases} 1.00 & 80 < n_{ij} \\ 0.70 & 20 < n_{ij} \leq 80 \\ 0.42 & 10 < n_{ij} \leq 20 \\ 0.05 & n_{ij} \leq 10 \end{cases} \qquad (7)\text{"}$$

**Comment 16:** The equation (9) calculates the responses of VOC to $O_3$, which is opposite to the statement in lines 245-246 that "The characteristic structure and reactivity could influence the contribution of VOCs to $O_3$ formation". Lines 249-252: The rationality of using 100 as a threshold of background $O_3$ should be justified. Why 100? Note that it is a quite high value, especially in cool seasons. It is also totally wrong to assign the VOC concentrations during the $O_3$-background time period as background VOC concentrations. In most cases, the patterns of $O_3$ and VOCs are inconsistent. For example, $O_3$ got lowest values at night and in early morning when VOCs are at high levels. Lines 251-252: What is the logic behind? Why is VOCs influenced by the variation of $O_3$, and not the other way around? Lines 254-255: The logarithmic conversion is also problematic. Equation 12 should be written as $\ln y = \ln a + b \ln x$.

**Response:**

Thank you for your comment. The calculation of VOC-$O_3$ sensitivity (VOCs-$S_{O3}$) was deleted after consideration.

The equation 11 was rewritten.

"$\ln y = \ln a + b \cdot \ln x$"

**Comment 17:** Inaccurate expressions and grammatical errors in this section include but are not limited to the followings. Lines 180-181: "The samples were condensed low-carbon ($C_2$-$C_6$) compounds and high-carbon ($C_6$-$C_{12}$) compounds …"; Line 185: "trace instruments"; Lines 211-212: "option solution", "greatest solution"; Line 214: "observe the back trajectories, source and direction of pollutants"; Line 218: "This study was determined the 24-h back trajectory".

**Response:**

Thank you for your comment. We have revised the statement in the new manuscript.

For example, lines 142-143:

> "*The samples were condensed for low-carbon ($C_2$-$C_5$) compounds at 15℃ and high-carbon ($C_6$-$C_{12}$) compounds at 30℃.*"

Lines 148-149:

> "*The $O_3$, $NO$-$NO_2$-$NO_x$ were characterized by trace gas instruments (49i ozone analyzer and 42i nitrogen oxide analyzer, produced by Thermo Environmental Instruments Inc., USA) with the detection limits of 0.50 and 0.40 ppb, respectively.*"

Lines 175-177:

> "*In this study, four to eleven factors were utilized to determine the option solution. $Q_{true}/Q_{robust}$ and and $Q_{true}/Q_{expected}$ are important parameters for characterizing the rationality of the PMF results (Brown et al., 2015). Seven factors were regarded as the optimal solution, comparing the ratios of $Q_{true}/Q_{robust}$, $Q_{true}/Q_{expected}$ and PMF results.*"

Lines 182-184:

> "*PSCF and Cluster were widely used to determine the back trajectories, source and direction of pollutants (Draxier and Hess, 1998; Hong et al., 2019; Liu et al., 2019), and designed to measure the potential VOC source and primary transport pathway of trace elements (Ashbaugh et al., 1985; Xie et al., 2007; Zheng et al., 2018; Liu et al., 2020).*"

Lines 185-186:

*"This study was determined by the 24-h back trajectories (one hour interval) at the height of 500 m via the MeteoInfoMap software."*

**Comment 18:** Line 266: "60 VOC species" is contradictory to the statement that "Totally 43 species of VOCs were observed" in line 183.

**Response:**

Thank you for your comments. We have revised the statement in the new manuscript.

Lines 263-265:

*"During the observation campaign, 43 VOC species including 16 alkanes, 11 alkenes, 16 aromatics and 1 alkyne were measured and the contributions of total VOCs (TVOCs) > 1 % were marked."*

**Comment 19:** Lines 271-273: What's the point of comparing the wind speed that is very spatially uneven?

**Response:**

Thank you for your comment. We want to highlight that the wind speed in our study is higher than those in the other studies. We revised the sentence in the new manuscript.

Lines 238-240:

*"The wind speed at the QP site (4.37 ± 1.47 m s$^{-1}$) was 2.29 and 1.36 times higher than those at the JS (1.91 ± 0.49 m s$^{-1}$) and PD (1.30 ± 0.62 m s$^{-1}$) sites, respectively, indicating the decreased dilution and diffusion conditions at the latter two sites."*

**Comment 20:** Lines 274-288: The comparisons are rather simple. Are there same number of species, same species, same sampling season and etc.? Without discussion on these factors, the comparisons are meaningless.

**Response:**

Thank you for your comment. We rewrote the description in the revised manuscript.

*"Compared with the relevant measurements performed previously in Shanghai at the same sampling sites, this study generally presented lower VOC concentrations (Cai et al., 2010b; Zhang et al., 2018; Zhang et al., 2020a). In*

*detail, at the JS site, the VOC concentration was approximately 4 times lower than the measurement of Zhang et al., (2018) (94.14 ppb). At the PD and QP sites, the results in this study were slightly lower than those reported by Cai et al. (2010b) (24.3 ppb) and Zhang et al. (2020a) (15.41 ppb). A variety of control strategies, such as prohibiting of fireworks in the open air, improving VOC detection standards and strengthening control technology were implemented, thus resulting in the low VOC concentrations herein. Particularly, the policy of "one factory, one strategy", targeted at mitigating VOC emissions, was published by Shanghai government in 2018."*

**Comment 21:** Lines 295-312: I do not see the necessity of discussing such simple facts with too many words.

**Response:**

Thank you for your comment. The original lines 295-312 have been deleted.

**Comment 22:** I am surprised to see such high levels of $O_3$ in the sampling period. Without any doubt, the authors made mistakes in calculation or unit conversion.

**Response:**

Thank you for your comment. We revised the unit in the new manuscript.

Lines 249-251:

*"During the observation period, the average $PM_{2.5}$ values were 45.57 ± 27.59, 48.51 ± 27.22 and 40.27 ± 27.78 µg m$^{-3}$, and the mean $O_3$ concentrations were averaged to be 73.59 ± 23.59, 57.48 ± 20.49 and 99.30 ± 24.00 µg m$^{-3}$ at the JS, PD and QP sites, respectively."*

**Comment 23:** Lines 315-317: Readers would have no idea what the point of this discussion is. Are the dates special?

**Response:**

Thank you for your comment. We wanted to illustrate when the minimum hourly $PM_{2.5}$ levels occurred, and highlight that VOCs was positively correlated with $PM_{2.5}$ while was negatively correlated with $O_3$. The original lines 315-317 have been deleted.

**Comment 24:** Lines 317-318: I do not think this was the reason for the correlation. Otherwise, did you see correlation between VOCs and $O_3$, where the former was also precursors of the later?

**Response:**

Thank you for your comment. We revised the sentence in the new manuscript.

Lines 269-270:

*"VOCs was found to be positively correlated with $PM_{2.5}$, and the pearson correlation coefficients ($R_{Pearson}$) were 0.58, 0.71 and 0.25 at the JS, PD and QP sites, respectively."*

**Comment 25:** Lines 320-324: Why not refer to sources of $PM_{2.5}$ and VOCs in Shanghai. Transportation as the main source of $PM_{2.5}$ and VOCs in different cities does not necessarily mean the homology $PM_{2.5}$ and VOCs in Shanghai.

**Response:**

Thank you for your comment. We added the references documenting the sources of $PM_{2.5}$ and VOCs in Shanghai.

Lines 272-275:

*"Moreover, $PM_{2.5}$ and VOCs present similar emission sources. For example, traffic exhaust was proven as the predominant contributor for both of them (Li et al., 2009; Cai et al., 2010a; Cai et al., 2010b; Wang et al., 2013; Kuo et al., 2014; Liu et al., 2019)."*

**Comment 26:** Lines 325-331: First, I do not think the correlation is worth discussing. In most cases, the diurnal patterns of VOCs and $O_3$ are opposite. Second, the opposite patterns are mainly due to inconsistent patterns of VOC emission (e.g., emissions in morning and evening rush hours) and $O_3$ formation (e.g., daytime). The discussions are far-fetched and I do not understand "and counteraction was imposed by uncertain factors during the formation of $O_3$."

**Response:**

Thank you for your comment. We rewrote the description in the revised manuscript.

*"However, the VOC concentrations were negatively correlated with $O_3$ ($R_{Pearson}$ = -0.24 at the JS site, $R_{Pearson}$ = -0.48 at the PD site and $R_{Pearson}$ = -0.25 at the QP*

*site, respectively). The termination and titration (NO + O$_3$ → NO$_2$ + O$_2$) were more efficient and lots of factors including sunshine duration, temperature and relative humidity rather than the emission of precursors, impacted on the surface O$_3$. Li et al. (2019b) emphasized that the absolute concentration of precursor was not the only factor during the O$_3$ formation in Zhengzhou, China."*

The other sections are revised in the new manuscript. Please review our revised manuscript. We greatly appreciate any comments and valuable suggestions from the reviewer. Thank you for your time in handling our manuscript.

- - - - - - - - - - - - - - - - - - - - - - - - - - - - - - - - - - - - - - - - - - - - - -

**Lastly, we would again express our appreciation to the reviewers and editor for their warmhearted help. Thank you very much!**

- - - - - - - - - - - - - - - - - - - - - - - - - - - - - - - - - - - - - - - - - - - - - -

**References**

Chen, C., Xia, Z. H., Wu, M. M., Zhang, Q. Q., Wnag, T., and Wang, L. P.: Concentrations, Source Identification, and Lung Cancer Risk Associated with Springtime PM$_{2.5}$-Bound Polycyclic Aromatic Hydrocarbons (PAHs) in Nanjing, China, Arch. Environ. Contam. Toxicol., 73, 391-400, https://doi.org/10.1007/s00244-017-0435-4, 2017.

Jookjantra, P., Thepanondh, S., Keawboonchu, J., Kultan, V., and Laowagul, W.: Formation potential and source contribution of secondary organic aerosol from volatile organic compounds, J. Environ. Qual., 1-40, https://doi.org/10.1002/jeq2.20381, 2022.

Kumar, A., Singh, D., Kumar, K. S., Singh, B. B., and Jain, V. K.: Distribution of VOCs in urban and rural atmospheres of subtropical India: Temporal variation, source attribution, ratios, OFP and risk assessment, Sci. Total Environ., 613-614, 492-501, https://doi.org/10.1016/j.scitotenv.2017.09.096, 2018.

Li, Y. J., Ren, B. N., Qiao, Z., Zhu, J. P., Wang, H. L., Zhou, M., Qiao, L. P., Lou, S. R., Jing, S. G., Huang, C., Tao, S. K., Rao, P. H., and Li, J.: Characteristics of atmospheric intermediate volatility organic compounds (IVOCs) in winter and summer under different air pollution levels, Atmos. Environ., 210, 58-65, https://doi.org/10.1016/j.atmosenv.2019.04.041, 2019.

Liu, J., Chu, B. W., Jia, Y. C., Cao, Q., Zhang, H., Chen, T. Z., Ma, Q. X., Ma, J. Z., Wang, Y. H., Zhang, P., and Hong, H.: Dramatic decrease of secondary organic aerosol formation potential in Beijing: Important contribution from reduction of coal combustion emission, Sci. Total Environ., 832, 155045, http://dx.doi.org/10.1016/j.scitotenv.2022.155045, 2022.

Song, M. D., Li, X., Yang, S. D., Yu, X. N., Zhou, S. X., Yang, Y. M., Chen, S. Y., Dong, H. B., Liao, K. R., Chen, Q., Lu, K. D., Zhang, N. N., Cao, J. J., Zeng, L. M., and Zhang, Y. H.: Spatiotemporal variation, sources, and secondary transformation potential of volatile organic compounds in Xi'an China, Atmos. Chem. Phys., 21, 4939-4958, https://doi.org/10.5194/acp-21-4939-2021, 2021.

Tang, J. H., Chan, L. Y., Chan, C. Y., Li, Y. S., Chang, C. C., Wang, X. M., and Wu, D.: Implications of changing urban and rural emissions on non-methane hydrocarbons in the Pearl River Delta

region of China. Atmos. Environ., 42 (16), 3780-3794, https://doi.org/10.1016/j.atmosenv.2007.12.069, 2008.

Wang, R., Xu, X. B., Jia, S. H., Ma, R. S., Ran, L., Deng, Z. Z., Lin, W. L., Wang, Y., and Ma, Z. Q: Lower tropospheric distributions of $O_3$ and aerosol over Raoyang, a rural site in the North China Plain, Atmos. Chem. Phys., 17, 3891-3903, https://doi.org/10.5194/acp-17-3891-2017, 2017.

Yoo, J. M., Jeong, M. J., Kim, D., Stockwell, W. R., Yang, J. H., Shin, H. W., Lee, M. I., Song, C. K., and Lee, S. D.: Spatiotemporal variations of air pollutants ($O_3$, $NO_2$, $SO_2$, CO, $PM_{10}$, and VOCs) with land-use types, Atmos. Chem. Phys., 15, 10857-10885, https://doi.org/10.5194/acp-15-10857-2015, 2015.

Zhan, J. L., Feng, Z. M., Liu, P. F., He, X. W., He, Z. M., Chen, T. Z., Wang, Y. F., He, H., Mu, Y. J., and Liu, Y. C.: Ozone and SOA formation potential based on photochemical loss of VOCs during the Beijing summer, Environ. Pollut., 285, 117444, https://doi.org/10.1016/j.envpol.2021.117444, 2021.

Zhang, Y. C., Li R., Fu H. B., Zhou D., and Chen J. M.: Observation and analysis of atmospheric volatile organic compounds in a typical petrochemical area in Yangtze River Delta, China, J. Environ. Sci., 71, 233-248, https://doi.org/CNKI:SUN:HJKB.0.2018-09-022, 2018.

---

## Author Comment (AC2)

**Response to the Comments of the Reviewers**

Dear Editor and Reviewers,

We acknowledge the comments and encouragement of two reviewers, and are also grateful to the efficient serving of the editor. Here we submit our revised manuscript **"Measurement report: VOC characteristics at different land-use types in Shanghai: spatio-temporal variation, source apportionment, and impact on secondary formations of ozone and aerosol" (Manuscript number: acp-2022-250)**, as well as a thorough, point-by-point response to each point raised from the reviewers. The revisions to the manuscript are highlighted in blue words in the provided "Response to the Comments of the Reviewers". Additionally, there is a clean revised manuscript as required.

We greatly appreciate those comments and valuable suggestions from the reviewers. The manuscript has been greatly improved. We do feel that we have demonstrated our efforts in the revised manuscript.

Yours sincerely,

Hongbo Fu and Co-authors.

Department of Environmental Science & Engineering

Fudan University

200000 Shanghai China

E-mail: fuhb@fudan.edu.cn

**Response to the Reviewer #2**

**General Comments:**

The manuscript presents a study based on the concurrent observations of volatile organic compounds (VOCs) at three supersites sites in Shanghai. The characteristics of VOCs, ozone formation potential, secondary organic aerosol formation potential and emission sources are discussed. The influence of different land use type on VOC profiles and atmospheric oxidation capacity is worthy of study. However, the discussion in this manuscript is not enough to highlight this viewpoint. The author should focus more on the discussion of the influence of land use type on VOC profile and atmospheric oxidation capacity. Moreover, I find irrationality of the methods and inaccuracy of some conclusion. Also, the grammar needs to be thoroughly revised. Generally, under its current version, this paper needs substantial revision to reach the standard for publication. However, the scope of this manuscript is good and the measurements can provide deeper understanding of the influence of anthropogenic activities on megacities or city clusters. A revised edition is encouraged for resubmission.

**Response:** We would like to thank reviewer #2 for your carefully review and valuable and constructive comments. Promoted by your suggestions, we have made significant efforts to improve the logicality and readability of the paper. The details can be seen in the revised manuscript.

**Comment 1:** As the author have stated in Line 110-113: "limited knowledge is available on the multi-site research at a city level .....", authors should well explain how does multi-site observation bring us new insights different from single site observations. Furthermore, to my knowledge, there are a lot of studies about VOC characteristics in Shanghai during different measurement period, at different locations and at different years. The authors should also elucidate how this work bring new insights different from just comparing the reported results. Finally, the authors should rearrange the whole manuscript emphasizing on the difference of the observation results of three sites and the discussion about the influence of different land use type.

**Response:**

Thank you for your comment. We rewrote the description in the revised manuscript. Lines 35-36:

> *"The findings here provide more information on the accurate air-quality control at a city level in China."*

The point in this study is to analyze the influence of land-use types on VOC concentrations, sources and ozone and SOA formation potentials in Shanghai. In detail, the long-term VOC emission inventory highlighted that the VOC emission varied with the land-use type (Li et al., 2019). The observation campaign also showed that VOC concentrations were largely influenced by the land-use type (Tang et al., 2008; Kumar et al., 2018; Zhang et al., 2018). Besides, the land-use types not only influence the VOC concentrations but also the sources especially the anthropogenic sources (Yoo et al., 2015; Chen et al., 2017; Wang et al., 2017; Jookjantra et al., 2022). Additionally, the diversities of VOC concentrations among the different land-use types could affect the ozone and SOA formation potentials (OFP and SOAFP), resulting in the variations of $O_3$ and SOA concentrations (Song et al., 2021; Zhan et al., 2021; Liu et al., 2022). Shanghai is regarded as an ideal area to perform atmospheric measurements with the different land-use types. However, many studies were mainly focused on the single-site measurements, particularly conducted at the urban site in Shanghai, resulting that the impact of land-use type on VOC characteristics is still unclear to date. Besides, they mainly concentrated on the $O_3$ characteristics, while the $O_3$ and SOA formations from VOCs and the relationship between VOCs together with $PM_{2.5}$ and $O_3$ were rarely analyzed. Given the factors mentioned above, in this study, the concurrent multiple-site and high time-resolution measurement of the VOCs with three typical land-use types in Shanghai for their characteristics, sources and ozone and SOA formation potentials was performed. The result therein could broaden the city-scale research from single-site measurement to multi-site observations, and simultaneously narrows the multi-site research from worldwide scale to city scale.

We added the discussion about the effects of land-use types on VOC concentrations, sources and ozone and SOA formation potentials in Sec. 4 (lines 515-650).

**Comment 2:** The instruments applied at three sites are different. I do not think the comparison is convincing, without any illustration about the data reconciliation. Since the main scope is to compare the VOC data observed at different sites, the data quality is of great importance to the final conclusion. The authors should discuss more about the detect of limit, accuracy of all measured VOC species.

**Response:**

Thank you for your comment. The three instruments (GC5000, GC580+TD300, GC866) all can be used to analysis VOCs. Their actual attainments were very similar in practice.

By using GC580+TD300, all PAMS substances meet the standard "curve correlation coefficient $\geq 0.995$", all substances meet the standard "precision $\leq 10\%$", more than 95% of the target compounds meet the standard "accuracy $\leq \pm 20\%$", all target compounds meet the standard "detection limit $\leq 0.15$ppb", and more than 90% of the target compounds have blank response less than 0.1 ppb.

By using GC 5000 BTX/VOC, more than 90% of PAMS substances meet the standard "curve correlation coefficient $\geq 0.995$", all substances meet the standard "precision $\leq 10\%$", more than 95% of the target compounds meet the standard "accuracy $\leq \pm 20\%$", more than 98% of the target compounds meet the standard "detection limit $\leq 0.15$ppb", and more than 95% of the target compounds have blank response less than 0.1 ppb.

By using GC866, more than 95% of PAMS substances meet the standard "curve correlation coefficient $\geq 0.995$", all substances meet the standard "precision $\leq 10\%$", more than 95% of the target compounds meet the standard "accuracy $\leq \pm 20\%$", more than 90% of the target compounds meet the standard "detection limit $\leq 0.15$ppb", and more than 90% of the target compounds have blank response less than 0.1 ppb.

Because VOCs in three sites were collected and analyzed separately, we used the most suitable detection instruments for the three stations.

**Comment 3:** The authors should discuss more on how to determine the final PMF solution before discussing the PMF results.

**Response:**

Thank you for your comment. We added the description in the revised manuscript.

Lines 174-180:

*"where MDL is the minimum detection limit, EF is the error fraction and can be set to 0.05-0.2 (Song et al., 2007). It was 0.1 in this study. In this study, four to eleven factors were utilized to determine the option solution. $Q_{true}/Q_{robust}$ and and $Q_{true}/Q_{expected}$ are important parameters for characterizing the rationality of the PMF results (Brown et al., 2015). Seven factors were regarded as the optimal solution, comparing the ratios of $Q_{true}/Q_{robust}$, $Q_{true}/Q_{expected}$ and PMF results. The $Q_{true}/Q_{robust}$ values were set to 1.0 at the three sampling sites. The $Q_{true}/Q_{expected}$ values were 1.3, 1.1, and 1.0 at the JS, PD and QP sites, respectively."*

**Comment 4:** The term" secondary formation potentials" in the title is confusing, it will be better to use "ozone and SOA formation potential".

**Response:**

Thank you for your comment. We revised the title in the new manuscript.

*"Measurement report: VOC characteristics at different land-use types in Shanghai: spatio-temporal variation, source apportionment, and impact on secondary formations of ozone and aerosol"*

**Comment 5:** Line 29-30: The sentence "The VOCs-$O_3$ sensitivity indicated that VOCs-$S_{O3}$ values varied at the different sites and were primarily controlled by the alkenerelated reactions". It is confusing at this place to see VOCs-$SO_3$ without illustrations of the methodologies. It will be better to just state the main collusion here.

**Response:**

Thank you for your comment. The calculation of VOC-$O_3$ sensitivity (VOCs-$S_{O3}$) was deleted after consideration.

**Comment 6:** Line 31-34: How the findings provide new insights into the accurate control of different land-use type? Moreover, how the results highlight the importance of multiple-site measurements?

**Response:**

Thank you for your comment. We rewrote the description in the revised manuscript.

Lines 35-36:

> *"The findings here provide more information on the accurate air-quality control at a city level in China."*

The results herein could provide scientific-based information for policymakers to establish targeted strategies of alleviating VOC pollution at the different land-use types. For example, the JS site exhibited higher fractions of aromatics and alkenes, particularly toluene and propylene, than those at the PD and QP sites. The VOC concentration in the early morning (5:00 LT) at the JS site was higher than those at the other two sites. This result did necessarily correlate with the fact that the JS site is close to the industrial area with heavy industrial emissions, suggesting that industrial activities were key factors of VOC pollution at the JS site. Moreover, the industrial emission and biogenic source showed slight contributions to VOC concentrations at the QP and JS/PD sites, respectively. It was consistent with the regional characteristics of anthrogopenic activities dominated by land-use types. Additionally, the results of VOCs-$S_{PM2.5}$ varied with the land-use types. The aromatics at the JS and PD sites, as well as alkanes at the QP site played crucial roles in the VOC-induced haze pollution. The relevant emission sources, which are thought to be the industrial production at the JS/PD sites and vehicle exhaust at the QP site, should be controlled in priority. Therefore, these findings could provide more information on the accurate VOCs control. The results shown herein highlight that the simultaneous multiple-site measurements with the different land-use type in the megacity or city cluster could be more appropriate to fully understand the VOC characteristics relative to a single-site measurement performed normally.

**Comment 7:** Line 35-139 I do not think it is necessary to put some many basic knowledge in the introduction part.

**Response:**

Thank you for your comment. We rewrote the introduction in the revised manuscript.

For example, lines 55-57:

> *"The long-term VOC emission inventory highlighted that there were significant spatial discrepancies of VOC emissions (Li et al., 2019a). The observation campaign also showed that VOC concentrations varied with the sampling sites.*

*These phenomena were attributed to the fact that VOC concentrations were closely correlated with the land-use types."*

Lines 63-65:

*"Besides, the land-use types not only influence the VOC concentrations but also the sources especially the anthropogenic sources (Yoo et al., 2015; Chen et al., 2017; Wang et al., 2017; Jookjantra et al., 2022)."*

Lines 72-73:

*"The diversities of VOC concentrations among the different land-use types could affect the ozone and SOA formation potentials (OFP and SOAFP), resulting in the variations of $O_3$ and SOA concentrations."*

Lines 77-78:

*"In detail, atmospheric VOCs undergoes degradation to produce oxidants ($HO_2$ and $RO_2$), which further oxidizes atmospheric NO, followed by producing $NO_2$ and the formation of $O_3$ finally via the photochemical pathways (Wang et al., 2017). "*

Lines 81-85:

*"As the key precursor of SOA, VOCs can be oxidized to produce the low VOCs, followed by the formation of SOA via homogeneous nucleation (Merikanto et al., 2009). Moreover, the partitioning of semi-volatile products from VOCs and oxidants gas-phase photochemical reactions to form SOA (Pankow, 1994; Lim et al., 2010). Additionally, low VOCs are produced via the aqueous-phase reactions in atmospheric waters e.g., clouds, fogs, and aerosol water which are largely retained in the particle-phase to generate SOA (Lim et al., 2010)."*

Lines 87-94:

*"The expanding urbanization and industrialization jointly aggravate the VOC pollution. Moreover, the $O_3$ concentration at the urban area in Shanghai increased by ~ 67 % from 2006 to 2015 with the growth rate of 1.1 ppbv pear year (Gao et al., 2017). The maximum 1-hour concentration of $O_3$ could exceed 380 $\mu g\ m^{-3}$ during polluted days (Shi et al., 2015; Gu et al., 2020). Such scenario suggested that $O_3$ played an important role in atmospheric pollution and*

*Shanghai was suffering from heavy $O_3$ pollution. Additionally, the large changes of land-use occurred in Shanghai due to the rapid development e.g., many cultivated areas became urban and/or industrial zones, resulting in the diverse land-use types (Tian et al., 2017). Therefore, Shanghai is regarded as an ideal area to perform atmospheric measurements with the different land-use types."*

**Comment 8:** Line 140-168: It will be much easier to compare the difference of three sites, if the authors can make a table about potential sources at different sites, and the results of published works conducted at the three sites (if there were).

**Response:**

Thank you for your comment. We added a table in the revised supporting information.

Table S1. The sources and land-use type at the sampling sites.

| Sites | Land-use type | Details | References |
|---|---|---|---|
| Jinshan Site (JS) | Industrial district | Site surrounded by chemical factories | Zhang et al., (2018) |
| Pudong Site (PD) | Residential and commercial mixed districts | Site surrounded by residences and administrative areas | Cai et al., (2010b) |
| Qingpu Site (QP) | Background district | Site surrounded by farmland and forests | Zhang et al., (2020) |

**Comment 9:** Line 177: what's TD300?

**Response:**

Thank you for your question. We added the illustration in the revised manuscript.

*"At the PD site, VOCs was measured by gas chromatography (GC580-FID, PE, USA) and TD300 (a transformer driver)."*

**Comment 10:** Line 181-183: $R^2$ of what? Calibration results? Please clarify. Odd expression about "accuracy of 95% of compounds". What about the accuracy of the left 5%?

**Response:**

Thank you for your question. $R^2$ is the curve correlation coefficient. We revised the sentence in the new manuscript.

Line 145:

> *"The curve correlation coefficient ($R^2$) of all of the VOCs were $\geq 0.995$."*

By using GC580+TD300, all PAMS substances meet the standard "curve correlation coefficient $\geq 0.995$", all substances meet the standard "precision $\leq 10\%$", more than 95% of the target compounds meet the standard "accuracy $\leq \pm 20\%$", all target compounds meet the standard "detection limit $\leq 0.15$ppb", and more than 90% of the target compounds have blank response less than 0.1 ppb.

By using GC 5000 BTX/VOC, more than 90% of PAMS substances meet the standard "curve correlation coefficient $\geq 0.995$", all substances meet the standard "precision $\leq 10\%$", more than 95% of the target compounds meet the standard "accuracy $\leq \pm 20\%$", more than 98% of the target compounds meet the standard "detection limit $\leq 0.15$ppb", and more than 95% of the target compounds have blank response less than 0.1 ppb.

By using GC866, more than 95% of PAMS substances meet the standard "curve correlation coefficient $\geq 0.995$", all substances meet the standard "precision $\leq 10\%$", more than 95% of the target compounds meet the standard "accuracy $\leq \pm 20\%$", more than 90% of the target compounds meet the standard "detection limit $\leq 0.15$ppb", and more than 90% of the target compounds have blank response less than 0.1 ppb.

**Comment 10:** what's SEAS site?

**Response:**

Thank you for your question. We added the illustration in the revised manuscript.

> *"The meteorological variables including temperature, RH and wind speed were simultaneously acquired from a weather station about 10 km northwest of the Shanghai Academy of Environmental Sciences."*

**Comment 11:** How can one site have spatial heterogeneity? Do you mean each pair of sites?

**Response:**

Thank you for your question. We revised the sentence in the new manuscript.

> *"The spatial heterogeneity of VOC concentration between two different sites was determined by the coefficient of divergence (COD) (Wongphatarakul et al., 1998; Sawvel et al., 2015)."*

**Comment 12:** Line 202: I think the authors need to rewrite the description about $g_{ik}$, $f_{kj}$, and $e_{ij}$.

**Response:**

Thank you for your comment. We rewrite the description in the revised manuscript.

*"$g_{ik}$ represents the species contribution of the kth source to the ith sample, $f_{kj}$ is the jth species fraction from the kth source, $e_{ij}$ is the residual result for jth species in ith sample."*

**Comment 13:** Line 205-212: The authors should explain how the Q value works in PMF model, for example, how it can help determining the PMF solution.

**Response:**

Thank you for your comment. We added the description in the revised manuscript.

*"where MDL is the minimum detection limit, EF is the error fraction and can be set to 0.05-0.2 (Song et al., 2007). It was 0.1 in this study. In this study, four to eleven factors were utilized to determine the option solution. $Q_{true}/Q_{robust}$ and and $Q_{true}/Q_{expected}$ are important parameters for characterizing the rationality of the PMF results (Brown et al., 2015). Seven factors were regarded as the optimal solution, comparing the ratios of $Q_{true}/Q_{robust}$, $Q_{true}/Q_{expected}$ and the PMF results. The $Q_{true}/Q_{robust}$ values were set to 1.0 at the three sampling sites. The $Q_{true}/Q_{expected}$ values were 1.3, 1.1, and 1.0 at the JS, PD and QP sites, respectively."*

**Comment 14:** How is weight function applied in PSCF?

**Response:**

Thank you for your comment. We added the description in the revised manuscript. In this study, the pollution trajectory was defined as the trajectories corresponding to the total VOC (TVOC) concentration that exceeded the 75th percentile concentration of TVOCs. The $m_{ij}$ is the number of endpoints of the pollution trajectory passing through the grid $(i, j)$, and $n_{ij}$ is the number of endpoints of all the trajectories falling within the grid $(i, j)$. The weight function $W_{ij}$ was used to increase the accuracy of the model.

*"Therefore, the $PSCF_{ij}$ can be calculated using the Eq. (7) as follows:*

$$PSCF_{ij} = \frac{m_{ij}}{n_{ij}} \times W_{ij} = \frac{m_{ij}}{n_{ij}} \times \begin{cases} 1.00 & 80 < n_{ij} \\ 0.70 & 20 < n_{ij} \leq 80 \\ 0.42 & 10 < n_{ij} \leq 20 \\ 0.05 & n_{ij} \leq 10 \end{cases} \qquad (7)"$$

**Comment 15:** Please confirm exactly how many VOC species were observed, 60 or 43? It will be better to list the observed VOC species in a table.

**Response:**

Thank you for your comments. We have revised the statement in the new manuscript. We list the observed VOC species in the Tab. S2.

*"During the observation campaign, 43 VOC species including 16 alkanes, 11 alkenes, 16 aromatics and 1 alkyne were measured and the contributions of total VOCs (TVOCs) > 1 % were marked."*

**Comment 16:** Line 277-284: How does the comparison meaningful, without clarifying the observed VOC species?

**Response:**

Thank you for your comment. We rewrote the description in the revised manuscript.

*"Compared with the relevant measurements performed previously in Shanghai at the same sampling sites, this study generally presented lower VOC concentrations (Cai et al., 2010b; Zhang et al., 2018; Zhang et al., 2020a). In detail, at the JS site, the VOC concentration was approximately 4 times lower than the measurement of Zhang et al., (2018) (94.14 ppb). At the PD and QP sites, the results in this study were slightly lower than those reported by Cai et al. (2010b) (24.3 ppb) and Zhang et al. (2020a) (15.41 ppb). A variety of control strategies, such as prohibiting of fireworks in the open air, improving VOC detection standards and strengthening control technology were implemented, thus resulting in the low VOC concentrations herein. Particularly, the policy of "one factory, one strategy", targeted at mitigating VOC emissions, was published by Shanghai government in 2018."*

**Comment 17:** Lin310-312: "This phenomenon was because there similar VOC emission intensity ….." . The discussion about different COD is too simple.

**Response:**

Thank you for your comment. We added the illustration in the revised manuscript.

*"Note that the distinct spatial heterogeneity of VOCs was also observed with the highest value of the coefficient of divergence (COD = 0.36) between the JS and QP sites, followed by the PD and QP sites (COD = 0.33), with that between the JS and PD sites (COD = 0.20) being the lowest. Hence, the spatial heterogeneity of VOCs between the JS and PD sites was narrow, while the QP site was largely different from other two sites. This result was ascribed to the fact that there were similar pollutant concentrations, meteorological factors, emission intensities and atmospheric conditions at the JS and PD sites, while these indexes in the QP site were rather different from those of the other sites.*

**Comment 18:** Line 317-318: "Statistically, VOCs were found to be positively correlated with $PM_{2.5}$ due to the fact that VOCs were a significant precursor of $PM_{2.5}$." I do not think this explain is correct.

**Response:**

Thank you for your comment. We rewrote the description in the revised manuscript.
Lines 251-252:

*"VOCs was found to be positively correlated with $PM_{2.5}$, and the pearson correlation coefficients ($R_{Pearson}$) were 0.58, 0.71 and 0.25 at the JS, PD and QP sites, respectively."*

**Comment 19:** Line 324-331: What's the view point the authors want to discuss? Determining the controlling factor of $O_3$ formation merely based on the ratio of VOCs/NOx ratio is too simple. Moreover, in Line 329 "a higher proportion of OH radical reacted with $NO_2$ to suppress the $O_3$ formation", how the authors get this conclusion from the VOCs/$NO_x$ ratio?

**Response:**

We want to highlight the progress of $O_3$ formation and why the VOCs was negatively correlated with the $O_3$. We rewrote the description in the revised manuscript.

*"However, the VOC concentrations were negatively correlated with $O_3$. The termination and titration ($NO + O_3 \rightarrow NO_2 + O_2$) were more efficient and lots of*

*factors, rather than the emission of precursors, impacted on the surface O₃. Li et*

*al. (2019b) emphasized that the absolute concentration of precursor was not the*

*only factor during the O₃ formation in Zhengzhou, China."*

**Comment 20:** The authors have highlighted some differences among three sites. Such

as in Line 356-357, "the contribution of toluene at JS site was markedly increased (~

3times) relative to the other two sites." However, such discussion about the difference

of VOC markers is insufficient in this section. The authors should discuss more about

such difference and make connection of the observed difference with the difference

of land use type or emission sources.

**Response:**

Thank you for your comment. We added the discussion about the effects of land-use

types on VOC concentrations, sources and O₃ and SOA formation potentials in the

Sec. 4.

For example, lines 527-528:

*"The mean VOC concentrations at the JS (21.88 ± 12.58 ppb) and PD (21.36 ±*

*8.58 ppb) sites were 1.83 and 1.79 times higher than that at the QP site (11.93 ±*

*6.33 ppb), implicating the impact of land-use type."*

Lines 534:

*"Besides, the land-use types could not only affect the VOC concentrations but*

*also compositions."*

Lines 553-555:

*"The VOC diurnal variations were analyzed with respect to each of the land-use*

*types. The result showed that the VOC concentration in the early morning (5:00*

*LT) at the JS site was 17.06 % and 52.91 % higher than those at the other two*

*sites, which was attributed mostly to the land-use types."*

Lines 560-561:

*"In addition to the diurnal variations, the "weekend effects" of VOCs also*

*appeared to be variable among the different land-use types."*

Lines 570:

*"VOC sources in this study were sensitive to the local emission with the different land-use types."*

Lines 612:

*"The OFP values were closely related with the land-use types."*

Lines 623-626:

*"The discrepancies of SOAFP values among the sampling sites were observed since they could be significantly influenced by the land-use types. At the JS site, the SOAFP value (1.00 ± 2.03 µg m⁻³) was 2.17 and 2.44 times higher than those at the PD (0.46 ± 0.88 µg m⁻³) and QP (0.41 ± 0.58 µg m⁻³) sites, which was consistent with variations of VOC concentrations especially aromatics and connected with the land-use types (Zhang et al., 2017; Jookjantra et al., 2022)."*

**Comment 21:** Line 369-406: the authors discuss too much about the diurnal variation pattern, which is similar with the reported diurnal characteristics in many other studies. There are some interesting parts such as at Line 390-391: "The VOC concentrations on the weekends were 3.31, 10.19 and 1.19% lower than those on the weekdays," and at Line 394-395: "It should be noted that there were narrow discrepancies of VOC concentrations at the site between the weekdays and weekends ….". The difference at weekends and workdays can be attributed to the influence of the local emission sources. The author should rearrange this section focusing on the discussion about the difference of weekend/ holiday effects.

**Response:**

Thank you for your comment. We rewrote the description in the revised manuscript, specified in newly lines 301-320.

**Comment 22:** Line 415-416: how is industrialization and urbanization results in the stagnant weather condition?

**Response:**

Thank you for your question. We revised this sentence as:

*"Such scenario could be attributed to the locations of JS and PD sites which implicated stagnant weather conditions and high anthropogenic emissions, therefore inducing the severe haze pollution."*

**Comment 23:** Line 430: How about the "clean-haze" discrepancy of other VOC species?

**Response:**

Thank you for your question. At the JS site, the "clean-haze" discrepancies were 1.38, 1.51, 1.63 and 1.02 of the alkanes, alkenes, aromatics and alkyne, respectively. At the PD site, the "clean-haze" discrepancies were 1.39, 1.58, 1.83 and 1.63 of the alkanes, alkenes, aromatics and alkyne, respectively. At the QP site, the "clean-haze" discrepancies were 1.37, 1.24, 1.35 and 1.32 of the alkanes, alkenes, aromatics and alkyne, respectively.

**Comment 24:** Line 36: "...both of which formation are ..." better to use two sentences here.

**Response:**

Thank you for your comment. We rewrote the sentence in the revised manuscript.

*"Serious air pollution in China is currently characterized by the high levels of ozone ($O_3$) and fine particulate matters (PM) especially $PM_{2.5}$ (PM with an aerodynamic diameter less than 2.5 μm). The atmospheric volatile organic compounds (VOCs) greatly influence the $O_3$ and $PM_{2.5}$ formations, and function as the important precursors (Carter, 1994; Liu et al., 2008; Yuan et al., 2013; Lu et al., 2018; Ma et al., 2019; Yu et al., 2021)."*

**Comment 25:** Line 180-181: "The samples were condensed low-carbon ($C_2$-$C_6$) compounds and high-carbon ($C_6$-$C_{12}$) compounds ...". Rearrange the sentence.

**Response:**

Thank you for your comment. We rewrote the sentence in the revised manuscript.

*"The samples were condensed for low-carbon ($C_2$-$C_5$) compounds at 15℃ and high-carbon ($C_6$-$C_{12}$) compounds at 30℃."*

**Comment 26:** Line 185: what is trace instruments?

**Response:**

Thank you for your question. It should be the "trace gas instruments". We rewrote the description in the revised manuscript.

*"The $O_3$, $NO$-$NO_2$-$NO_x$ were characterized by trace gas instruments (49i ozone analyzer and 42i nitrogen oxide analyzer, produced by Thermo Environmental Instruments Inc., USA) with the detection limits of 0.50 and 0.40 ppb, respectively."*

**Comment 27:** Line 211: "greatest solutions" should be optimal solutions.

**Response:**

Thank you for your comment. We rewrote the sentence in the revised manuscript.

*"Seven factors were regarded as the optimal solution, comparing the ratios of $Q_{true}/Q_{robust}$, $Q_{true}/Q_{expected}$ and PMF results."*

**Comment 28:** Line 218: Should be "This study was determined by the 24-h back trajectory

**Response:**

Thank you for your comment. We rewrote the sentence in the revised manuscript.

*"This study was determined by the 24-h back trajectories (one hour interval) at the height of 500 m via the MeteoInfoMap software."*

**Comment 29:** Line 266-267: "The temperatures were averaged to be .......

**Response:**

Thank you for your comment. We rewrote the sentence in the revised manuscript.

*"The average temperatures were 8.69 ± 3.24, 9.02 ± 3.24 and 7.73 ± 2.92 ℃, and the mean RHs were 83.77 ± 11.38, 75.37 ± 13.29 and 71.80 ± 9.28 % at the JS, PD and QP sites, respectively."*

**Comment 30:** Line 300: "a large number of organizations...". Improper use of organizations here.

**Response:**

Thank you for your comment. This sentence has been deleted.

**Comment 31:** Line 304: "The reduced VOC concentrations coincide...". This sentence is confusing.

**Response:**

Thank you for your comment. The original line 304 has been deleted.

**Comment 32:** Line 356-357: "the contribution of toluene at JS site was markedly increased (~ 3times) relative to the other two sites." Should be higher instead of increased.

**Response:**

Thank you for your comment. We rewrote the sentence in the revised manuscript.

Lines 533-534:

*"Specifically, the JS site exhibited higher fractions of aromatics and alkenes, particularly toluene and propylene, than those at the PD and QP sites."*

**Comment 33:** Line 375: Should be "the VOC concentrations also tended to increase"

**Response:**

Thank you for your comment. We rewrote the sentence in the revised manuscript.

*"During the rush-hour traffic at 18:00 to 21:00 LT, the VOC concentrations also tended to increase, and the evening peak values were 18.46, 20.82 and 10.22 ppb at the JS, PD and QP sites, respectively."*

**Comment 34:** Line 424: should be "At the QP site".

**Response:**

Thank you for your comment. We rewrote the sentence in the revised manuscript.

*"At the QP site, alkanes (2, 2, 4-trimethylpentane, n-hexane, n-heptane) presented significant 'clean-haze' discrepancy (~ 36.58 % uplift), implying the great influences of vehicle exhaust and fuel evaporation."*
* * *
**Lastly, we would again express our appreciation to the reviewers and editor for their warmhearted help. Thank you very much!**
* * *
**References**

Chen, C., Xia, Z. H., Wu, M. M., Zhang, Q. Q., Wnag, T., and Wang, L. P.: Concentrations, Source Identification, and Lung Cancer Risk Associated with Springtime PM$_{2.5}$-Bound Polycyclic Aromatic Hydrocarbons (PAHs) in Nanjing, China, Arch. Environ. Contam. Toxicol., 73, 391-400, https://doi.org/10.1007/s00244-017-0435-4, 2017.

Jookjantra, P., Thepanondh, S., Keawboonchu, J., Kultan, V., and Laowagul, W.: Formation potential and source contribution of secondary organic aerosol from volatile organic compounds, J. Environ. Qual., 1-40, https://doi.org/10.1002/jeq2.20381, 2022.

Kumar, A., Singh, D., Kumar, K. S., Singh, B. B., and Jain, V. K.: Distribution of VOCs in urban and rural atmospheres of subtropical India: Temporal variation, source attribution, ratios, OFP and risk

assessment, Sci. Total Environ., 613-614, 492-501, https://doi.org/10.1016/j.scitotenv.2017.09.096, 2018.

Li, Y. J., Ren, B. N., Qiao, Z., Zhu, J. P., Wang, H. L., Zhou, M., Qiao, L. P., Lou, S. R., Jing, S. G., Huang, C., Tao, S. K., Rao, P. H., and Li, J.: Characteristics of atmospheric intermediate volatility organic compounds (IVOCs) in winter and summer under different air pollution levels, Atmos. Environ., 210, 58-65, https://doi.org/10.1016/j.atmosenv.2019.04.041, 2019.

Liu, J., Chu, B. W., Jia, Y. C., Cao, Q., Zhang, H., Chen, T. Z., Ma, Q. X., Ma, J. Z., Wang, Y. H., Zhang, P., and Hong, H.: Dramatic decrease of secondary organic aerosol formation potential in Beijing: Important contribution from reduction of coal combustion emission, Sci. Total Environ., 832, 155045, http://dx.doi.org/10.1016/j.scitotenv.2022.155045, 2022.

Song, M. D., Li, X., Yang, S. D., Yu, X. N., Zhou, S. X., Yang, Y. M., Chen, S. Y., Dong, H. B., Liao, K. R., Chen, Q., Lu, K. D., Zhang, N. N., Cao, J. J., Zeng, L. M., and Zhang, Y. H.: Spatiotemporal variation, sources, and secondary transformation potential of volatile organic compounds in Xi'an China, Atmos. Chem. Phys., 21, 4939-4958, https://doi.org/10.5194/acp-21-4939-2021, 2021.

Tang, J. H., Chan, L. Y., Chan, C. Y., Li, Y. S., Chang, C. C., Wang, X. M., and Wu, D.: Implications of changing urban and rural emissions on non-methane hydrocarbons in the Pearl River Delta region of China. Atmos. Environ., 42 (16), 3780-3794, https://doi.org/10.1016/j.atmosenv.2007.12.069, 2008.

Wang, R., Xu, X. B., Jia, S. H., Ma, R. S., Ran, L., Deng, Z. Z., Lin, W. L., Wang, Y., and Ma, Z. Q: Lower tropospheric distributions of $O_3$ and aerosol over Raoyang, a rural site in the North China Plain, Atmos. Chem. Phys., 17, 3891-3903, https://doi.org/10.5194/acp-17-3891-2017, 2017.

Yoo, J. M., Jeong, M. J., Kim, D., Stockwell, W. R., Yang, J. H., Shin, H. W., Lee, M. I., Song, C. K., and Lee, S. D.: Spatiotemporal variations of air pollutants ($O_3$, $NO_2$, $SO_2$, CO, $PM_{10}$, and VOCs) with land-use types, Atmos. Chem. Phys., 15, 10857-10885, https://doi.org/10.5194/acp-15-10857-2015, 2015.

Zhan, J. L., Feng, Z. M., Liu, P. F., He, X. W., He, Z. M., Chen, T. Z., Wang, Y. F., He, H., Mu, Y. J., and Liu, Y. C.: Ozone and SOA formation potential based on photochemical loss of VOCs during the Beijing summer, Environ. Pollut., 285, 117444, https://doi.org/10.1016/j.envpol.2021.117444, 2021.

Zhang, Y. C., Li R., Fu H. B., Zhou D., and Chen J. M.: Observation and analysis of atmospheric volatile organic compounds in a typical petrochemical area in Yangtze River Delta, China, J. Environ. Sci., 71, 233-248, https://doi.org/CNKI:SUN:HJKB.0.2018-09-022, 2018.

---

## Referee Report (RR1)

**Comments on ACP ms – VOCs in Shanghai**

The authors have made lots of efforts to revise this manuscript according to the comments from reviewers. Additional information and discussions were provided, and the English writing was improved. However, some of the questions were not fully elaborated in terms of the research highlight, the rationality of the methods, and the result interpretations. Therefore, I think the current manuscript is not ready for publication until the below comments are addressed.

1. The authors emphasized that the VOCs characteristics and their impacts on $O_3$ and SOA under different land-use types remained a gap and highlighted that the multi-site measurements would be able to fill this gap. In fact, the abundance, compositions, sources, and the $O_3$ and SOA chemistry have been extensively investigated not only in Shanghai but also around the globe. Several studies have measured VOCs in the same locations (line 240-244). Despite the different TVOC concentrations, the results in this study were consistent with previous studies, including the VOC compositions and source contributions. Therefore, I wonder what new insight this work would bring in addition to comparing the reported results. What's more, a similar study has been published, which measured VOCs at the same 3 sites concurrently and investigated the VOC characteristics, sources, and secondary formation potentials (Wang et al., 2022).

2. The causation between land-use and VOC sources was confusing. For example, in line 63-64, "the land-use types influence not only the VOC concentrations but also the sources, especially the anthropogenic sources". Actually, the spatial variations of VOC abundance and compositions were due to the different major emission sources in different land-use types. Meanwhile, the definition of different land-use types in this study was based on the functional types of the areas, which include different source sectors inside. Thus, the results in this study are mostly attributable to different emission sources, and they can hardly be concluded to the impacts of the different land-use types directly as in the Discussion section. Besides, though the term "land-use" was repeatedly emphasized throughout the manuscript, the relation between the findings and the land-use types was weak.

3. Line 150-152: The meteorological parameters were measured at one weather station, while the meteorological factors were different among the 3 sites in the results (Figure 1). The authors need to clarify the data source.

4. The COD values were used to estimate the divergence of TVOC concentrations among sites. What were the thresholds of high and low similarity? The values 0.20, 0.33, and 0.36 seem to be small, as CODs of 0.269 and 0.783 were used to determine the similar and dissimilar sites in the cited paper (Wongphatarakul et al., 1998).

5. The sensitivity analysis is confusing. Firstly, why were background values needed in the calculation? What were the "specific $PM_{2.5}$ gradients" in line 213? What were the "corresponding VOC concentrations" in line 220? Secondly, how to derive equations 14 and 15? Which parameter in the calculation indicates sensitivity? Most importantly, why do we need to analyze the sensitivity of VOCs to $PM_{2.5}$ instead of the other way around? Line 220-221: "the concentration of VOCs was greatly affected by the variations of $PM_{2.5}$ concentration." And Line 229.

6. As shown in Figure 1, two prominent peaks of VOC concentrations were observed at JS. In spite of the peaks, the TVOC level at JS was comparable and even lower than that at PD. While the authors only discussed the mean TVOC concentrations at 3 sites, the special events were concealed.

7. Line 248-249: Why were the average $PM_{2.5}$ concentrations comparable among 3 sites, whereas the $O_3$ levels were higher at QP than at the other sites?

8. Line 257: How to infer that "The termination and titration ($NO + O_3 \rightarrow NO_2 + O_2$) were more efficient"?

9. Line 250-260: Pearson correlation coefficient was applied to estimate the correlations of VOCs and $O_3$ and VOCs and $PM_{2.5}$. The Pearson correlation coefficient is to measure the linear correlation between two datasets. However, $O_3$ and $PM_{2.5}$ have non-linear relationships with VOCs. Hence, the feasibility of this estimation should be justified.

10. Line 305-306: haze days were defined as "visibility $< 10$ km and RH $> 80$ %". Given the high RH, how to distinguish haze from fog?

11. Line 313-314: Why JS and PD had stagnant weather while QP didn't? How to define stagnant weather, and what were the weather conditions on haze days?

12. Line 316-317: The authors attributed the elevated VOC concentrations on haze days to enhanced emissions. Is there any evidence for the enhanced emissions, especially the vehicle exhausts?

13. Figure 5: How to define the different thresholds of VOC ratios? If the reference values were obtained from previous studies, are they comparable and suitable for this study?

14. Line 355 and 367: What do you mean by "during the VOC pollution"?

15. The source profiles apportioned from PMF results were highly doubtful. The problems are not limited to the followings. 1) Industrial source was not resolved at QP, while the biogenic source was not resolved at JS and PD. Are the source apportionment results comparable among these sites? 2) From Figure 6c, a high proportion of 1,2,3-trimethylbenzene was apportioned to the biogenic source at QP. The proportions of aromatics were even higher than isoprene in the biogenic source. 3) Isoprene, as a tracer for biogenic sources, was apportioned to the industrial source at JS and paint solvent usage at PD. 4) At QP, $C_2$ species were apportioned to vehicle exhausts, while they were also tracers for coal combustion. In addition, coal combustion at QP contained a large proportion of ethylbenzene, which was not explained. 5) Line 413-414: Fuel evaporation was identified by $C_3$-$C_7$ species. However, high proportions of aromatics, such as benzene at JS, and isopropylbenzene at PD, were also apportioned to this source. In addition, the contributions of fuel evaporation to TVOC at QP (20.15%) were much higher than those of the other sites. What could be the reason for that? 6) How about the correlations between every two sources (G-Space plot)?

16. According to the PSCF results, regional transport of VOCs, especially from northern regions, also contributed to the VOC concentrations at 3 sites. While the VOCs were apportioned to local sources in the PMF model, which part of the VOCs was accountable for the regional transport?

17. Line 472-473 and 497: How to quantify the percentage of OFP and SOAFP to $O_3$ and $PM_{2.5}$ concentrations?

18. As discussed in line 500-504, large uncertainty existed for the SOAFP method. Indeed, the OFP and SOAFP methods are only based on the reference reactivity of VOCs, which species with high reactivity tend to have larger secondary yields. The results herein are rather general and widely known. It is hard to tell what new findings we can get from this method.

*Reference*

Wang, S., Zhao, Y., Han, Y., Li, R., Fu, H., Gao, S., . . . Chen, J. (2022). Spatiotemporal variation, source and secondary transformation potential of volatile organic compounds (VOCs) during the winter days in Shanghai, China. Atmospheric Environment, 286, 119203. doi:https://doi.org/10.1016/j.atmosenv.2022.119203

Wongphatarakul, V., Friedlander, S. K., & Pinto, J. P. (1998). A comparative study of PM2.5 ambient aerosol chemical databases. Journal of Aerosol Science, 29, S115-S116. doi:https://doi.org/10.1016/S0021-8502(98)00164-5

---

## Referee Report (RR2)

The revised manuscript is significantly improved, compared with the last version. The manuscript presents a study based on the concurrent observations of volatile organic compounds (VOCs) at three supersites sites in Shanghai during the first three months of 2019. The characteristics of VOCs (chemical composition, weekend effects and discrepancy of clean and polluted days), ozone formation potential, secondary organic aerosol formation potential and emission sources are discussed. Moreover, the authors have added detailed discussions about the influence of different land use type on the VOC characteristics. This paper can bring new insight to the VOC study in the developed region in China. However, the paper still needs grammatical revisions and corrections of writing errors in its present form. The paper could be accepted after minor revisions.

**Specific comments:**

Line 14: VOCs is precursor of SOA, which is accounted for large proportion of aerosol. It is Aerosol that have great impacts on climate change. I think the expression that VOCs have important impacts on climate change in not appropriate here.

Line 61-62: Should be: "similar as/in agreement with the finding of Tang et al. (2008)".

Line 174: Why do the author set the EF to be 0.1, same for all VOCs species used for PMF? And there are two and after $Q_{true}/Q_{robust}$.

Line 212: $\Delta O_3$ should be $\Delta PM_{2.5}$.

Line226: $Ln(\Delta PM2.5./BVOCs)$ should be $Ln(\Delta PM2.5./B\ PM2.5)$

Line 246: There was two prominent peaks of VOCs concentration at JS, which is attributed to the elevated traffic and industrial VOC emissions and stagnant synoptic conditions. How about the VOCs concentration at the other two sites during the same period? Comparison during high VOCs concentration episodes can reflect weather the different influence of land use is the key factor on VOCs concentration (or synoptic condition?).

Line 249: should be 38.85%.

Line 252: should be "there were pronounced enhancement or increase in the industrial

production….".

Line 251: The spearman correlation coefficient between VOCs and $PM_{2.5}$ at QP was low (0.34) and was much lower than the other two sites. It was not proper to make the conclusion that the elevated VOCs lead to the elevation of $PM_{2.5}$ at QP site, or VOCs and $PM_{2.5}$ at QP have similar emission sources.

Line 322: The author suggested that ethyl toluene, ethylbenzene and trimethyl benzene at JS and PD was related with vehicle exhaust. However, the PMF results indicated that the contribution of vehicle exhaust to these VOCs is quite minor. Please check.

Line 327: The author suggested that the main source of n-hexane was vehicle exhaust and fuel evaporation in QP, which is not consistent with the results shown in Figure 6c. The PMF results suggested that vehicle exhaust contributed almost zero to n-hexane, while fuel production and evaporation and paint solvent usage was the main sources.

Line 373: Miss a period.

Add explanation of the figure 7 in the caption: what the contour means in the left penal? How did the author calculate P_TVOCs of each cluster?

Line 485-Line 488: Alkenes (mainly ethene and propene) were the main contributor of OFP at the three sites. What's the source of these reactive VOC species at the three sites?

---

## Author Response (AR2)

Dear Prof. Wang,

We would like to thank the anonymous reviewer for the great efforts. On the basis of the reviewer's suggestions, we have updated this manuscript greatly. We greatly appreciate those comments and valuable suggestions from the reviewer. Also, we are grateful to your efficient serving for this manuscript.

Here we submit our revised manuscript **"Measurement report: VOC characteristics at different land-use types in Shanghai: spatio-temporal variation, source apportionment, and impact on secondary formations of ozone and aerosol" (Manuscript number: acp-2022-250)**. In the attachments, a point-by-point response to each point raised from the reviewer was uploaded. The revised version marked for reviewing and the clean version for editing were supplied, respectively.

Yours sincerely,

Hongbo Fu

Department of Environmental Science & Engineering
Fudan University
200000 Shanghai China
E-mail: fuhb@fudan.edu.cn

**Response to the reviewer**

**General Comments:**

The authors have made lots of efforts to revise this manuscript according to the comments from reviewers. Additional information and discussions were provided, and the English writing was improved. However, some of the questions were not fully elaborated in terms of the research highlight, the rationality of the methods, and the result interpretations. Therefore, I think the current manuscript is not ready for publication until the below comments are addressed.

**Response:**

We appreciate the thoughtful and valuable suggestions by the reviewer, which are helpful for us to improve the MS quality greatly. We have updated the manuscript based on these valuable suggestions. The updated version has provided more elaborations on the research highlight, the rationality of the methods, and the result interpretations. We also made efforts to smooth the English writing thoroughly.

**Comment 1:** The authors emphasized that the VOCs characteristics and their impacts on $O_3$ and SOA under different land-use types remained a gap and highlighted that the multi-site measurements would be able to fill this gap. In fact, the abundance, compositions, sources, and the $O_3$ and SOA chemistry have been extensively investigated not only in Shanghai but also around the globe. Several studies have measured VOCs in the same locations (line 240-244). Despite the different TVOC concentrations, the results in this study were consistent with previous studies, including the VOC compositions and source contributions. Therefore, I wonder what new insight this work would bring in addition to comparing the reported results. What's more, a similar study has been published, which measured VOCs at the same 3 sites concurrently and investigated the VOC characteristics, sources, and secondary formation potentials (Wang et al., 2022).

**Response:**

Thank you for your valuable suggestions. Indeed, several field-based VOC observations have been performed in the same locations. However, the research times and objectives of previous studies were different from those in this study. Cai et al.

(2010a) did not study the VOC sources, SOA formation potentials and the sensitivity of VOCs to PM$_{2.5}$ at the PD site. The previous study at the JS site did not calculate the SOA formation potentials and the sensitivity of VOCs to PM$_{2.5}$ which was the single-site measurement (Zhang et al., 2018). The study of Zhang et al. (2020) at the QP site did not determine VOC sources, ozone and SOA formation potentials and the sensitivity of VOCs to PM$_{2.5}$ which was also the single-site measurement. The policy of "one factory, one strategy", targeted at mitigating VOC emissions, was published by the Shanghai government in 2018. Our study could reflect the variations of VOC characteristics since the implementation of the policy.

The main objectives of Wang et al. (2022) were to study VOC concentrations, sources and O$_3$ and SOA formation potentials at three sites, rather than the horizontal comparisons among multiple sites. Especially, the study mentioned did not discuss the effects of land-use types on VOC characteristics, sources, and the sensitivity of VOCs to PM$_{2.5}$. Thus, the scientific problems concerned by Wang et al. (2022) and this work were different greatly, especially based on the measurement data at different time spans. In this study, the concurrent multiple-site and high-time resolution measurement of VOCs with the typical land-use types in Shanghai for their concentrations, sources, ozone and SOA formation potentials and sensitivity of VOCs to PM$_{2.5}$ were performed. The results shown herein highlight that the simultaneous multiple-site measurements with the different land-use types at a megacity or city cluster level could be more appropriate to fully understand the VOC characteristics, which could provide more information on the accurate air-quality control inside a megacity worldwide.

**Comment 2:** The causation between land-use type and VOC source was confusing. For example, in line 63-64, "the land-use types influence not only the VOC concentrations but also the sources, especially the anthropogenic sources". Actually, the spatial variations of VOC abundance and compositions were due to the different major emission sources in different land-use types. Meanwhile, the definition of different land-use types in this study was based on the functional types of the areas, which include different source sectors inside. Thus, the results in this study are mostly

attributable to different emission sources, and they can hardly be concluded to the impacts of the different land-use types directly as in the Discussion section. Besides, though the term "land-use" was repeatedly emphasized throughout the manuscript, the relation between the findings and the land-use types was weak.

**Response:**

Thank you for your comments. The fundamental reason that caused the spatial variations of VOC abundance, sources, and sensitivity to the $O_3$ and SOA formation was the land-use type which was one of the objectives of this study. In detail, the JS site is located in the Second Jinshan Industrial Area of Shanghai as the industrial district and is surrounded by many chemical factories. The PD site is located in the Pudong New Area as the residential and commercial mixed district and is surrounded by residences and administrative areas. The QP site is located near the southeast of Dianshan Lake as the background district and is surrounded by many farmlands and forests. The land-use types of the JS and PD sites led to high anthropogenic emissions which resulted in the fact that the VOC concentrations were approximately twice higher than those at the QP site. Moreover, the distinct land-use types among the sampling sites also led to the difference in VOC sources. The vehicle exhaust was determined as the predominant source at the three sites. The second largest VOC contributor was identified as industrial production at the JS and PD sites, whereas it proved to be the fuel production and evaporation at the QP site. The limited influence of industrial sources at the QP site was observed which was related to the land-use type of this site. Relative to the QP site, JS and PD sites were less affected by biomass burning which was consistent with the regional characteristics of anthropogenic activities dominated by land-use types. Additionally, the $O_3$ and SOA formations were strongly affected by the land-use types. The higher OFPs ($50.85 \pm 2.63$ and $33.94 \pm 1.52$ ppb) and SOAFPs ($1.00 \pm 2.03$ and $0.46 \pm 0.88$ µg m$^{-3}$) at the JS and PD sites relative to those at the QP site ($24.26 \pm 1.43$ ppb for OFPs and $0.41 \pm 0.58$ µg m$^{-3}$ for SOAFPs, respectively) were observed, in connection with the land-use types. Further, the VOCs-PM$_{2.5}$ sensitivity analysis showed that VOCs at the QP site showed a more rapid increment along with the increase of PM$_{2.5}$ values compared with the

other two sampling sites. Of the four VOC categories, aromatics at the JS and PD sites and alkanes at the QP site were more sensitive to PM$_{2.5}$. Therefore, the findings in this study were greatly related to the land-use types. We have rewritten the description in the revised manuscript.

Page 2, lines 63-65:

> "*Besides, the land-use types were also related to the VOC sources, especially the anthropogenic sources (Yoo et al., 2015; Chen et al., 2017; Wang et al., 2017; Jookjantra et al., 2022).*"

**Comment 3:** Line 150-152: The meteorological parameters were measured at one weather station, while the meteorological factors were different among the 3 sites in the results (Figure 1). The authors need to clarify the data source.

**Response:**

Thank you for your comments. The data source has been revised.

Page 5, lines 150-151:

> "*The meteorological variables including temperature, RH and wind speed were acquired from each air monitoring station.*"

**Comment 4:** The COD values were used to estimate the divergence of TVOC concentrations among sites. What were the thresholds of high and low similarity? The values 0.20, 0.33, and 0.36 seem to be small, as CODs of 0.269 and 0.783 were used to determine the similar and dissimilar sites in the cited paper (Wongphatarakul et al., 1998).

**Response:**

Thank you for your comments. There was no clear definition of high and low similarity thresholds. The values of COD can evaluate the degree of air pollutant concentration difference between two different sampling sites, which varies with the study areas and times (Ma et al., 2019). A greater COD value means less redundancy of the data between two sites (Ma et al., 2019). Wongphatarakul et al. (1998) found that the COD between Taipei and downtown Los Angeles (COD = 0.783) was higher than that between Teplice and downtown Los Angeles (COD = 0.269), illustrating that the most dissimilar and the greatest similar at the corresponding two sites were

observed. Meanwhile, Song et al. (2017) found that there were spatial variations of PM$_{2.5}$ (COD = 0.34), NO$_2$ (COD = 0.34), PM$_{10}$ (COD = 0.45) and CO (COD = 0.32) among different Chinese cities. Similarly, the COD values herein between the JS-QP, PD-QP and JS-PD were 0.36, 0.33 and 0.20, respectively, indicating that the spatial heterogeneity of VOCs between the JS and PD sites was narrow, while the QP site was largely different from other two sites. We added the description in the revised manuscript.

Page 18, lines 549-552:

*" Note that the distinct spatial heterogeneity of VOCs was also observed with the highest value of the coefficient of divergence (COD = 0.36) between the JS and QP sites, followed by the PD and QP sites (COD = 0.33), with that between the JS and PD sites (COD = 0.20) being the lowest. A greater COD value means less redundancy of the data between two sites (Ma et al., 2019)."*

**Comment 5:** The sensitivity analysis is confusing. Firstly, why were background values needed in the calculation? What were the "specific PM$_{2.5}$ gradients" in line 213? What were the "corresponding VOC concentrations" in line 220? Secondly, how to derive equations 14 and 15? Which parameter in the calculation indicates sensitivity? Most importantly, why do we need to analyze the sensitivity of VOCs to PM$_{2.5}$ instead of the other way around? Line 220-221: "the concentration of VOCs was greatly affected by the variations of PM$_{2.5}$ concentration." And Line 229.

**Response:**

Thank you for your suggestions. The method of calculating the sensitivity of VOCs to PM$_{2.5}$ was used for the gradient model (Eq. 11), which needed the background levels.

$$VOCs\text{–}S_{PM_{2.5}} = \frac{\Delta_{VOCs}/B_{VOCs}}{\Delta_{PM_{2.5}}/B_{PM_{2.5}}} \tag{11}$$

The "specific PM$_{2.5}$ gradients" in line 212 was the five levels of PM$_{2.5}$: clean level (PM$_{2.5}$ < 35 µg m$^{-3}$), slight pollution level (35 < PM$_{2.5}$ < 75 µg m$^{-3}$), medium pollution level (75 < PM$_{2.5}$ < 120 µg m$^{-3}$), heavy pollution level (120 < PM$_{2.5}$ < 180 µg m$^{-3}$) and extreme pollution level (PM$_{2.5}$ > 180 µg m$^{-3}$).

The "corresponding VOC concentrations" in line 219 was the VOC concentrations under the different PM$_{2.5}$ values.

The $x$-axis was PM$_{2.5}$/B$_{PM2.5}$ and the $y$-axis was the values of VOCs-S$_{PM2.5}$. We put the corresponding values into the Eq. (13).

$$\ln y = \ln a + b \cdot \ln x \tag{13}$$

We revised the Eq. (14) in line 224.

$$\ln VOCs - S_{PM_{2.5}} = \ln a + b \cdot \ln \frac{\Delta_{PM_{2.5}}}{B_{PM_{2.5}}} \tag{14}$$

According to the Eq. (11), we derived the Eq. (15). The evolution of calculation as follow.

$$\ln \frac{\Delta_{VOCs} / B_{VOCs}}{\Delta_{PM_{2.5}} / B_{PM_{2.5}}} = \ln a + b \cdot \ln \frac{\Delta_{PM_{2.5}}}{B_{PM_{2.5}}}$$

$$\ln \frac{\Delta_{VOCs}}{\Delta_{PM_{2.5}}} = \ln a + (1 + b) \cdot \ln \frac{\Delta_{PM_{2.5}}}{B_{PM_{2.5}}}$$

$$\ln \frac{\Delta VOCs}{BVOCs} = k \cdot \ln \frac{\Delta_{PM_{2.5}}}{B_{PM_{2.5}}} + c \tag{15}$$

The values of VOCs-S$_{PM2.5}$ was the sensitivity between the VOCs and PM$_{2.5}$. The larger the VOCs-S$_{PM2.5}$ value, the more sensitive the VOC concentrations were to PM$_{2.5}$. We rewrote line 647 into "*The VOCs at the QP site showed a more rapid increment along with the increase of PM$_{2.5}$ values.*"

We could quantitatively reveal patterns of variations between the VOCs and PM$_{2.5}$, and evaluate the degree by which VOCs were impacted by PM$_{2.5}$ *via* analyzing the sensitivity of VOCs to PM$_{2.5}$ under the different land-use types. Accordingly, the VOC-induced haze pollution could be efficiently controlled by decreasing corresponding VOC concentrations. Han et al. (2017) also calculated the sensitivity of VOCs to PM$_{2.5}$ and showed that alkanes and alkenes were more sensitive to PM$_{2.5}$ than aromatic compounds.

We have rewritten the description in detail in the revised manuscript.

Page 8, lines 219:

*"The higher value of VOCs-$S_{PM2.5}$, the more sensitive VOCs to the $PM_{2.5}$ concentrations."*

Page 8, lines 226-227:

*"This method was appropriate for understanding the sensitivity of VOC concentrations to $PM_{2.5}$."*

**Comment 6:** As shown in Figure 1, two prominent peaks of VOC concentrations were observed at JS. Despite the peaks, the TVOC level at JS was comparable and even lower than that at PD. While the authors only discussed the mean TVOC concentrations at 3 sites, the special events were concealed.

**Response:**

Thank you for your valuable suggestions. The two prominent peaks of VOC concentrations were observed at the JS site on 23 January and 11 March, respectively. The highest VOC concentrations appeared on 11 March. There was leaking of the chemical factories *via* inquiring about the local workers. Therefore, the highest VOC concentration on 11 March might be attributed to the leaking of the chemical factories. The second highest VOC concentration was observed on 23 January, which was due to the increased vehicle exhaust and industrial processes and the decreased wind speed. Based on the reviewer's comments, we added the new discussions about the special events in the revised manuscript.

Page 9, lines 246-255:

*"The two prominent peaks of VOC concentrations were observed at the JS site on 23 January and 11 March, respectively. The highest VOC concentration appeared on 11 March with a value of 84.49 ppb. This phenomenon might be attributed to the leaking of chemical factories. The second highest VOC concentration was observed on 23 January with a value of 77.71 ppb which was 38.85 and 57.07 % higher than those at the PD and QP sites, respectively. According to the Shanghai Municipal Bureau of Statistics (http://tjj.sh.gov.cn), the traffic flow in January was ~ 10 % higher than that in the following two months. Such scenario was likely due to the Spring Festival Travel rush, i.e., population travel intensively occurred around the Chinese Spring Festival.*

*Moreover, there were pronounced in the industrial production in January compared with those in February (~ 36 % uplift) and March (~ 6 % uplift) (http://tjj.sh.gov.cn). The phenomena could be responsible for the elevated VOC emissions. Additionally, the lowest WS was observed on 23 January with the value of 0.85 m s$^{-1}$, which was adverse to dispersive dilution and convection of VOCs, causing high concentrations of VOC at the JS site (Kumar et al., 2018).*"

**Comment 7:** Line 248-249: Why were the average $PM_{2.5}$ concentrations comparable among 3 sites, whereas the $O_3$ levels were higher at QP than at the other sites?

**Response:**

Thank you for your questions. The influence factors were different between the $O_3$ and $PM_{2.5}$. The $O_3$ concentrations were primarily affected by the values of precursors including VOCs, $NO_x$ and CO (Hu et al., 2022), meteorological factors including light intensity and radiation, wind speed and temperature (Haberer et al., 2006), and the strength of anthropogenic emissions such as transportation and industrial sources (Yang et al., 2021). However, the $PM_{2.5}$ levels are influenced by many factors which were related to the composition. The primary composition of $PM_{2.5}$ is water-soluble ions such as $SO_4^{2-}$, $NO_3^-$ and $NH_4^+$, all of which are closely correlated with the atmospheric $SO_2$, $NO_x$ and $NH_3$ concentrations (Liang et al., 2019). The carbonaceous aerosols accounting for approximately 40-60% of $PM_{2.5}$ which included element carbon and organic carbon are influenced by emissions from different combustion methods, biogenic sources, VOC and radical concentrations and light intensity (Yao, 2016; Ryou et al., 2018). The metal elements such as Ca, Si, Fe, Mg, Zn and Mn are also components of $PM_{2.5}$ which varied with emission types (Zhao et al., 2021). For example, the dominating source of V and Ni is diesel and fuel oil combustion (Zhao et al., 2021), while the non-tailpipe emissions from motor vehicles are an important emission source of Cu, Ba and Sb (Nicolas, 2009).

**Comment 8:** Line 257: How to infer that "The termination and titration (NO + $O_3$ → $NO_2$ + $O_2$) were more efficient"?

**Response:**

Thank you for your comment. We rewrote the description in the revised manuscript.

Page 10, lines 265-266:

> *"Lots of factors including $NO_x$ levels, sunshine duration, temperature and relative humidity not only the emission of precursors, impacted on the surface $O_3$."*

**Comment 9:** Line 250-260: Pearson correlation coefficient was applied to estimate the correlations of VOCs and $O_3$ and VOCs and $PM_{2.5}$. The Pearson correlation coefficient is to measure the linear correlation between two datasets. However, $O_3$ and $PM_{2.5}$ have non-linear relationships with VOCs. Hence, the feasibility of this estimation should be justified.

**Response:**

Thank you for your comment. We analyzed the non-linear relationships of $PM_{2.5}$ and $O_3$ with VOCs. The results showed the concentrations of VOCs were found to be positively correlated with that of $PM_{2.5}$, and the spearman correlation coefficients ($R_{Spearman}$) were 0.72, 0.74 and 0.34 at the JS, PD and QP sites, respectively. However, the VOC concentrations were negatively correlated with $O_3$ ($R_{Spearman}$ = -0.39 at the JS site, $R_{Spearman}$ = -0.50 at the PD site and $R_{Spearman}$ = -0.40 at the QP site, respectively). We have written the description in detail in the revised manuscript.

Page 9, lines 258-259:

> *"VOCs was found to be positively correlated with $PM_{2.5}$, and the spearman correlation coefficients ($R_{Spearman}$) were 0.72, 0.74 and 0.34 at the JS, PD and QP sites, respectively. "*

Page 10, lines 263-265:

> *"However, the VOC concentrations were negatively correlated with $O_3$ ($R_{Spearman}$ = -0.39 at the JS site, $R_{Spearman}$ = -0.50 at the PD site and $R_{Spearman}$ = -0.40 at the QP site, respectively)."*

**Comment 10:** Line 305-306: haze days were defined as "visibility < 10 km and RH > 80 %". Given the high RH, how to distinguish haze from fog?

**Response:**

Thank you for your question. We have revised the statement in the revised manuscript.

Page 11, lines 312-313:

> "*Referring to the previous documents (Li et al., 2017; Hui et al., 2019), haze pollution was defined as the condition with visibility < 10 km and RH < 80 %.*"

**Comment 11:** Line 313-314: Why JS and PD had stagnant weather while QP didn't? How to define stagnant weather, and what were the weather conditions on haze days?

**Response:**

Thank you for your question. Herein, the impacts of human activities at the JS and PD sites were heavier than that at the QP site because of the land-use types. This phenomenon caused the high emission of VOCs i.e., VOC concentrations at the JS (21.88 ± 12.58 ppb) and PD (21.36 ± 8.58 ppb) sites were approximately twice higher than that at the QP site (11.93 ± 6.33 ppb). Moreover, the wind speed at the QP site (4.37 ± 1.47 m s$^{-1}$) was 2.29 and 1.36 times higher than those at the JS (1.91 ± 0.49 m s$^{-1}$) and PD (1.30 ± 0.62 m s$^{-1}$) sites, respectively, which decreased the dilution and diffusion conditions at the latter two sites. The above phenomena resulted in the stagnant weather appearing at the JS and PD sites compared with the QP site. We have written the description in the revised manuscript.

Page 11, lines 319-321:

> "*Such scenario could be attributed to the locations of JS and PD sites which led to high anthropogenic emissions and low wind speed and implicated stagnant weather conditions, therefore inducing the severe haze pollution.*"

The stagnate weather always accomplishes low wind speed for more than 24 h which could decrease dispersive dilution and convection phenomenon and greatly contribute to the formation of haze days (Zhang et al., 2016).

The weather conditions on haze days were visibility < 10 km and RH < 80 %. At the JS site, the average wind speed and temperature were 1.67 m s$^{-1}$ and 9.24 °C on haze days, respectively. At the PD site, the average wind speed and temperature were 1.20 m s$^{-1}$ and 9.21 °C on haze days, respectively. At the QP site, the average wind speed and temperature were 4.03 m s$^{-1}$ and 7.64 °C on haze days, respectively.

**Comment 12:** Line 316-317: The authors attributed the elevated VOC concentrations on haze days to enhanced emissions. Is there any evidence for the enhanced emissions, especially the vehicle exhausts?

**Response:**

Thank you for your comment. The concentrations of aromatics especially *m*-ethyltoluene, *p*-ethyltoluene, ethylbenzene and 1, 2, 4-trimethylbenzene were significantly higher than those on clean days at the JS and PD sites. Industrial production was characterized by high contributions of aromatics including *m*-ethyltoluene and *p*-ethyltoluene (Hui et al., 2019). Ethylbenzene was closely related to the painting/coating (Cai et al., 2010b; Ling et al., 2011; Hui et al., 2019). Trimethylbenzene was closely correlated to vehicle exhaust, furniture manufacturing and painting/coating (Chan et al., 2006; Ling et al., 2011; Liu et al., 2008; Hui et al., 2019). Thus, the elevated VOC concentrations on haze days were greatly influenced by enhanced emissions. We have written the description in the revised manuscript.

Page 11, lines 323-325:

> "*The above VOC compounds were related with the industrial production, painting/coating and vehicle exhaust, and elevated concentrations reflected the concentrated emission sources.*"

**Comment 13:** Figure 5: How to define the different thresholds of VOC ratios? If the reference values were obtained from previous studies, are they comparable and suitable for this study?

**Response:**

Thank you for your valuable suggestions. The special VOC ratios were widely used to preliminary distinguish the VOC sources. The reference values were obtained from previous studies which were used widely by many following works. For example, the T/B ratios ranged from 1.4 ± 0.8 to 5.8 ± 3.4 by different industrial processes in the different studies (Mo et al., 2015; Shi et al., 2015; Song et al., 2021). Moreover, many previous studies found that the ratios of T/B ranging from 0.9 ± 0.6 to 2.2 ± 0.5 indicated traffic-related sources (Qiao et al., 2012; Dai et al., 2013; Wang et al., 2013; Yao et al., 2013; Zhang et al., 2013; Yao et al., 2015; Mo et al., 2016; Deng et al., 2018; Song et al., 2021). The special VOC ratios were distributed within the reference range of one source, indicating that the source contributed more VOCs than other sources. In this study, most ratios (68.89 and 84.15 %) distributed the range of 0.9-2.2

and 1.4-5.8, suggesting that vehicle emissions and industrial emissions exerted a significant impact on VOC concentrations at the JS and PD sites. Nearly half of T/B ratios (43.02 %) distributed the range of 0.9-2.2 and 0.2-0.4, suggesting that vehicle emissions and burning emissions contributed significantly to VOC pollution at the QP site.

**Comment 14:** Line 355 and 367: What do you mean by "during the VOC pollution"?

**Response:**

Thank you for your question. The "during VOC pollution" means "when the VOC concentrations were high". We rewrote the description in the revised manuscript.

Page 13, lines 363-365:

> *"The P/P ratios were distributed within the range of 2.2-3.8 at the sampling sites when the VOC concentrations were high, indicating the great impact of vehicle emission on VOC pollution, which was in agreement with the report of Song et al. (2021)."*

Page 13, lines 374-375:

> *"When the VOC concentrations were high, the X/E ratios were approximately 2.3, 2.5 and 1.8 at the JS, PD and QP sites, respectively.*

**Comment 15:** The source profiles apportioned from PMF results were highly doubtful. The problems are not limited to the followings. 1) Industrial source was not resolved at QP, while the biogenic source was not resolved at JS and PD. Are the source apportionment results comparable among these sites? 2) From Figure 6c, a high proportion of 1,2,3-trimethylbenzene was apportioned to the biogenic source at QP. The proportions of aromatics were even higher than isoprene in the biogenic source. 3) Isoprene, as a tracer for biogenic sources, was apportioned to the industrial source at JS and paint solvent usage at PD. 4) At QP, $C_2$ species were apportioned to vehicle exhausts, while they were also tracers for coal combustion. In addition, coal combustion at QP contained a large proportion of ethylbenzene, which was not explained. 5) Line 413-414: Fuel evaporation was identified by $C_3$-$C_7$ species. However, high proportions of aromatics, such as benzene at JS, and isopropylbenzene at PD, were also apportioned to this source. In addition, the contributions of fuel

evaporation to TVOC at QP (20.15%) were much higher than those of the other sites. What could be the reason for that? 6) How about the correlations between every two sources (G-Space plot)?

**Response:**

Thank you for your suggestions. 1) The anthropogenic source was the dominating source of VOCs at the JS and PD sites, and the impact of biomass burning on VOCs was limited because of the land-use types. The QP site is located near the southeast of Dianshan Lake as a city background site where the tracers of industrial sources were limited. Thus, the industrial emission and biomass burning source showed slight contributions to VOC concentrations at the QP and JS/PD sites, respectively. Song et al. (2021) studied the VOC sources at three sites in Xi'an, China, and found the biogenic source was not resolved at the CB site while the fuel evaporation was not resolved at the other two sites (DHS and QL sites).

2) We added new sentences in the revised manuscript.

Page 15, lines 434-438:

> "*Biomass burning was distinguished by the isoprene and some aromatics (Schauer et al., 2001; Liu et al., 2020; Yang et al., 2023). Factors that coincide with the specific characteristic were regarded as biomass burning in this study. Our analysis showed that the contribution of biomass burning to VOC concentrations was 11.39 % at the QP site. However, this emission factor could not be reproduced at the JS and PD sites, implying the limited impacts of biomass burning in the population- and industrialization-concentrated areas that were primarily controlled by the anthropogenic emissions.*"

3) Wood combustion could be used in industrial processes. Meanwhile, wood was also regarded as the raw material in painting. Thus, the above sources could release the isoprene. The previous study also showed that a certain amount of isoprene could be found in the industrial source, vehicle exhaust and paint solvent usage (Song et al., 2021).

4) Previous studies showed that $C_2$ species like ethylene and ethyne could be found both in vehicle exhaust and coal combustion (Liu et al., 2008; Ling et al., 2011; Song et al., 2018; Song et al., 2021). We rewrote the description in the revised manuscript.

Page 14, lines 406-407:

"*Coal combustion factor was characterized by $C_2$-$C_3$ alkenes such as ethylene and propylene, some alkanes, ethyne, benzene and ethylbenzene (Liu et al., 2008; Ling et al., 2011; Song et al., 2018).*"

5) We rewrote the description in the revised manuscript after consideration.

Page 14, lines 420-428:

"*Fuel production and evaporation could be identified by $C_3$-$C_7$ alkanes, especially n-pentane and iso-pentane (Zheng et al., 2020), the $C_3$-$C_5$ alkenes such as trans/cis-2-butene (Geng et al., 2009; Hui et al., 2018; Zhang et al., 2018a; Zheng et al., 2020) and some aromatics (Liu et al., 2008; Xiong et al., 2020). There were high contributions of n-pentane (37.94, 25.20 and 71.95 %), iso-pentane (41.45, 28.18 and 23.16 %) and butene (26.73, 70.83 and 22.52 %) at the JS, PD and QP sites, respectively. The contributions of some aromatics such as benzene at the JS site (44.08 %), isopropylbenzene at the PD site (66.71 %) and 1, 2, 4-trimethylbenzene at the QP site (32.90 %) were also high. It was well documented that alkanes like n-pentane and iso-pentane were gasoline tracers, and some alkanes and aromatics could evaporate from the unburned fuels (Guo et al., 2004; Liu et al., 2008; Wang et al., 2013; Xiong et al., 2020). The VOC contributions from fuel production and evaporation were calculated to be 4.62, 10.35 and 20.15 % at the JS, PD and QP sites, respectively.*"

The QP site is located around Dianshan Lake which is a tourist attraction, with large contributions of fuel production and evaporation from vehicles and gas stations (Chen et al., 2021).

6) There were no correlations between the two sources.

Table 1: The correlations between two sources at the JS, PD and QP sites.

| JS site | Source 1 | Source 2 | Source 3 | Source 4 | Source 5 |
|---|---|---|---|---|---|
| Source 2 | -0.15 | | | | |
| Source 3 | -0.27 | -0.25 | | | |
| Source 4 | -0.14 | -0.64 | -0.16 | | |
| Source 5 | -0.08 | -0.13 | -0.14 | -0.29 | |
| Source 6 | -0.20 | -0.29 | -0.20 | -0.30 | -0.13 |

| PD site | Source 1 | Source 2 | Source 3 | Source 4 | Source 5 |
|---|---|---|---|---|---|
| Source 2 | -0.25 | | | | |
| Source 3 | 0.09 | -0.24 | | | |
| Source 4 | -0.24 | -0.16 | -0.18 | | |
| Source 5 | 0.05 | -0.26 | 0.14 | -0.19 | |
| Source 6 | -0.03 | 0.10 | -0.15 | -0.22 | -0.09 |

| QP site | Source 1 | Source 3 | Source 4 | Source 5 | Source 6 |
|---|---|---|---|---|---|
| Source 3 | -0.24 | | | | |
| Source 4 | -0.13 | -0.21 | | | |
| Source 5 | -0.15 | -0.25 | -0.13 | | |
| Source 6 | 0.18 | -0.24 | -0.11 | -0.28 | |
| Source 7 | -0.16 | 0.17 | -0.26 | -0.12 | -0.25 |

*Source 1: Vehicle exhaust; Source 2: Industrial source; Source 3: LPG usage; Source 4: Paint solvent usage; Source 5: Fuel production and evaporation; Source 6: Coal combustion; Source 7: Biomass burning

**Comment 16:** According to the PSCF results, regional transport of VOCs, especially from northern regions, also contributed to the VOC concentrations at 3 sites. While the VOCs were apportioned to local sources in the PMF model, which part of the VOCs was accountable for the regional transport?

**Response:**

Thank you for your comments. The region with high PSCF levels indicates high potential regional transport sources (Hui et al., 2018). Based on the PSCF results, at the JS and QP sites, high values were observed in the north of Shanghai. At the PD site, high values were observed in the northeast of Shanghai. All three sampling sites presented that the highest PSCF levels appeared in areas near the JS, PD and QP sites, suggesting that local source was a significant contributor to the VOC pollution compared with regional transport. Similarly, Song et al. (2021) studied the PSCF values at three different sites in Xi'an, China, and also showed that Xi'an had a strong local source, which was also consistent with the findings in Wuhan, China (Hui et al., 2018; Hui et al, 2019), suggesting that local source was a significant contributor to the VOC pollution. We added the description in the revised manuscript.

Page 16, lines 473-475:

"*All three sampling sites presented that the highest PSCF levels appeared in areas near the JS, PD and QP sites, suggesting that local source was a significant contributor to the VOC pollution compared with regional transport.*"

**Comment 17:** Line 472-473 and 497: How to quantify the percentage of OFP and SOAFP to $O_3$ and $PM_{2.5}$ concentrations?

**Response:**

Thank you for your comments. The percentages of OFP and SOAFP to $O_3$ and $PM_{2.5}$ concentrations could be calculated by the equations as followed.

$$\text{OFP to O}_3 = \frac{[\text{OFP}]}{[\text{O}_3]}$$

$$\text{SOAFP to PM}_{2.5} = \frac{[\text{SOAFP}]}{[\text{PM}_{2.5}]}$$

where [OFP] and [SOAFP] were the values of OFP and SOAFP, the $[O_3]$ and $[PM_{2.5}]$ were the $O_3$ and $PM_{2.5}$ concentrations. We added the description in the revised manuscript.

Page 16, lines 481-482:

"*The photo-induced transformation of VOCs could account for 69.15, 59.05 and 24.43 % of the $O_3$ concentrations in the above sampling sites (OFP values/$O_3$ concentrations).*"

Page 17, lines 506-508:

*"The SOAFP values accounted for 2.19, 0.95 and 1.02 % of the $PM_{2.5}$ concentrations at the above sampling sites (SOAFP values/$PM_{2.5}$ concentrations), all of which were lower than the results in Nanjing, China (3.46 %) (Mozaffar et al., 2020) and Wangdu, China (8.4 and 17.84 % under high-$NO_x$ and low-$NO_x$ conditions, respectively) (Zhang et al., 2020b)."*

**Comment 18:** As discussed in line 500-504, large uncertainty existed for the SOAFP method. Indeed, the OFP and SOAFP methods are only based on the reference reactivity of VOCs, which species with high reactivity tend to have larger secondary yields. The results herein are rather general and widely known. It is hard to tell what new findings we can get from this method.

**Response:**

Thank you for your comment. The objective of this study was not to calculate the values of OFP and SOAFP. One of the purposes of this study was to determine the effects of VOCs under the different land-use types on $O_3$ and SOA formation *via* calculating the OFP and SOAFP values. We deleted the uncertainties of the SOAFP in the revised manuscript. The results herein showed that the predominate OFP contributors were alkenes and aromatics, and the relevant emission sources, which are thought to be the industrial production and vehicle exhaust at the sampling sites, should be controlled in priority. Meanwhile, we found that the OFP values were closely related to the land-use types i.e., the higher OFPs at the JS and PD sites (50.85 ± 2.63 and 33.94 ± 1.52 ppb) relative to that at the QP site (24.26 ± 1.43 ppb) were observed. Under the high OFPs, the concentrations of $O_3$ at the JS and PD sites (73.59 ± 23.59 and 57.48 ± 20.49 μg m$^{-3}$) were unexpectedly lower than that at the QP site (99.30 ± 24.00 μg m$^{-3}$). In term of SOAFP values, the aromatics especially toluene was determined to be the main SOA contributor, and the SOAFP value at the JS site was significantly higher than those at the PD and QP sites due to the effect of land-use types. Because the close associations between VOCs and SOA could induce the sensitive response of VOC concentrations to the different pollution degrees of $PM_{2.5}$, we further explored the influence of VOCs on the atmospheric $PM_{2.5}$ abundance *via*

analyzing VOCs-PM$_{2.5}$ sensitivity. The results showed that the VOCs at the QP site showed a more rapid increment along with the increase of PM$_{2.5}$ values. The four groups of VOCs displayed similar linkages with VOCs, and the higher values of $k$ were attributed to the aromatics at the JS and PD sites, while the alkanes at the QP site. Thus, the optimal choices for controlling VOC species varied with the land-use types.
* * *
**Lastly, we would again express our appreciation to the reviewer and editor for their warmhearted help. Thank you very much!**
* * *
**References**

Cai, C., Geng, F., Yu, Q., An, J., Han, J.: Source apportionment of VOCs at city centre of Shanghai in summer, Acta Sci. Circumst., 30 (5), 926-934, https://doi.org/10.1631/jzus.A1000244, 2010b.

Cai, C., Geng, F. H., Tie, X. X, Yu, Q., and An, J. L., Characteristics of ambient volatile organic compounds (VOCs) measured in Shanghai, China, Sensors, 10, 7843-7862, https://doi.org/10.3390/s100807843, 2010a.

Chan, L., Chu, K., Zou, S., Chan, C., Wang, X., Barletta, B., Blake, D.R., Guo, H., and Tsai, W.: Characteristics of nonmethane hydrocarbons (NMHCs) in industrial, industrial urban, and industrial-suburban atmospheres of the Pearl River Delta (PRD) region of south China. J. Geophys. Res. 111 (D11304), 2005JD006481, https://doi.org/10.1029/2005JD006481, 2006.

Chen, C., Wang, L. R., Zhang, Y. J., Zheng, S. S., and Tang, L. L.: Spatial and Temporal Distribution Characteristics and Source Apportionment of VOCs in Lianyungang City in 2018, Atmosphere, 12, 1598, https://doi.org/10.3390/atmos12121598, 2021.

Dai, P., Ge, Y., Lin, Y., Su, S., and Liang, B.: Investigation on characteristics of exhaust and evaporative emissions from passenger cars fueled with gasoline/methanol blends, Fuel, 113, 10-16, https://doi.org/10.1016/j.fuel.2013.05.038, 2013.

Deng, C. X., Jin, Y. J., Zhang, M., Liu, X. W., and Yu, Z. M.: Emission Characteristics of VOCs from On-Road Vehicles in an Urban Tunnel in Eastern China and Predictions for 2017–2026, Aerosol Air Qual. Res., 18, 3025-3034, https://doi.org/10.4209/aaqr.2018.07.0248, 2018.

Haberer, K., Jaeger, L., and Rennenberg, H.: Seasonal patterns of ascorbate in the needles of Scots Pine (Pinus sylvestris L.) trees: Correlation analyses with atmospheric O$_3$ and NO$_2$ gas mixing ratios and meteorological parameters, Environ. Pollu., 139, 224-231, https://doi.org/10.1016/j.envpol.2005.05.015, 2006.

Han, D. M., Wang, Z., Cheng, J. P., Wang, Q., Chen, X. J., and Wang, H. L.: Volatile organic compounds (VOCs) during non-haze and haze days in Shanghai: characterization and secondary organic aerosol (SOA) formation, Environ. Sci. Pollut. Res., 24, 18619-18629, https://doi.org/10.1007/s11356-017-9433-3, 2017.

Hu, J., Zhao, T. L., Liu, J., Cao, L., Wang, C. G., Li, Y. Q., Shi, C. C., Tan, C. H., Sun, X. Y., Shu, Z. Z., and Li, J.: Exploring the ozone pollution over the western Sichuan Basin, Southwest China: The impact of diurnal change in mountain-plains solenoid, Sci. Total Environ., 839, 156264, https://doi.org/10.1016/j.scitotenv.2022.156264, 2022.

Hui, L. R., Liu, X. G., Tan, Q. W., Feng, M., An, J. L., Qu, Y., Zhang, Y. H., and Cheng, N. L.: VOC characteristics, sources and contributions to SOA formation during haze events in Wuhan, Central China, Sci. Total Environ., 650, 2624-2639, https://doi.org/10.1016/j.scitotenv.2018.10.029, 2019.

Hui, L. R., Liu, X. G., Tan, Q. W., Feng, M., An, J. L., Qu, Y., Zhang, Y. H., and Jiang, M. Q.: Characteristics, source apportionment and contribution of VOCs to ozone formation in Wuhan, Central China, Atmos. Environ., 192, 55-71, https://doi.org/10.1016/j.atmosenv.2018.08.042, 2018.

Liang, X. X., Huang, T., Lin, S. Y., Wang, J. X., Mo, J. Y., Gao, H., Wang, Z. X., Li, J. X., Lian, L. L., and Ma, J. M.: Chemical composition and source apportionment of $PM_1$ and $PM_{2.5}$ in a national coal chemical industrial base of the Golden Energy Triangle, Northwest China, Sci. Total Environ., 659, 188-199, https://doi.org/10.1016/j.scitotenv.2018.12.335, 2019.

Ling, Z.H., Guo, H., Cheng, H.R., and Yu, Y. F.: Sources of ambient volatile organic compounds and their contributions to photochemical ozone formation at a site in the Pearl River Delta, Southern China. Environ. Pollut. 159 (10), 2310-2319, https://doi.org/10.1016/j.envpol.2011.05.001, 2011.

Liu, Y., Shao, M., Fu, L. L., Lu, S. H., Zeng, L. M., and Tang, D. G.: Source profiles of volatile organic compounds (VOCs) measured in China: Part I, Atmos. Environ., 42(25), 6247-6260, https://doi.org/10.1016/j.atmosenv, 2008.

Liu, Y., Song, M., Liu, X., Zhang, Y., Hui, L., Kong, L., Zhang, Y., Zhang, C., Qu, Y., An, J., Ma, D., Tan, Q., and Feng, M.: Characterization and sources of volatile organic compounds (VOCs) and their related changes during ozone pollution days in 2016 in Beijing, China, Environ. Pollut., 257, 113599, 1-12, https://doi.org/10.1016/j.envpol.2019.113599, 2020.

Ma, T., Duan, F., He, K., Qin, Y., Tong, D., Geng, G., Liu, X., Li, H., Yang, S., Ye, S., Xu, B., Zhang, Q., and Ma, Y.: Air pollution characteristics and their relationship with emissions and meteorology in the Yangtze River Delta region during 2014-2016, J. Environ. Sci., 83, 8-20, https://doi.org/10.1016/j.jes.2019.02.031, 2019.

Mo, Z., Shao, M., and Lu, S.: Compilation of a source profile database for hydrocarbon and OVOC emissions in China, Atmos. Environ., 143, 209-217, https://doi.org/10.1016/j.atmosenv.2016.08.025, 2016.

Mo, Z., Shao, M., Lu, S., Qu, H., Zhou, M., Sun, J., and Gou, B.: Process-specific emission characteristics of volatile organic compounds (VOCs) from petrochemical facilities in the Yangtze River Delta, China, Sci. Total Environ., 533, 422-431, https://doi.org/10.1016/j.scitotenv.2015.06.089, 2015.

Nicolas, B.: Real-world emission factors for antimony and other brake wear related trace elements: size-segregated values for light and heavy duty vehicles, Environ. Sci. Technol., 21, 8072e8078, https://doi.org/10.1021/es9006096, 2009.

Qiao, Y. Z., Wang, H. L., Huang, C., Chen, C. H., Su, L. Y., Zhou, M., Xu, H., Zhang, G. F., Chen, Y. R., Li, L., Chen, M. H., and Huang, H. Y.: Source Profile and Chemical Reactivity of Volatile Organic Compounds from Vehicle Exhaust, Huanjing Kexue, 33, 1071-1079, 2012.

Ryou, H. G., Heo, J. B., and Kim, S. Y.: Source apportionment of $PM_{10}$ and $PM_{2.5}$ air pollution, and possible impacts of study characteristics in South Korea, Environ. Pollut., 240, 963-972, https://doi.org/10.1016/j.envpol.2018.03.066, 2018.

Schauer, J. J., Kleeman, M. J., Cass, G. R., and Simoneit, B. R. T.: Measurement of emissions from air pollution sources. 3. C-1-C-29 organic compounds from fireplace combustion of wood, Environ. Sci. Technol., 35, 1716e1728, https://doi.org/10.1021/es001331e, 2001.

Shi, J., Deng, H., Bai, Z., Kong, S., Wang, X., Hao, J., Han, X., and Ning, P.: Emission and profile characteristic of volatile organic compounds emitted from coke production, iron smelt, heating station and power plant in Liaoning Province, China, Sci. Total Environ., 515, 101-108, https://doi.org/10.1016/j.scitotenv.2015.02.034, 2015.

Song, C. B., Wu, L., Xie, Y. C., He, J. J., Chen, X., Wang, T., Lin, Y. C., Jin, T. S., Wang, A. X., Liu, Y., Dai, Q. L., Liu, B. S., Wang, Y. N., and Mao, H. J.: Air pollution in China: Status and spatiotemporal variations, Environ. Pollut., 227, 334-347, https://doi.org/10.1016/j.envpol.2017.04.075, 2017.

Song, M., Tan, Q., Feng, M., Qu, Y., Liu, X., An, J. and Zhang, Y.: Source Apportionment and Secondary Transformation of Atmospheric Nonmethane Hydrocarbons in Chengdu, Southwest China, J. Geophys. Res. Atmos., 123(17), 9741-9763, https://doi.org/10.1029/2018JD028479, 2018.

Song, M. D., Li, X., Yang, S. D., Yu, X. N., Zhou, S. X., Yang, Y. M., Chen, S. Y., Dong, H. B., Liao, K. R., Chen, Q., Lu, K. D., Zhang, N. N., Cao, J. J., Zeng, L. M., and Zhang, Y. H.: Spatiotemporal variation, sources, and secondary transformation potential of volatile organic compounds in Xi'an China, Atmos. Chem. Phys., 21, 4939-4958, https://doi.org/10.5194/acp-21-4939-2021, 2021.

Wang, J., Jin, L., Gao, J., Shi, J., Zhao, Y., Liu, S., Jin, T., Bai, Z., and Wu, C.-Y.: Investigation of speciated VOC in gasoline vehicular exhaust under ECE and EUDC test cycles, Sci. Total Environ., 445, 110-116, https://doi.org/10.1016/j.scitotenv.2012.12.044, 2013.

Wang, S., Zhao, Y., Han, Y., Gao, S., Li, R. and Fu, H. B.: Spatiotemporal variation, source and secondary transformation potential of volatile organic compounds (VOCs) during the winter days in Shanghai, China, Atmospheric Environment, 286, 119203, https://doi.org/10.1016/j.atmosenv.2022.119203, 2022.

Wongphatarakul, V., Friedlander, S. K., and Pinto, J. P: A comparative study of $PM_{2.5}$ ambient aerosol chemical databases. Journal of Aerosol Science, 29, S115-S116. https://doi.org/10.1016/S0021-8502(98)00164-5, 1998.

Xiong, Y., and Du, K.: Source-resolved attribution of ground-level ozone formation potential from VOC emissions in Metropolitan Vancouver, BC, Sci. Total Environ., 721, 137698, https://doi.org/10.1016/j.scitotenv.2020.137698, 2020.

Yang, M. R., Li, F. X., Huang, C. Y., Tong, L., Dai, X. R., and Xiao, H.: VOC characteristics and their source apportionment in a coastal industrial area in the Yangtze River Delta, China, J. Environ. Sci., 127, 483-493, https://doi.org/10.1016/j.jes.2022.05.041, 2023.

Yang, X. Y., Wu, K., Lu, Y. Q., Wang, S. G., Qiao, Y. H., Zhang, X. L., Wang, Y. R., Wang, H. L., Liu,Z. H., Liu, Y. L., and Lei, Y. : Origin of regional springtime ozone episodes in the Sichuan Basin, China: Role of synoptic forcing and regional transport, 278, 116845, https://doi.org/10.1016/j.envpol.2021.116845, 2021.

Yao, L.: Chemical composition, source and secondary generation of atmospheric $PM_{2.5}$ in typical areas of Shandong Province [D], Shangdong University, 2016.

Yao, Y. C., Tsai, J. H., and Wang, I. T.: Emissions of gaseous pollutant from motorcycle powered by ethanol-gasoline blend, Appl. Energy, 102, 93-100, https://doi.org/10.1016/j.apenergy.2012.07.041, 2013.

Zhang, H. D., Lü, M. Y., Zhang, B. H., An, L. C., and Rao, X., Q.: Analysis of the stagnant meteorological situation and the transmission condition of continuous heavy pollution course from February 20 to 26, 2014 in Beijing-Tianjin-Hebei, Acta Sci. Circumst., 36, 4340-4351, 2016.

Zhang, K., Li, L., Huang, L., Wang, Y. J., Huo, J. T., Duan, Y. S., Wang, Y. H., and Fu, Q. Y.: The impact of volatile organic compounds on ozone formation in the suburban area of Shanghai, Atmos. Environ., 232(137617), 1-11, https://doi.org/10.1016/j.atmosenv.2020.117511, 2020.

Zhang, Y., Wang, X., Zhang, Z., Lu, S., Shao, M., Lee, F. S. C., and Yu, J.: Species profiles and normalized reactivity of volatile organic compounds from gasoline evaporation in China, Atmos. Environ., 79, 110-118, https://doi.org/10.1016/j.atmosenv.2013.06.029, 2013.

Zhang, Y. C., Li R., Fu H. B., Zhou D., and Chen J. M.: Observation and analysis of atmospheric volatile organic compounds in a typical petrochemical area in Yangtze River Delta, China, J. Environ. Sci., 71, 233-248, https://doi.org/CNKI:SUN:HJKB.0.2018-09-022, 2018.

Zhao, S., Tian, H. Z., Luo, L. N., Liu, H. J., Wu, B. B., Liu, S. H., Bai, X. X., Liu, W., Liu, X.Y., Wu, Y. M., Lin, S. M., Guo, Z. H., Lv, Y. Q., and Xue, Y. F.: Temporal variation characteristics and source apportionment of metal elements in PM$_{2.5}$ in urban Beijing during 2018-2019, Environ. Pollut., https://doi.org/10.1016/j.envpol.2020.115856, 2021.

---

## Author Response (AR3)

Dear Prof. Wang,

We would like to thank the anonymous reviewer for the great efforts. On the basis of the reviewer's suggestions, we have updated this manuscript greatly. We greatly appreciate those comments and valuable suggestions from the reviewer. Also, we are grateful to your efficient serving for this manuscript.

Here we submit our revised manuscript **"Measurement report: VOC characteristics at the different land-use types in Shanghai: spatio-temporal variation, source apportionment, and impact on secondary formations of ozone and aerosol" (Manuscript number: acp-2022-250)**. In the attachments, a point-by-point response to each point raised from the reviewer was uploaded. The revised version marked for reviewing and the clean version for editing were supplied, respectively.

Yours sincerely,

Hongbo Fu

Department of Environmental Science & Engineering
Fudan University
200000 Shanghai China
E-mail: fuhb@fudan.edu.cn

**Comment 1:** Line 14: VOCs is precursor of SOA, which is accounted for large proportion of aerosol. It is Aerosol that have great impacts on climate change. I think the expression that VOCs have important impacts on climate change in not appropriate here.

**Response:**

Thank you for your questions. The aromatics accounted for large proportion of aerosol (Zhang et al., 2017; Li et al., 2020). We have rewritten the description in the revised manuscript.

Page 1, line 15:

"*Volatile organic compounds (VOCs) have important impacts on air quality, atmospheric chemistry and human health.*"

**Comment 2:** Line 62: Should be: "similar as/in agreement with the finding of Tang et al. (2008)".

**Response:**

Thank you for your comments. We have rewritten the description in the revised manuscript.

Page 2, line 61:

"*Kumar et al. (2018) found that the VOC concentration at the urban area was approximately twice higher than that at the rural area in Delhi, India, in agreement with the finding of Tang et al. (2008).*"

**Comment 3:** Line 213: $\Delta_{O3}$ should be $\Delta_{PM2.5}$.

**Response:**

Thank you for your comments. We have rewritten the description in the revised manuscript.

Page 8, line 213:

"*where $\Delta_{VOCs}$ and $\Delta_{PM2.5}$ is the concentrations of VOCs and PM$_{2.5}$ in the specific PM$_{2.5}$ gradients, respectively.*"

**Comment 4:** Line 228: $Ln(\Delta_{PM2.5}/B_{VOCs})$ should be $Ln(\Delta PM_{2.5.}/B_{PM2.5})$.

**Response:**

Thank you for your comments. We have rewritten the description in the revised manuscript.

Page 8, line 227:

> "where $k$ represents the linear coefficient between $ln(\Delta_{VOCs}/B_{VOCs})$ and $ln(\Delta_{PM2.5}/B_{PM2.5})$, $c$ is the intercept."

**Comment 5:** Line 251: The pearson correlation coefficient between VOCs and $PM_{2.5}$ at QP was low (0.25) and was much lower than the other two sites. It was not proper to make the conclusion that the elevated VOCs lead to the elevation of $PM_{2.5}$ at QP site, or VOCs and $PM_{2.5}$ at QP have similar emission sources.

**Response:**

Thank you for your comments. The pearson correlation coefficient between VOCs and $PM_{2.5}$ at QP was low (0.25), while the spearman correlation coefficient ($R_{Spearman}$) between the VOCs and $PM_{2.5}$ was 0.34 ($p < 0.01$) indicated that VOCs was positively correlated with $PM_{2.5}$. We have rewritten the description in the revised manuscript.

Page 9, line 260:

> "It was well documented that the elevated VOC concentrations indicated the increasing rate of $PM_{2.5}$ production via photochemical oxidation, gas-particle partition and/or heterogeneous absorption (Seinfeld et al., 2001; Yang et al., 2015; Han et al., 2017)."

We deleted the description that $PM_{2.5}$ and VOCs had similar emission sources in the revised manuscript.

**Comment 6:** Line 571: Is the much higher contribution of industrial sources lead to the relative lower contribution of vehicle in JS? Or do you have data of automobile density to support this conclusion?

**Response:**

Thank you for your comments. The JS site is located in the industrial regions favouring the contribution accumulations of industrial production and coal combustion which led to the relative lower contribution of vehicle exhaust compared with the PD and QP sites. We have written the description in the revised manuscript.

Page 19, line 574:

> "*It is interesting to note that the vehicle contribution was expected to be lower at the JS site than those at the PD and QP sites. This finding reflected the fact that the JS site is located in the industrial regions favouring the contribution accumulations of industrial production and coal combustion (Yoo et al., 2015).*"

**Comment 7:** "Under the high OFPs, ……These results……. (Yoo et al., 2015)." High OFPs is not always occurs synchronously with high $O_3$ episodes, since $O_3$ formation needs good photochemical conditions, which lead to strong photochemical removal of VOCs. These sentences were hard for me to understand, please consider rearrange the words.

**Response:**

Thank you for your comments. We have rewritten the description in the revised manuscript.

Page 20, line 614:

> "*The higher OFPs at the JS and PD sites (50.85 ± 2.63 and 33.94 ± 1.52 ppb) relative to that at the QP site (24.26 ± 1.43 ppb) were observed. However, the concentrations of $O_3$ at the JS and PD sites (73.59 ± 23.59 and 57.48 ± 20.49 µg $m^{-3}$) were lower than that at the QP site (99.30 ± 24.00 µg $m^{-3}$), indicating the poor $O_3$ formation conditions. Specifically, the locations of JS and PD sites resulted in the high emission strength, which could release to high pollutant concentrations and lead to severe atmospheric pollution (Cai et al., 2010a, b; Zhang et al., 2018). This phenomenon could change the strength of solar radiation and further decreased the intensity of $O_3$ photochemical reactions (Kumar et al., 2018).*"

**Comment 8:** Line 644: "The four groups of VOCs…." These sentences were hard for me to understand, please consider rearrange the words.

**Response:**

Thank you for your comments. We have rewritten the description in the revised manuscript.

Page 21, line 645:

"*The VOC compositions displayed similar linkages with VOCs, and the higher values of k were attributed to the aromatics at the JS and PD sites, while the alkanes at the QP site (Fig. S5).*"
* * *
**Lastly, we would again express our appreciation to the reviewer and editor for their warmhearted help. Thank you very much!**
* * *
**References**

Li, Q. Q., Su, G. J., Li, C. Q., Liu, P. F., Zhao, X. X., Zhang, C. L., Sun, X., Mu, Y. J., Wu, M. G., Wang, Q. L., and Sun, B. H.: An investigation into the role of VOCs in SOA and ozone production in Beijing, China, Sci. Total Environ., 720, 137536, https://doi.org/10.1016/j.scitotenv.2020.137536, 2020.

Zhang, G., Wang, N., Jiang, X., and Zhao, Y.: Characterization of ambient volatile organic compounds (VOCs) in the area adjacent to a petroleum refinery in Jinan, China, Aerosol Air Qual. Res., 17(4), 944-950, https://doi.org/10.4209/aaqr.2016.07.0303, 2017.